# Nonlinear shifts in infectious rust disease due to climate change

Joan Dudney [1,2✉], Claire E. Willing [2,3], Adrian J. Das [4], Andrew M. Latimer [1], Jonathan C. B. Nesmith[5] & John J. Battles[2]

Range shifts of infectious plant disease are expected under climate change. As plant diseases move, emergent abiotic-biotic interactions are predicted to modify their distributions, leading to unexpected changes in disease risk. Evidence of these complex range shifts due to climate change, however, remains largely speculative. Here, we combine a long-term study of the infectious tree disease, white pine blister rust, with a six-year field assessment of drought-disease interactions in the southern Sierra Nevada. We find that climate change between 1996 and 2016 moved the climate optimum of the disease into higher elevations. The nonlinear climate change-disease relationship contributed to an estimated 5.5 (4.4–6.6) percentage points (p.p.) decline in disease prevalence in arid regions and an estimated 6.8 (5.8–7.9) p.p. increase in colder regions. Though climate change likely expanded the suitable area for blister rust by 777.9 (1.0–1392.9) km$^2$ into previously inhospitable regions, the combination of host-pathogen and drought-disease interactions contributed to a substantial decrease (32.79%) in mean disease prevalence between surveys. Specifically, declining alternate host abundance suppressed infection probabilities at high elevations, even as climatic conditions became more suitable. Further, drought-disease interactions varied in strength and direction across an aridity gradient—likely decreasing infection risk at low elevations while simultaneously increasing infection risk at high elevations. These results highlight the critical role of aridity in modifying host-pathogen-drought interactions. Variation in aridity across topographic gradients can strongly mediate plant disease range shifts in response to climate change.

[1] Department of Plant Sciences, UC Davis, Davis, CA, USA. [2] Department of Environmental Science Policy and Management, University of California, Berkeley, Berkeley, CA, USA. [3] Department of Biology, Stanford University, Stanford, CA, USA. [4] U.S. Geological Survey, Western Ecological Research Center, Three Rivers, CA, USA. [5] Sierra Nevada Network Inventory & Monitoring Program, Three Rivers, CA, USA. ✉email: jdudney@berkeley.edu

nfectious plant diseases are reshaping ecosystems, disrupting global food supplies, and threatening human health[1–3]. Range expansions of maize lethal necrosis (MLN), for example, threaten global corn production[4], and both Dutch elm disease and chestnut blight fundamentally altered forests in North America[5,6]. Experimental and field studies have demonstrated that infectious diseases can be highly responsive to changes in temperature[7] and moisture conditions[8] and many studies predict range expansions of infectious plant disease under climate change[9–13]. Surprisingly little evidence, however, directly links climate change to plant disease range expansions[14,15] (though see ref. [16]). This may be a result of data limitations and the presence of confounding factors, such as land-use change and species translocations, that can often obfuscate the climate signal[1,17].

As infectious diseases extend beyond the leading edge of their range in response to climate change, contractions may also occur at the trailing edge where conditions become too hot and dry[11,18]. Infectious diseases have both high and low-temperature tolerances and moisture requirements, or climate optima[7,19,20]. This nonlinear, hump-shaped relationship indicates that climate change can shift the location of climate optima in space, leading to both increases and decreases in disease prevalence across the geographic range[21–23] (Fig. 1a,a.1). Recent studies demonstrating climate change-induced increases in disease used observational data spanning only a narrow portion of the climatic range[21–23].

While the climate–disease relationship in these regions is more likely monotonic, the relationship could become nonlinear under future climate change scenarios, leading to disease declines in some areas[14]. Though nonlinear disease range shifts (i.e., both increases and decreases in prevalence across a pathogen's range) have long been predicted[11,17], few studies have identified this relationship in situ[15].

Drought frequency and severity are often forecasted to increase under climate change[24], which could influence the prevalence of plant pathogens as they shift in space. Drought impacts on plant diseases are difficult to predict, however, as droughts can both reduce atmospheric humidity necessary for pathogen infection and reproduction[25–27] (though not always[28]), and concurrently increase host susceptibility[25,29]. In contrast, for plant pathogens that infect through stomata, increasing aridity may reduce infection rates by inducing stomatal closure, thereby decreasing the access of hosts to pathogens[2,30]. Drought impacts on plant pathosystems may also vary in magnitude across the aridity gradient of the pathogen's range. Thus predicted increases in drought frequency could result in more skewed and/or variable disease distributions under climate change (Fig. 1b, b.1).

Additionally, host–pathogen interactions could also mediate climate change-induced range shifts[31–33]. As disease distributions shift in space, biotic interactions with host populations may cause increases or decreases in prevalence relative to climate

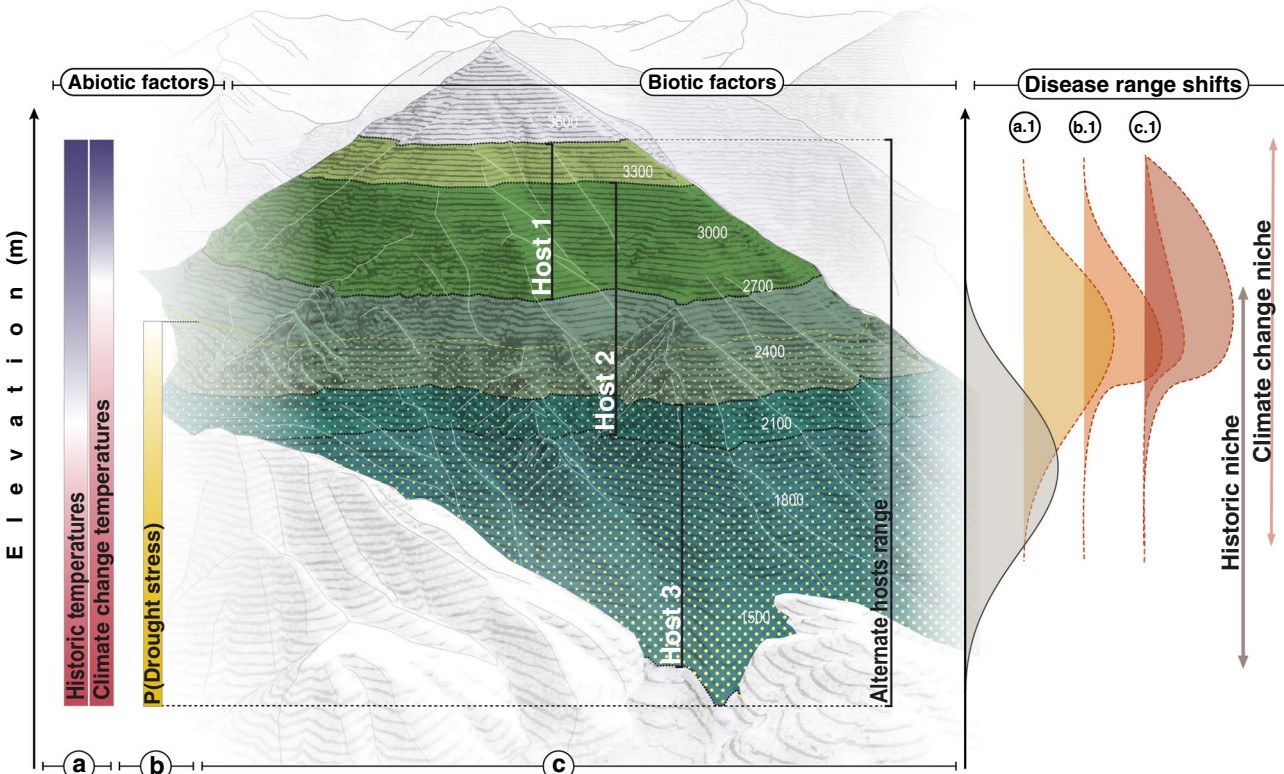

**Fig. 1 Conceptual figure of the biotic–abiotic factors that can interact with *Cronartium ribicola* as it moves into higher elevations in response to climate change (sensu the biotic–abiotic-migration (BAM) framework[15,109]).** These interactions can further modify the spatial position and shape of the white pine blister rust distribution. Specifically, climate change is predicted to shift the pathogens' climate optimum in space (**a**), which could lead to an upslope migration (**a.1**) (shift in temperature denoted by change in color gradient, hot = red, cold = blue). As the pathogen shifts, emergent abiotic and abiotic interactions can alter the disease distribution. Ongoing and/or climate change-induced increases in stochastic disturbances, like droughts (**b**), can interact with both the host and the pathogen at different spatial scales. The probability of drought stress, for example, is likely higher at low elevations where water is more limiting, potentially resulting in a skewed distribution (**b.1**). Additionally, host–pathogen interactions (**c**), including the varying density and susceptibility of different hosts and/or alternate hosts, can also modify the size and shape of the distribution. Hosts and alternate hosts are also expected to shift at a slower pace than pathogens in response to climate change, resulting in highly lagged impacts on disease prevalence. The combination of these spatially varying drought and host–pathogen interactions can modify disease range shifts to increase or decrease prevalence depending on the direction of the interacting effects (**c.1**). Mountain figure designed by Zuzanna Drozdz.

change impacts alone[17,29] (Fig. 1c, c.1). Infectious diseases emerging at higher altitudes or latitudes, for example, may encounter new host–pathogen pairings that modify infection probabilities, depending on the new host's spatial distribution and susceptibility[15,32,34]. Furthermore, if pathogen growth rates increase with rising temperatures, higher pathogen loads could result in elevated infection risk. Pathogen transmission could decline, however, if host survival rates also decrease under hotter, drier temperatures[35]. These complex, host–pathogen interactions[36] may lead to highly variable disease range shifts (Fig. 1c), resulting in less predictable climate change impacts on disease risk[17,32].

Though plant diseases are purportedly responsive to changes in temperature and moisture[7,14,37], surprisingly little evidence directly links climate change to geographic expansions (though see ref. [13]), and even less demonstrates nonlinear range shifts[11] (Fig. 1a.1). This is likely due to data limitations and the presence of confounding factors[17,38]. Measuring range shifts, for instance, requires a strong climate gradient and long-term data that capture disease absence, both of which are rare, particularly in regions where land-use change is not confounding[14,15,39]. Additionally, conclusive evidence that host–pathogen interactions and stochastic disturbance events, like drought, can modify prevalence as climate change shifts plant disease in space remains elusive, due in part to the dearth of longitudinal data that capture multiple stressors[40,41].

Here we test for a climate change signal on infectious tree disease by leveraging a long-term observational dataset of white pine blister rust (Cronartium ribicola Fisch., blister rust). The prevalence data included two surveys that occurred ~20 years apart, encompassing over 7800 individual hosts in Sequoia and Kings Canyon National Parks (SEKI). Blister rust infects white pines (Genus *Pinus*, Subgenus *Strobus*) across Europe and North America. As a result of the severe impacts, blister rust ranks as one of the worst tree pandemics in modern history[42]. To help control for potentially confounding factors and enable stronger causal inference[43], we combined a panel modeling approach with a mechanistic in situ test of drought impacts on physiological processes of host trees. Specifically, we asked whether: (1) climate change nonlinearly shifted disease prevalence, leading to geographic expansions and contractions within SEKI (Fig. 1a.1) and (2) spatially varying drought and host interactions modified the range shift, thereby changing the distribution and mean disease prevalence (Fig. 1b.1-c.1).

We found evidence that climate change between 1996 and 2016 moved the climate optimum of blister rust into higher elevations. This shift decreased prevalence by an estimated 5.5 percentage points (p.p.) with a 95% confidence interval of 4.4–6.6 p.p. in arid regions and increased prevalence by and estimated 6.8 (5.8–7.9) p.p. in colder regions. Warmer conditions also likely expanded the prevalence into higher elevations by an estimated 777.9 (1.0–1392.9) km² and contracted the range at low elevations by 4.7 (0.04–17.2) km². Forecasts under Representative Concentration Pathway 4.5 (RCP4.5) projected an even greater upward expansion from the counterfactual range limit (i.e., a no climate change scenario) by 1024.9 (1.0–1504.4) km², which would expose the majority of high-elevation white pines in SEKI. Though climate suitability for blister rust increased under climate change, host–pathogen–drought interactions contributed, in part, to a 32.79% decrease in observed prevalence between surveys. Drought and higher rates of mortality in arid regions, for example, likely accelerated disease declines at low elevations, while lower alternate host occurrence at high elevations dampened infection probabilities, even as the climatic conditions become more hospitable. We provide some of the first evidence that host–pathogen–drought interactions, which shifted in strength and direction across an aridity

gradientcan modify disease range expansions in response to climate change.

## Detecting a nonlinear climate–disease relationship

*Cronartium ribicola*, the causal agent of blister rust, is a macrocyclic heteroecious rust that requires the presence of both the pine hosts, as well as the alternate host species from the genera *Ribes, Castellja* and *Pedicularis*[44,45], to complete its life cycle (Fig. 2a). Blister rust is considered a cool weather disease and experimental studies have demonstrated thermal and moisture tolerances for both spore stages[46,47] (Supplementary Note 1). The four abundant white pine hosts (*Pinus* subgenus *Strobus*) in SEKI have overlapping ranges (Fig. 2a) and include sugar pine (*P. lambertiana* Dougl.), western white pine (*P. monticola* Dougl.), foxtail pine (*P. balfourniana* Grev. and Balf.), and whitebark pine (*P. albicaulis* Engelm.). These four species are all highly susceptible to impacts of blister rust. Here we define infection as an individual white pine stem expressing blister rust symptoms and prevalence as the percent of live infected individuals per plot[48].

The blister rust pathosystem in Sequoia and Kings Canyon National Parks (SEKI) was uniquely positioned for detecting a nonlinear climate–disease relationship, as well as a climate change fingerprint on infectious diseases, for five reasons. First, isolating nonlinear relationships is facilitated by strong environmental gradients and measures of disease absence[17,49]. The four white pine blister rust hosts in our study system span a large elevational gradient (1300–3500 m across some of the highest mountains in the contiguous United States) where there are historic records of disease absence (Fig. 2a). Second, because blister rust often occurs in montane zones, it has greater exposure to extreme weather and may respond more directly to changes in temperature and humidity[1,14,21]. Third, SEKI is located at the expansion front or current latitudinal range edge of blister rust invasion in California that began in British Columbia over 100 years ago[50]. Many have suggested that the climatic conditions are too hot and dry south of SEKI for blister rust to successfully disperse and infect hosts[9]; blister rust may be more strongly controlled by climate here than in more northern latitudes where the disease is more abundant. Fourth, studies of climate change effects on species distributions are often confounded by land-use change[51]. SEKI is a national park, however, and has experienced little land-use change in recent decades. Fifth, because blister rust is an invasive pathogen, frequencies of genetic resistance and associated mechanisms are very low (≤0.06) across southern Sierra white pines and alternate hosts[52]. Due to the long life expectancy of host white pines (~300–1500 years) and ontogenetic resistance[53], selection for genetically resistant individuals likely acts on longer times scales than our study captures (~20 years). Thus, changes in genetic resistance are unlikely to confound the climate–disease relationship in this study (see Supplementary Note 2 for more details).

## Results

**Nonlinear relationship between disease prevalence and VPD.** Between the first and second surveys (1996–2016), mean climatic conditions became hotter and drier compared to the previous 21 years (1975–1995) (Fig. 2b–d). Specifically mean annual vapor pressure deficit (VPD) increased from $9.74 \pm 0.87$ (mean ± standard deviation) to $11.15 \pm 1.0$ (hPa) (Fig. 2b) and mean minimum temperatures increased from $-2.48 \pm 0.68\,°C$ to $-1.24 \pm 0.80\,°C$ (Fig. 2d). In contrast, mean annual precipitation declined from $953.45 \pm 349.81$ mm to $870.82 \pm 329.79$ mm (Fig. 2c).

We used VPD as our climate predictor variable because it resulted in the most parsimonious model (Supplementary Table 1). VPD's statistical performance presumably reflects the biological reality that it integrates the climate variables that are critical

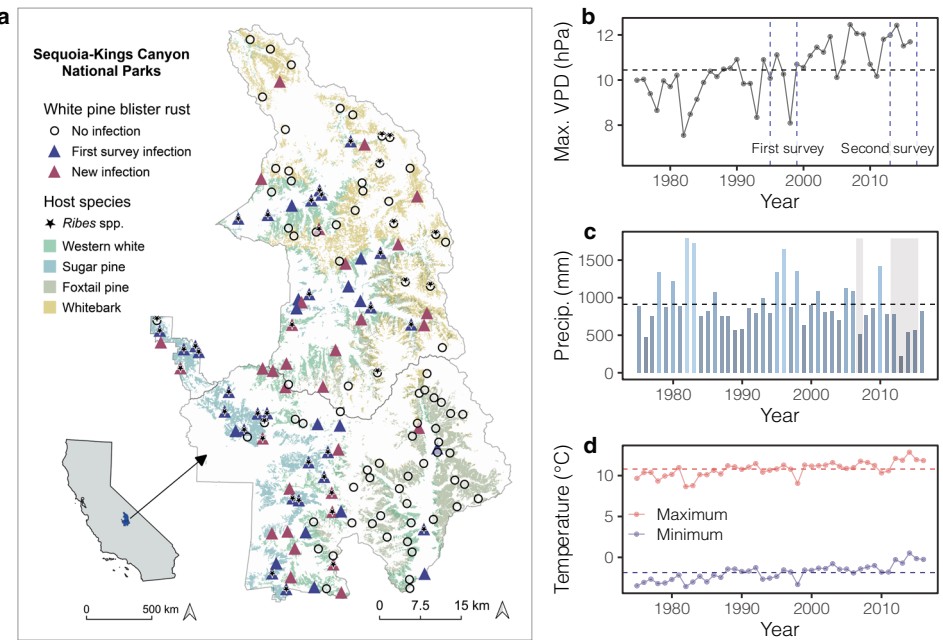

**Fig. 2 Site description. a** Map of Sequoia and Kings Canyon National Parks (SEKI) and the long-term monitoring plots. Includes plots with white pine blister rust infections in the first survey (purple triangles), plots with new infections in the second survey (red triangles), and plots that remained uninfected (black circles). Green shades on the map denote ranges of white pine hosts and stars illustrate plots with the presence of alternate hosts (*Ribes* spp.). **b** Mean annual vapor pressure deficit (VPD) between 1975 to 2016; blue dashed lines denote the time period of the first and second survey. **c** Total annual precipitation (blue bars; high precip. = light blue, low precip. = dark blue) across all plots. Drought years 2007 and 2012–2015 highlighted in gray rectangles. **d** Mean annual maximum (red line) and minimum (blue line) temperatures; dashed horizontal lines show the mean weather value across all years.

for blister rust reproduction—air moisture and temperature[54–56]. Previous studies of various pathosystems have also concluded that VPD can outperform other climate variables including temperature[57–59]. VPD is an absolute measure of atmospheric moisture state and is derived from the difference between the saturation vapor content of air at a specific temperature point and its actual vapor pressure[60]. In this study, high VPD values represent hotter and drier conditions, or more arid conditions (Fig. 2b).

The fixed effects (FE) panel model demonstrated that rising VPD between surveys was significantly and nonlinearly associated with changes in blister rust prevalence (Fig. 3e, f; Supplementary Table 2). Assessing how changes in VPD were associated with changes in blister rust prevalence using the FE panel model, we effectively controlled for time-invariant factors—topographic, edaphic, microclimate, host species identity, and unmeasured environmental variables—that might otherwise complicate causal interpretation of the estimated relationship between VPD and prevalence. The FE panel model therefore provided a robust estimate of the causal link between changes in VPD and changes in disease prevalence.

Additionally, both the positive and nonlinear relationship between blister rust and VPD were corroborated by generalized linear mixed effects models (GLMMs) estimating first and second survey infections, underscoring the persistent nonlinear relationship between VPD and disease prevalence (Fig. 3a–d; Supplementary Table 3). The nonlinear relationship with VPD and maximum temperature (Supplementary Table 4) suggested that there was an optimal climatic range for blister rust with climatic suitability declining beyond this optimal range, both in hotter, drier (more arid) regions, as well as colder, high-elevation systems (Fig. 3a, c). Additionally, these GLMMs provided insight into other environmental correlates of blister rust infections (a contrast to the FE panel model results, which controlled for the time-invariant

variables to isolate the effect of VPD). Alternate host occurrence, for example, was significantly correlated with both first and second survey infections (Fig. 3b, d), highlighting that close proximity of *Ribes* spp. was consistently important for blister rust infections in white pines through time.

The climatic gradient in SEKI likely did not capture the full climatic niche of blister rust, however. Peak infections occurred near the highest VPD ranges in the first survey (Fig. 4b), suggesting that the arid range limit existed south of SEKI, where blister rust has not been observed, but may have occurred at low frequencies. Additionally, replacing VPD with temperature in both the GLMMs (Supplementary Table 4, Supplementary Fig. 1) and FE panel models (Supplementary Table 6) resulted in similar estimates of the climate–disease relationship, but the models were less parsimonious and the second survey GLMM and FE model with temperature explained less of the variation. Finally, when we compared the performance of models with more flexible functional forms of VPD (e.g., quadratic vs. cubic terms), we found the quadratic to return the lowest AIC. Nevertheless, this difference was marginal and inconsistent across surveys (Supplementary Table 5). Therefore, we reproduced our primary results using a cubic VPD term and include these results in the supplement (Supplementary Fig. 2).

The consistency of the nonlinear relationship between the first and second survey periods indicated that the fundamental relationship between VPD and blister rust had not changed significantly though time due to unmeasured, stochastic factors (e.g., pathogen dispersal that was independent of climatic conditions and/or pathogen adaptation to local climatic conditions). Though there was a slight shift in the estimated peak prevalence when visually comparing the marginal effects of VPD between the first and second surveys (Supplementary Fig. 3), a test of the interaction between time and VPD demonstrated that

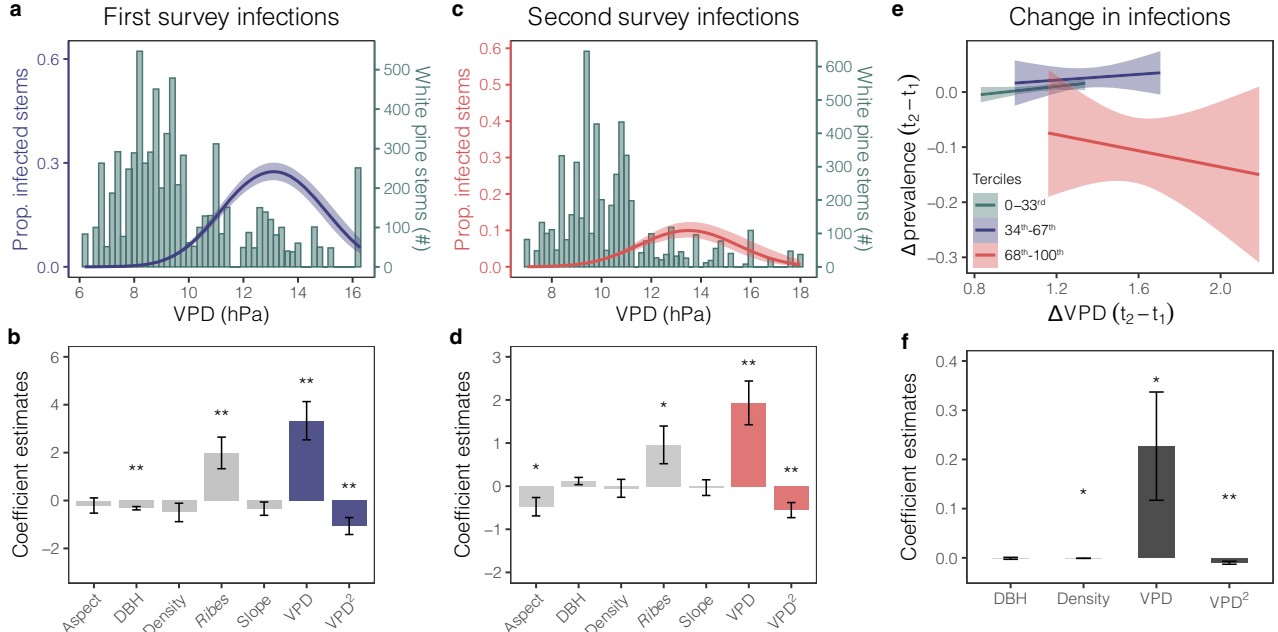

**Fig. 3 Nonlinear relationship between blister rust and VPD. a, c** Proportion of infected white pines (blue and red smoothed line estimated using a generalized linear logistic regression with a quadratic term) and the number of tree stems across vapor pressure deficit (VPD) (green bars) for the first (1995–1999) and second (2013–2016) surveys; blue and red shaded region represents the 95% C.I. band. **b, d** Coefficient estimates with corresponding standard error bars from logistic GLMMs explaining tree-level blister rust infections from the first ($n = 7031$) and second surveys (n = 5416). See Supplementary Tables 2 and 3 for all relevant statistics. Blue and red bars highlight the VPD terms. **e** Plot-level change in prevalence (proportion infected/ total live stems) as a function of the change in VPD between surveys. Plots were divided into VPD terciles defined by the second survey ($t_2$) (highlighted by three colors), thereby demonstrating both an increase in blister rust prevalence in cold regions (low VPD) and a decrease in prevalence in more arid regions (higher VPD). Showing shaded 95% C.I. bands; $t_1$ represents the first survey. **f** Coefficient estimates of four explanatory variables from the FE panel model with standard error bars (p-values include: tree density $p = 0.034$, VPD $p = 0.04$; VPD$^2$ $p = 0.001$; **$p < 0.01$, *$p < 0.05$).

this shift was not statistically significant (VPD*time: Estimate = −0.52, $P = 0.231$, VPD$^2$*time: Estimate = 0.23, $P = 0.169$; Supplementary Table 7). The stability of blister rust's relationship with VPD indicated that disease prevalence would track changes in climatic conditions (i.e., changes in VPD would affect blister rust prevalence), as the VPD optimum moved upslope into higher elevations in response to climate change.

**Estimated impact of climate change on disease prevalence.** To isolate the effect of rising VPD on prevalence and account for the FE panel model uncertainty in this estimation, we used a Monte Carlo (MC) simulation. Our results suggested that climate change decreased prevalence by 5.5 (4.4–6.6) p.p. in arid regions (lower elevation tercile) and increased prevalence by 6.8 (5.8–7.9) p.p. in the coldest regions (upper elevation tercile) (Fig. 4d). Warmer conditions resulted in a significant upward expansion into higher elevations by an estimated 381.5 m, which corresponded to an area expansion of 777.9 (1.0–1392.9) km$^2$ beyond the counterfactual range limit. At low elevations, disease prevalence contracted by an estimated 50.6 m, which corresponded to a 4.7 (0.04–17.2) km$^2$ area contraction in arid regions. Using a more flexible VPD term (cubic) resulted in a similar increase at high elevations and decrease at low elevations; however, the cubic model estimated an even greater increase at high elevations and a more moderate decrease at low elevations (Supplementary Fig. 2). While the direction of the results was consistent between quadratic and cubic models, the magnitude of the climate change effect was sensitive to changes in functional form. Therefore, the magnitude of the climate change effect should be interpreted cautiously. Furthermore, the range contraction is likely an underestimate of the contraction in the broader southern Sierra Nevada region. As we previously noted, our plots within SEKI

likely did not capture the full blister rust or sugar pine niches, both of which may have extended south of SEKI. Thus, the range contraction in the southern Sierra region may be greater than we measured within the geographic boundary of SEKI.

Future projections indicated that disease range shifts will continue if average conditions become hotter and drier (Fig. 4a). Incorporating the predicted uncertainty of 20 CMIP5[61] RCP4.5 (2056–2060) scenarios, as well as the FE model uncertainty, we estimated blister rust could expand from the counterfactual upper range limit by 1024.9 (1.0–1504.4) km$^2$, which would expose the majority of high-elevation white pine and alternate host species (Fig. 4e). In contrast, at low elevations, blister rust may continue to contract by an estimated 45.9 (0.04–236.8) km$^2$ (subtracted from the counterfactual range limit) (Fig. 4e), which could reduce disease prevalence in the hotter, drier regions of the sugar pine population. Additionally, if climatic conditions had warmed to the levels predicted under RCP4.5 (2020–2060), the prevalence may have declined by 13.7 (3.7–3.7) p.p. at low elevations and increased by only 3.9 (2.0–6.6) p.p. at high elevations (Fig. 4d), suggesting that even as the disease spreads into higher elevations, overall mean prevalence could decline further under climate change.

These projected changes in prevalence under RCP4.5 2056-60 do not account for future potential changes in drought severity and host distributional shifts that may also occur in response to future climate change. Though host species' recruitment rates vary spatially and by species[62], white pine demographic changes will likely lag behind blister rust expansion, particularly at high elevations where trees are very slow growing (pine hosts can live for ~1500 years) and slow recruiting (~0.2 %/year)[62]. Thus, using the past ~20 years of demographic change as a control in our projection of the next ~30 years is likely a reasonable

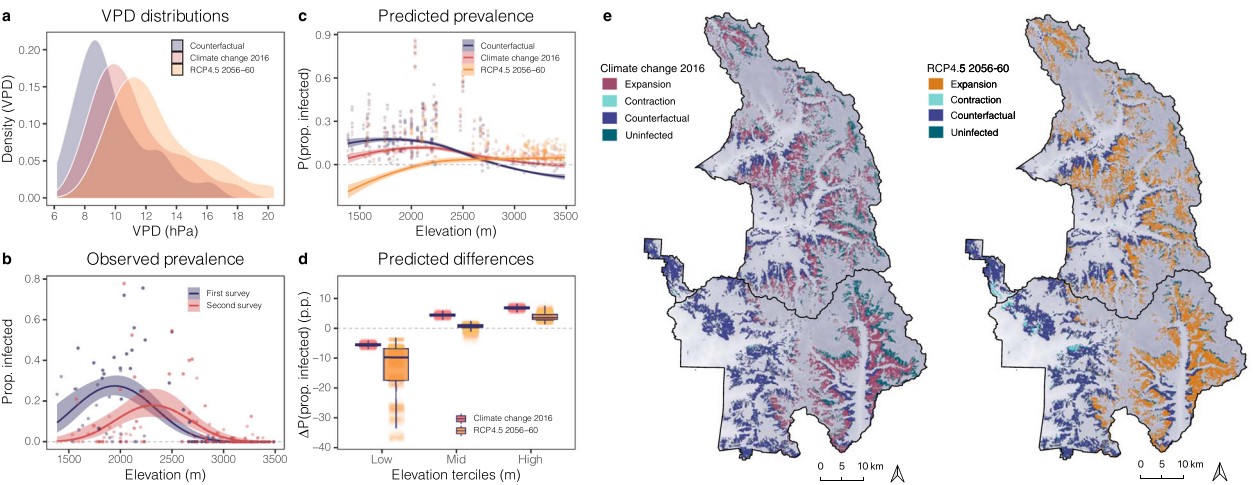

**Fig. 4 Climate change impacts on blister rust in SEKI. a** Distribution of mean vapor pressure deficit (VPD) between the two survey periods, 1975–1994 (blue) and 1995–2016 (red), as well as predicted VPD for 20 CMIP5 RCP4.5 (orange) scenarios. **b** Observed plot-level prevalence (proportion (prop.) infected trees/plot) across elevation for the first (blue) and second (red) surveys. Points are the local y maxima for quasi-binomial smoothed lines and 95% C.I. bands. **c** Random sample of predicted blister rust prevalence values, $n = 1000$ (sampled from the full Monte Carlo (MC) simulation for the three scenarios across elevation). Counterfactual is the no climate change scenario (blue), Climate change 2016 corresponds to the observed change in VPD between surveys (red), and RCP4.5 2056-60 corresponds to the predicted future climate change scenario (orange). Points represent a random sample of the MC simulation at the local y maxima for loess smoothed lines with 95% C.I. bands ($P$ = predicted; prop. = proportion). **d** Estimated percentage point (p.p.) difference ($\Delta$) in predicted prevalence attributed to a change in VPD estimated by the MC simulation. Climate change 2016 (red) and RCP4.5 2056-60 (orange) values were differenced from the counterfactual at the three elevation terciles (low = 1387–2655 m, mid = 2656–3132 m, and high = 3133–3486 m). Boxplots show the 25–75% quantile range and the 50% quantile center line. Whiskers depict data points within 1.5 times the interquartile range; includes jittered data points ($n = 1000$). **e** Estimated blister rust expansion (red and orange) and contraction (aqua) for the two climate change scenarios compared to the counterfactual (blue) in Sequoia and Kings Canyon National Parks. First map shows estimated expansion into higher elevation under observed climate change 2016 (red) compared to the estimated counterfactual (blue). Second map shows estimated expansion into higher elevations (orange) under RCP4.5 2056-60 and estimated contraction at lower elevation (aqua). Green shading in both maps illustrates the remaining uninfected host range. Gray gradient fill on both maps illustrates the digital elevation model (DEM; darker shading indicates higher elevation).

approach. In contrast, *Ribes* spp. often respond positively to fire[63]. Predicted changes in fire activity[64] may lead to greater *Ribes* spp. recruitment; we did not account for potential fire- or climate change-related *Ribes* spp. shifts in our future climate change projections. Finally, climate change may shift the blister rust pathosystem farther from equilibrium with the environment, which could lead to greater variability in disease prevalence and shift the current estimated climate–disease relationship. These combined effects, in addition to predicted changes in drought frequency and severity[24] (see below), could modify disease prevalence in the future. Consequently, our forecasts of climate change impacts on blister rust should be interpreted very cautiously.

**Drought and host interactions likely modified disease prevalence.** Though climate change shifted the climate optimum of blister rust into higher elevations, thereby increasing the number of vulnerable hosts to infection (Fig. 4e), mean observed prevalence declined between surveys. Specifically, increases in new infections at high elevations were more than offset by decreases at low elevations (Fig. 4b). In aggregate, this resulted in a 32.79% decrease in mean prevalence between surveys. Though many factors likely contributed to this decline, here we focus on the combination of host–pathogen interactions, a water availability gradient across elevation, and spatially varying drought impacts. We suggest that these abiotic-biotic interactions modified the disease distribution through two pathways: A) fewer alternate hosts at high elevations dampened infection probabilities, even as warmer conditions ameliorated the historic climatic constraints on spread and B) greater water deficit in arid regions decreased prevalence, an effect that was likely amplified by drought (Figs. 5, 6c, d).

**Pathway A: Fewer alternate hosts at high elevations dampened infection probabilities.** Host–pathogen interactions likely reduced infection probabilities at high elevations (Fig. 6a). GLMMs estimating blister rust infections, for instance, highlighted that *Ribes* spp. are important for white pine infection in this system (Fig. 3b, d), as it is necessary for blister rust to complete its life cycle[45]. *Ribes* spp., however, occur less frequently in colder, higher elevation regions in SEKI (Fig. 6a, Supplementary Table 8). Because fragile basidiospores travel short distances from *Ribes* spp. to infect white pines (meters to few kilometers)—which is a stark contrast to aeciospores that can infect *Ribes* spp. up to 1,200 km away[45]—the lower occurrence of *Ribes* spp. at high elevations may have suppressed white pine infection rates. Consequently, even as climatic conditions became more suitable for blister rust at high elevations, the probability of infection remained lower (Supplementary Fig. 4c, d). Infections at high elevations could increase in the future, however, if high-elevation *Ribes* populations densify and spread in response to climate change.

**Pathway B: Water deficit linked to declines in prevalence at low elevations.** Spatially varying abiotic interactions with drought and water availability gradients also likely modified the blister rust distribution as it shifted in response to climate change. To disentangle these complex, interacting effects, we combined a GLMM of infected host mortality (using data from our observational field study) with an in situ test of physiological responses to drought in sugar pine hosts. Our results highlighted two important mechanisms that contributed to faster declines in disease prevalence in arid regions (low elevations) compared to mesic regions (high elevations): 1) water deficit likely accelerated the rate of infected host mortality in arid regions, and 2) water deficit likely inhibited new infections in arid regions.

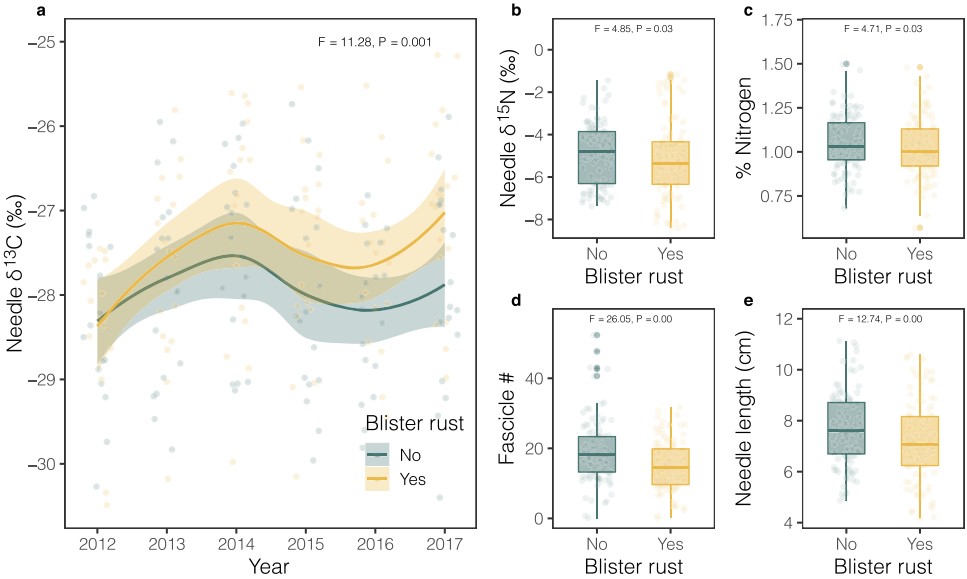

**Fig. 5 Differential responses to drought between infected and uninfected sugar pine hosts. a** Differences in δ13C between infected (yellow) and uninfected (green) hosts across years (n = 212); showing 95% C.I. bands. Drought years = 2012–15, post-drought = 2016–17; points are jittered and bands reflect 95% C.I. for respective treatments. **b** Boxplots showing differences in δ15N between uninfected (green) and infected (yellow) hosts across the study period 2012–2017 (n = 212). **c** Boxplots showing difference in needle % N (n = 212). **d** Differences in needle shedding (fewer fascicles) (n = 216) and (**e**) needle protraction (needle length) (n = 213) between infected and uninfected hosts. All boxplots show the 25–75% quantile range and the 50% quantile center line. Whiskers depict data points within 1.5 times the interquartile range; includes outlier points in dark shades and jittered data points in light shades. Infected trees highlighted by yellow and uninfected by green. All panels show F- and p-values estimated from repeated measures analysis of variance (one-sided) models (see Supplementary Note 3).

*Mechanism 1*: water deficit likely accelerated the rate of infected host mortality in arid regions. Specifically, the proportion of dead to live infected trees increased with aridity (Supplementary Fig. 5). Thus, infected trees were more likely to die in arid regions (Fig. 6b, Supplementary Table 9). This elevated mortality led to faster declines in disease prevalence in arid regions that was not offset by new infections in arid or mesic regions (see Pathway A above). Many factors contributed to this pattern of mortality. Sugar pines growing in the most arid regions of SEKI likely experienced higher levels of water deficit—both during drought and non-drought years —that amplified physiological responses to infections. Our in situ drought study, for example, demonstrated that growth and carbon sequestration declined more in infected hosts than uninfected hosts during the extreme drought between 2012–2015. The combination of tighter stomatal regulation and greater changes in carbon allocation during drought (Fig. 5) indicated that infected trees experienced greater water stress at the cost of carbon acquisition[65] (Supplementary Note 3). These physiological shifts were consistent with mortality trends suggesting that infected hosts were more likely to die (Estimate = 15.61, P = 0.065) (Supplementary Fig. 6) than uninfected hosts following drought. Thus, greater water deficit had more severe physiological consequences for infected trees during drought. This water deficit may have contributed to higher rates of mortality in arid regions, both during drought, as well as non-drought years (i.e., infected host mortality was higher on average in arid regions compared to more mesic regions (Fig. 6b)).

Though water deficit had more severe consequences for infected hosts, the cause of death was likely a combination of factors related to altered host physiology and increased vulnerability to biotic attack[66]. Both mountain pine beetle (*Dendroctonus ponderosae*) and bark weevils (*Pissodes* spp.) can preferentially select trees weakened by blister rust and/or drought[62,67]; though during extreme drought, bark beetles can kill trees indiscriminately[68]. Over the past ~20 years, these biotic attacks occurred more frequently at low elevations in SEKI where aridity is higher and pests and pathogens

are more abundant[62,68]. Additionally, recent climate change may have increased physiological stress in infected sugar pine growing near their range limits, potentially increasing the probability of mortality[62] from blister rust infections or other biotic agents. Thus trees growing in arid regions were more likely to die following infection, and drought events may have accelerated mortality, particularly in arid regions.

Importantly, our in situ drought study suggested that drought impacts varied across elevation, negatively impacting growth in arid regions but positively impacting growth at high elevations (Fig. 6c, d). While sugar pines exhibited signs of drought stress (Figs. 5, 6d) at low elevations, whitebark pine at high elevations demonstrated increased needle expansion (Fig. 6c). Increased growth may have been a result of an extended growing season during drought that was driven, in part, by decreased snowpack and warmer air temperature[69]. The drought-stress gradient in SEKI was further corroborated by a recent study demonstrating that the highest probability of mortality during the drought occurred between 1000 and 2000 m[70], which largely captures the sugar pine range. Thus, the impacts of drought on host physiology were spatially varying, likely increasing infection probabilities at high elevations and amplifying disease declines in arid regions (see Mechanism 2 below).

*Mechanism 2*: water deficit likely inhibited new infections in arid regions. Two lines of evidence supported this mechanism. First, faster rates of infected host mortality in arid regions likely suppressed the number of new infections in *Ribes* spp. Blister rust has an obligate sexual life cycle with white pine and alternate hosts (e.g., *Ribes* spp.) that is dependent on infrequently occurring wave years (i.e., climatic conditions that are suitable for blister rust reproduction and spread, estimated to occur every 5–10 years[48]). Most alternate hosts are deciduous and drop their foliage in fall, thereby shedding infections each year[71]. The presence of live pine hosts when the next wave year occurs is therefore critical for blister rust to spread from the pine host to

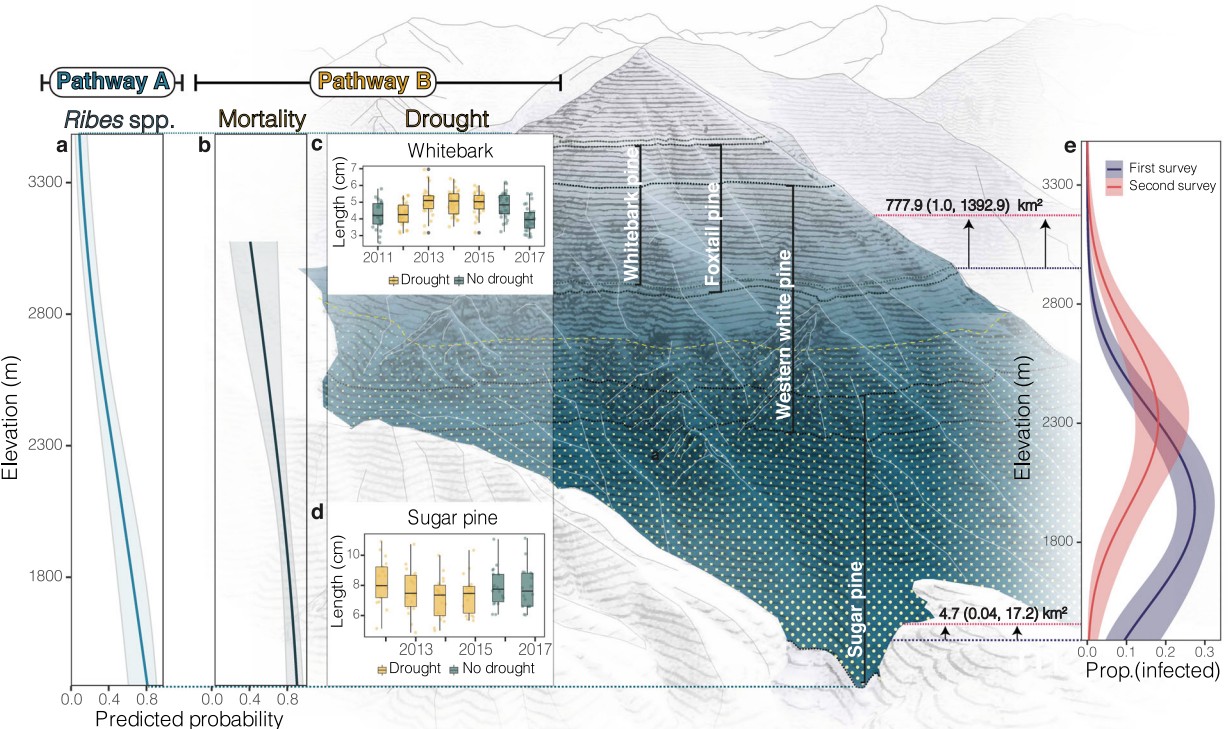

**Fig. 6 Spatially varying abiotic-biotic interactions that likely modified blister rust range shifts. a** Predicted probability of *Ribes* spp. occurrence (blue line) across elevation with 95% C.I. bands. Blue shade on mountain displays the corresponding declining abundance of *Ribes* spp. hosts with increasing elevation. **b** Predicted mortality of infected hosts (black line) across elevation; showing 95% C.I. bands. **c, d** Boxplots show difference in needle expansion (length) between high and low elevation regions during drought (yellow) and non-drought (green) years for whitebark pine (*n* = 193) and sugar pine (*n* = 107). Boxplots show the 25–75% quantile range and the 50% quantile center line. Whiskers depict data points within 1.5 times the interquartile range; includes outlier points in dark gray and jittered data points in light shades. Yellow dots on the mountain show the predicted range in which drought led to decreases in white pine host needle growth. **e** Observed blister rust prevalence (prop. = proportion) from the first (red) and second (blue) surveys across elevation; includes quasi-binomial smoothed lines and 95% C.I. bands. Black vertical lines on the mountain illustrate the ranges of the susceptible white pine hosts. Mountain figure designed by Zuzanna Drozdz.

the alternate host to complete its life cycle[45]. Consequently, higher rates of infected host mortality had cascading impacts on disease risk and may have accelerated disease declines, both in arid regions and regionally (within a 1,200 km radius[45]).

Second, water deficit that occurred more frequently in arid regions likely reduced susceptibility in white pine hosts. Our in situ drought study demonstrated that infected hosts exhibited tighter stomatal regulation during drought (Fig. 5a), which has been shown to reduce infection probabilities of pathogens, like *Cronartium ribicola*, that enter hosts through stomatal pores on the needle[30,72]. Thus, the probability of new infections at low elevations may have also declined during drought[14], as both infected and uninfected hosts demonstrated signs of water deficit and tighter stomatal regulation as the drought progressed (Fig. 5a). Furthermore, the production of teliospores on *Ribes* spp. has been linked to periods of high humidity. The drought-induced reductions in white pine host exposure to blister rust (i.e., stomatal closure leading to less exposed needle surface area), combined with the predicted decline in teliospore abundance and spread during drought[72] highlight how disease likely declined both during drought, as well as in arid regions of the blister rust's range in non-drought years[73].

## Discussion

Here we demonstrated that observed climate change between 1996 and 2016 in SEKI shifted the climate optimum for blister rust upslope. Although climate change likely expanded the geographic area suitable for blister rust infection in SEKI, the change in disease prevalence was nonlinear, resulting in a decrease in blister rust prevalence at low elevations and an increase at high elevations (Fig. 4d). Even with small increases in mean VPD between surveys, the impact on the blister rust pathosystem was considerable—an effect that has long been predicted[15,29]. Under future climate change scenarios, blister rust prevalence may contract further at low elevations, where climatic conditions become too hot and dry. In contrast, more hospitable climatic conditions at high elevations may simultaneously expose the majority of high-elevation white pine and alternate hosts in SEKI (Fig. 4e). These results present a robust estimate of a climate change-induced range shift in an infectious plant disease.

Though blister rust expanded its range into higher elevations in SEKI, climate change contributed to a surprising decline in mean infection risk due to complex, spatially varying host–pathogen–drought interactions. These results highlighted that the direct effect of climate change (e.g., a range expansion) can be strongly mediated by indirect effects (e.g., complex interactions). Host–pathogen interactions, for example, likely suppressed infection probabilities at high elevations due to a decline in alternate host species (Fig. 6a). These emergent biotic interactions have long been anticipated[1,17,41,74], and we provide evidence supporting the prediction that host availability is critical for forecasting shifts in endobiotic pathogen distributions[32]. Additionally, drought events, as well as an aridity gradient, interacted with white pine hosts and the pathogen, leading to faster declines in infections in arid regions that were not offset by

increases at high elevations (Fig. 6e). Arid regions, for instance, experienced higher rates of infected host mortality and lower probabilities of infection, particularly during drought (Fig. 6b–d). Thus, the blister rust range shift was modified by abiotic-biotic interactions that varied in strength and direction across an aridity gradient. Furthermore, these interactions may become nonlinear as global warming proceeds[75,76]. Investigating the context dependencies of host–pathogen–drought interactions (e.g., aridity gradients, forecasted shifts in drought frequencies, nonlinear interactions, and spatially varying host species distributions) will help disentangle the indirect effects of climate change on plant disease risk.

Though climate change is often predicted to increase susceptibility to pathogens at the host range edge, our results suggest the opposite. The thermal mismatch hypothesis, for example, predicts that host susceptibility increases as climate change shifts optimal temperatures away from hosts[29], which has been found in infectious amphibian disease[77]. Drought has also been associated with increased host susceptibility to infection. For example, root rot pathogen genera, including *Armillaria* and *Heterobasidion*, are predicted to expand their ranges during drought because they successfully colonize stressed trees[9,78]. In contrast, climate warming and drought impacts on pathogens that infect through needle stomata likely have opposing effects on host susceptibility across an aridity gradient. Specifically, hotter, drier conditions during drought increased infection probabilities at the leading range edge in SEKI, where the growing season was extended (Fig. 6c), but decreased infection probabilities at the arid range edge where growth declined (Fig. 6d). Identifying how host infection surfaces (i.e., roots, leaves, stem tissue) respond to warming and water deficit across aridity gradients will be critical to forecast climate change-induced disease range shifts.

Blister rust expansion into the subalpine suggests that climate change may pose grave risks to high-elevation forests. The low diversity of tree species that can persist in extreme climatic conditions have often evolved with low frequencies of major biotic disturbances[79,80]. Though climate change has been linked to increased growth and expansion of subalpine forests[81], these forests are also more vulnerable to endemic and novel pathogens[80,82]. Increases in blister rust and mountain pine beetle in western North America, for example, contributed to the widespread decline of whitebark pine, recently proposed for listing under the Endangered Species Act[83]. In contrast, whitebark pine in the southern Sierra Nevada historically experienced some of the lowest rates of white pine blister rust infection compared to more northern regions of its range. The projected expansion of blister rust under climate change, however, reduces the likelihood that the southern Sierra Nevada remains a blister rust refugium[62] for whitebark pine. Additionally, while disease risk is predicted to decline further in lower elevation white pines (Fig. 4d, e), this may not result in a host population rebound, but rather a shift in the forest community[84]. The compounding effects of blister rust[48] and predicted increases in drought severity[24] will likely continue to threaten the long-term sustainability of host populations.

## Methods

### Isolating the climate–disease relationship

*Long-term blister rust.* We used tree and plot-level data from two surveys, conducted between 1995–1999 and 2013–2017 that spanned 147 plots containing a total of 7,809 white pine stems (Fig. 2a). Tree-level data measured in the field included diameter at breast height (1.37 m, DBH), the occurrence of blister rust symptoms (see Supplementary Note 4 for details), and mortality (alive = 0, dead = 1). Plot-level data measured in the field included elevation (m), slope (°), aspect (°; south, southeast, and southwest facing = 1; north, northeast, and northwest facing = 0), presence/absence of alternative host species (i.e., *Ribes* spp.), and white pine host density (# trees/ha).

*Climate data.* We used downscaled 4 km resolution PRISM historical climate data[85], including monthly averages of daily maximum VPD (hPa), precipitation (mm), max and min temperatures (°C), and mean dewpoint temperature (°C). To compare how climate had changed over time, we averaged the climate variables across two time periods of equal length: 1) the time period between the first and second surveys (1996–2016) and the previous 21 years before the first surveys (1975–1995). These averaged VPD values were used in all statistical models (see below).

We obtained predicted 4 km resolution VPD for the Representative Concentration Pathway RCP4.5 2056–2060 from 20 Coupled Model Inter-Comparison Project 5 (CMIP5) experiments, available through the Multivariate Adaptive Constructed Analogs MACAv2-METDATA[86]. These datasets used the gridMet[87] daily dataset (4 km grid from 1979–2012) that is a combination of PRISM and NLDAS-2[88] data. We calculated the mean predicted percent change for each plot between the time periods 2016–2020 and 2056–2060 and used the percent change for each plot to estimate the predicted percent change in maximum VPD (Supplementary Fig. 7). We did not use historical VPD from MACAv2-METDATA dataset because these estimated averages do not correspond to the actual historical values; MACA data are not meant to be used as a hindcast of weather. RCP4.5 was selected as a moderate, reasonably likely emissions pathway[89]; the time period 2056–2060 was selected to be relatively consistent with the previous sample period.

*Estimating the relationship between climate and disease prevalence.* We used complementary analyses to rigorously quantify the impacts of climate change on blister rust prevalence. To test whether blister rust was nonlinearly related to climate, we fitted two generalized linear mixed models (GLMMs) explaining first and second survey tree-level infections. These two disease models also helped identify the important variables that explained the probability of an infection occurring at both time periods. To better estimate the causal link between climate and prevalence, we used a fixed effects (FE) panel model that allowed us to control for potentially confounding, time-invariant factors. We then used the estimated FE panel model to simulate the effect of climate change and compared these results to a counterfactual of no climate change. All statistical analyses were conducted with R software[90].

First, to identify the climate variable that resulted in the most parsimonious disease model, we fitted GLMMs that estimated blister rust infections as a function of five climate variables and a null model without climate. Independent variables included *Ribes* spp. occurrence (presence/absence), DBH (cm), tree density (# trees/ha), slope (°), and aspect (°). We compared AIC values (Supplementary Table 1) to select the most parsimonious model. The model with annual maximum VPD had the lowest AIC score among all climate variables. This result presumably reflects the biological reality that VPD integrates two climate variables that are critical for blister rust reproduction—air moisture and temperature[54–56]. VPD is an absolute measure of atmospheric moisture state and is derived from the difference between the saturation vapor content of air at a specific temperature point and its actual vapor pressure[60]; high VPD values represent hotter and drier conditions (Fig. 2b).

To explain the probability of infection at the time of the first and second survey, we developed two GLMMs that estimated: 1) probability that a tree was infected at the time of the first survey using first-survey infection status of all live trees as the (0/1) response variable ($n = 7,031$); and 2) probability that an uninfected tree became infected between the first and second surveys using second survey infection status of live, previously uninfected trees as the (0/1) response variable ($n = 5,416$). For both GLMMs, the explanatory variables included a quadratic VPD term to allow for potentially nonlinear disease response and the following independent variables: *Ribes* spp. occurrence (presence/absence), DBH (cm), tree density (# trees/ha), slope (°), and aspect (°). We included species and plot as crossed random effects. All non-binary variables were standardized across both survey periods data to have a mean of zero and a standard deviation of 1. We only included live trees that were uninfected at the time of the first survey ($t_1$) in the second survey models ($t_2$) in part to test whether unmeasured variables (e.g., stochastic dispersal dynamics, pathogen climate adaptation, and/or changes in virulence) may have affected the probability of infection following the first survey (Supplementary Fig. 3). We also verified that models that included all trees (i.e., both live and dead trees) gave similar coefficient estimates and p-values (Supplementary Table 10).

Additionally, we tested whether independent variables were highly correlated (Pearson $r > 0.5$). Because elevation was highly correlated with other variables (Supplementary Fig. 8), we followed previous blister rust modeling studies[56,91] and excluded elevation from the model to reduce multicollinearity[92]. We verified that each model was not overdispersed following methods outlined by Kabacoff (2011). Goodness of fit estimates were extracted for each model using conditional pseudo-$R$ squared[93]. Finally, though blister rust dispersal distances between white pine hosts and alternate hosts were much greater than the spatial scale of this study (i.e., aeciospores can travel up to 1200 km[45], which is a much greater distance than SEKI encompasses: ~100 km × 70 km), we verified that plot random effects effectively controlled for spatial-autocorrelation in our GLMMs following methods outlined by Zuur et al.[94].

We also estimated first and second survey blister rust infections using maximum temperatures (Supplementary Fig. 1, Supplementary Table 4). These results were consistent with our VPD analysis. However, the models were less parsimonious and explained a lower proportion of the variation in the second

survey model. To determine whether a quadratic VPD term or a more flexible cubic term was preferable in this study, we fit first and second survey GLMMs described above with a cubic VPD term and compared models using AIC (Supplementary Table 5). We found that the quadratic term resulted in a slightly more parsimonious model. Moreover, the overall interpretation of our model results—that climate change led to both increases and decreases in prevalence across elevation and a range expansion of disease—does not change between the quadratic and cubic models (Supplementary Note 5, Supplementary Fig. 2).

Furthermore, we added a robustness check to test whether unmeasured variables were potentially biasing our GLMM coefficient estimates. A significant change in the relationship between VPD and blister rust through time, for example, could suggest that unmeasured variables (e.g., stochastic dispersal dynamics, changes in pathogen virulence, and/or pathogen climate adaptation), were biasing our climate–disease relationship. Specifically, we tested whether the VPD-blister rust relationship changed through time by fitting GLMMs that contained an interaction between time period and the linear and quadratic VPD terms[95] (Supplementary Note 6). Significant interactions with time would suggest that the relationship between infection and VPD had changed, likely due to unmeasured variables. We found that the interaction terms were not significant (Supplementary Table 7), indicating that these unmeasured variables were not confounding our VPD coefficient estimates. We also visually assessed the VPD-blister rust relationship using the predicted marginal effects of VPD across the same standardized VPD range (from R package ggeffects[96], which varied VPD while holding non-focal variables constant using a proportional average) (Supplementary Fig. 3).

To isolate the relationship between VPD and prevalence, we used a fixed effects (FE) panel model that can enable causal inference[43]. Though studies estimating climate change impacts on disease have long-applied generalized linear modeling (GLM) approaches[23,51,95], these models have also been criticized, in part because they often do not control for endogeneity bias (i.e., when a statistical model's explanatory variables are correlated with the error term)[43,97]. Endogeneity bias can occur when correlated variables are omitted (i.e., omitted variables bias), there is feedback between the outcome variable back to the explanatory variables (i.e., reverse causality), or measurement error in the outcome variable is correlated with the explanatory variables[98]. Using a FE panel model, we effectively controlled for time-invariant endogeneity bias from observable and unobservable variables. Elevation (an observable variable), for example, was highly correlated with VPD and blister rust. White pine and alternate host species were also correlated with VPD and elevation, making it difficult to isolate the VPD-blister rust relationship using GLMMs. The FE panel model, however, controlled for both unobservable time-invariant variables (e.g., soil type, solar radiation) and observable variables, including elevation, and white pine and alternate host species effects (i.e., *Ribes* spp.), thereby better isolating the causal link between VPD and blister rust.

Specifically, the FE estimated the relationship between changes in VPD and changes in blister rust between surveys[43] (Fig. 3e, f). Our outcome variable was the proportion of live trees in plot $i$ at time $t$ that were infected by blister rust ($y_{it}$ = #infections/total live trees). We modeled this outcome as a function of plot-level independent variables, including slope, aspect, tree density, VPD, VPD$^2$, elevation, *Ribes* spp., plot, white pine species, and DBH. We derived the FE equation as follows[98]:

$$y_{it} = \beta_1 x_{it} \ldots + a_i + \delta_t + u_{it}, t = 1, 2 \qquad (1)$$

for each plot ($i$), we averaged the equation over time ($t$), yielding:

$$\bar{y}_i = \beta_1 \bar{x}_i \ldots + \delta + a_i + \bar{u}_i \qquad (2)$$

Subtracting (2) from (1):

$$\ddot{y}_{it} = \beta_1 \ddot{x}_{it} \ldots + \ddot{\delta}_t + \ddot{u}_{it}, t = 1, 2 \qquad (3)$$

where $\ddot{y}_i = y_{it} - \bar{y}_i$ is the time-demeaned blister rust prevalence, $\ddot{\delta}_t$ is the time fixed effect, and $\beta_1 \ddot{x}_{it} = \beta_1 x_{it} - \beta_1 \bar{x}_i \ldots$ represent the time-demeaned fixed effects (e.g., plot, white pine species, VPD). Subtracting the mean value from each dependent and independent variable mathematically subtracted out the observed time-invariant variables (including plot, elevation, slope, aspect, white pine host, and *Ribes* spp.), and the unobserved time-invariant variables ($a_i$). In this way, the panel model allowed us to control for time-invariant but unobservable variables which might otherwise confound our analysis[43] of the relationship between VPD and blister rust. What remained in the model were the time-varying variables (i.e., changes in tree size (growth or DBH), changes in tree density (due to mortality), and changes in VPD), as well as the exogenous time-varying error ($\ddot{u}_{it}$) (Eq. 3).

We estimated the FE panel model using the feols function from the fixest package[99]. To control for temporal serial correlation in the error terms within plots, standard errors were clustered by the plot. We also estimated the FE panel model using maximum temperatures instead of VPD to show that these results can be interpreted in temperature space but that VPD explains more of the variation (Supplementary Table 6). While FE panel model approaches allow strong control of time-invariant unmeasured variables, we were unable to control for unobservable time-variant variables. However, we believe that we adequately addressed the unmeasured variables that would likely bias our results—stochastic dispersal patterns, climate adaptation (Supplementary Note 6, Supplementary Table 7), and changes in genetic resistance (Supplementary Note 2).

We simulated the effect of climate change on blister rust prevalence using the estimated parameters from the FE panel model. Our approach predicted prevalence for three different scenarios: 1) a counterfactual of no climate change over the past ~20 years using historic VPD between 1975–1995 (counterfactual), 2) observed climate change over the past ~20 years using VPD between 1996–2016 (climate change 2016), and 3) predicted climate change under RCP4.5 2056-60 using 20 CMIP5 experiments of VPD projections (RCP4.5 2056-60). The counterfactual provided an estimate of prevalence if no climate change had occurred between surveys, but other observed measured variables had changed, including tree density and growth (i.e., we isolated the effect of VPD by controlling for changes in the remaining time-varying variables). This counterfactual scenario was compared to the climate change scenarios to produce estimates of the observed and future climate change impacts.

Specifically, we used the R function predict from the R stats package rms[100] to obtain predicted prevalence values for the three scenarios. All predictions included the same demographic changes in mean DBH and tree density but varied VPD by the three scenarios, thereby isolating the effect of a change in VPD on blister rust prevalence. The uncertainty of the panel model predictions was captured using a Monte Carlo simulation. First, we randomly sampled 10,000 draws of the FE panel model coefficients using their estimated mean values and variance-covariance matrix. Then we predicted prevalence 10,000 times for the three different scenarios using random draws of these coefficients.

To estimate the impact of climate change on disease prevalence (calculated as a percentage point change (p.p.)) across elevation, we subtracted the predicted prevalence for the climate change 2016 and RCP4.5 2056–60 scenarios from the counterfactual 10,000 times. To capture the nonlinear effect of climate (i.e., both increases and decreases in prevalence across elevation), we estimated the percentage point change in prevalence across elevation terciles. From the 10,000 iterations, we extracted the mean and 95% C.I., which captured both the uncertainty of the FE panel model and the RCP4.5 2056-60 predictions. We also estimated the effect of observed climate change (climate change 2016) on blister rust prevalence and the corresponding range shift using the estimated parameters from a FE panel model that included a cubic term. We found the interpretation of our observed climate change results was consistent between models but the magnitude of the climate change effect varied (Supplementary Note 5, Supplementary Fig. 2). Finally, to propagate uncertainty in the future climate change predictions, we randomly sampled VPD from the 20 CMIP5 models provided by MACA for each iteration of the MC simulation. Future predictions used the changes in host size and tree density from the past 20 years as a proxy for demographic changes in the future, and therefore should be interpreted with caution.

We estimated the mean and 95% C.I. of the range expansion at high elevations and contraction at low elevations by estimating the max/min elevation of the counterfactual, observed, and future climate change scenarios 10,000 times. Specifically, for each scenario, we calculated the range (min/max elevation) of plots with predicted probabilities of prevalence >2% (or the equivalent of ~1 infected tree/plot to reduce the likelihood of capturing random disease occurrences that are unlikely to effectively reproduce and spread[101,102]). Extracting the area (km$^2$) for each 1 meter elevation band (from a raster that clipped a digital elevation model (DEM) to the SEKI white pine range in QGIS[103]), we calculated the area between the predicted max/min elevation of the counterfactual and both the observed and future climate change scenarios. We used the 10,000 differenced area values to calculate the mean and 95% C.I. of the estimated range expansion at high elevation and range contraction at low elevation for the observed and future climate change scenarios.

*Mortality and Ribes spp. GLMMs.* To test whether infected white pines were more likely to die at higher levels of VPD, we fitted a mixed effects logistic regression of mortality (1 = dead, 0 = alive) for all infected white pine species at the time of the first survey, using R package lme4[104]. To control for biotic and abiotic factors known to affect mortality in our system, we included slope, aspect, white pine basal area, and tree size as explanatory variables[105]; we also included plot and species as crossed random effects (Supplementary Table 9). We then estimated the exact same model described above, but replaced VPD with elevation to show that predicted mortality also shifted with elevation (Supplementary Table 9). To visualize the changes in predicted probabilities of mortality across elevation, we estimated the marginal effect of elevation while holding non-focal variables constant (Fig. 6b).

To test whether *Ribes* spp. occurrence shifted across the VPD-elevation gradient, we developed two logistic regression GLMMs estimating plot-level *Ribes* spp. (Supplementary Table 8). The first model predicted presence/absence (1 = present, 0 = absent) of *Ribes* spp. with mean VPD between the first and second survey, slope, and aspect as independent variables. The second model used exactly the same independent and explanatory variables except replaced VPD with elevation. No random effects were specified in this model because plot equaled the number of observations (Supplementary Table 8). Elevation was excluded in the first model because it led to unreliable coefficient estimates.

## Testing for drought–disease interactions
*Sampling design.* To test the role of drought on infected host physiology, sugar pine trees were sampled in summer 2017 and resurveyed in September 2018 to estimate

mortality. Between June and September 2017, we identified blister rust infected sugar pines within SEKI (between 36°33′10″N - 36°38′05″N and 118°45′42″W - 118°49′15″W). All infected trees had bole cankers and at least one branch canker, indicating severe infections[48]. To control for the potential variation in microclimate and soil moisture that could affect tree physiological processes in response to drought and blister rust, we implemented a case-control sampling design. We collected 266 needle samples from 36 (18 uninfected and 18 infected) live trees with similar DBH (mean and standard deviation difference among pairs: 4.77 ± 4.73 cm). To ensure sampling across the potential range of physiological responses, trees were selected across a range of maturity, from saplings to mature individuals; the DBH range across all sampled trees was 6.3–70 cm. Paired asymptomatic control trees were selected within 40 m of diseased trees with consistent slope, aspect, and distance to streams.

For each tree, we sampled needles produced in each year between 2012–2017 (Supplementary Note 7) from south-facing, sunlit branches. Mean VPD values for the study period (2012–2017) were 17.67 (hPa) with a range of 16.82–18.79 (hPa). We returned to the same trees in September 2018 to assess tree health (live or dead). This study is unavoidably survivor biased because we only selected live trees (i.e., trees that had recently survived the extreme 2012–2015 drought). Consequently, our results may underestimate the true mean drought effect on sugar pine physiology and mortality.

While recent research and a report on high-elevation whitebark pine demonstrated that both physiological stress[106] and mortality[71] decreased at higher elevations in SEKI during the extreme drought, 2012–2015, we verified this result. Specifically, to test whether needle expansion during and following the drought varied between low and high-elevation regions (suggesting that drought did not cause high physiological stress at high elevations), we measured 225 needles on 30 healthy whitebark pine following protocols outlined above (Supplementary Note 7). Three areas near long-term monitoring plots used in this study were selected to obtain a range of high-elevation climatic conditions (plot 1: 37°12′52.49″N, 118°47′23.77″W; plot 2: 36°35′13 ″N, 118°39′60″W; and plot 3: 36°46′20″N 118°24′80″W). Only healthy, uninfected trees were sampled because blister rust infections occurred at very low density in whitebark pine (~1%) and foxtail pine (~0%), and it was not feasible to exactly replicate the sampling design we conducted in sugar pine.

*Stable isotope analyses.* To determine the variation in stable carbon and nitrogen isotope compositions of the sugar pine needles between 2012–2017, 5–10 fascicles were pulverized to a fine powder, weighed (8 mg), and encapsulated in tin. Samples were analyzed for both carbon ($\delta^{13}C$ and % C) and nitrogen ($\delta^{15}N$ and % N). Stable isotope ratios were determined for dried needle material at the University of California, Berkeley Center for Stable Isotope Ecology. Isotope ratios were estimated via continuous flow dual isotope analysis using a CHNOS Elemental Analyzer interfaced with an IsoPrime100 mass spectrometer. Long-term external precision for C and N isotope determinations at this facility was ± 0.10 ‰ and ± 0.20 ‰, respectively.

Isotopic values were reported in a standard notation as delta:

$$\delta = \frac{R_{\text{sample}} - R_{\text{standard}}}{R_{\text{standard}}} \qquad (4)$$

where $R$ represents the ratio of the heavy to light isotope in the sample and in the standard. The $\delta^{13}C$ results were reported in values relative to the Vienna Pee Dee Belemnite standard, based on the anomalously high $^{13}C{:}^{12}C$ ratio (0.01118) of the Cretaceous marine fossil, *Belemnitella americana*, from the Peedee Formation in South Carolina; the fossil's ratio is the zero standard for $\delta^{13}C$. The $\delta^{15}N$ measure of the ratio of the two stable isotopes of nitrogen, $^{15}N{:}^{14}N$ and the standard is atmospheric $N_2$ (0.3663 atom% ~ $^{15}N$).

Based on the delta values above, needle carbon isotope discrimination ($\Delta$) was calculated:

$$\Delta = \frac{\delta_a - \delta_p}{1 + \delta_p} \qquad (5)$$

where $\delta_a$ is the $\delta^{13}C$ of source $CO_2$ (ca. −8.0‰), and $\delta_p$ is the $\delta^{13}C$ of the foliar samples. Time-integrated intercellular $CO_2$ concentrations ($C_i$) of foliar samples were obtained by the following equation[107]:

$$C_i = \frac{c_a(\delta_a - \delta_p - a)}{b - a} \qquad (6)$$

where $c_a$ is atmospheric $CO_2$ concentration (380 ppm) (annual $CO_2$ concentrations were corrected for rising atmospheric $CO_2$ using La Jolla Pier records, available: https://scrippsco2.ucsd.edu/data/atmospheric_co2/ljo.html), $a$ is the fractionation caused by diffusion in air (4.4‰) and $b$ is the net fractionation caused by carboxylation (27‰).

*Statistical analyses.* To test blister rust effects on sugar pine physiological processes during and two years following the extreme drought in California, we developed six repeated measures analysis of variance models. Specifically, we tested whether infected trees had (1) higher $\delta^{13}C$, (2) lower $\delta^{15}N$, (3) lower % C, (4) lower % N, (5) shorter needle length, (6) and fewer fascicles than uninfected trees (Supplementary Note 3). We included error terms for paired trees nested within years to

account for the non-independence (i.e., repeat measures) in our sampling design. We confirmed homogeneity of variance and normality of residuals. Finally, to test whether blister rust infections increased the probability of mortality, we fitted a logistic regression model with blister rust and DBH (standardized with a mean of zero and a standard deviation of one) as independent variables and paired trees as random effects (Supplementary Fig. 6). To test for differences in the physiological traits between tree pairs with mortality (ten trees, 57 observations), we used the non-parametric Wilcoxon signed rank test to compare $\delta^{13}C$, $\delta^{15}N$, and the number of fascicles (Supplementary Fig. 6).

**Reporting summary.** Further information on research design is available in the Nature Research Reporting Summary linked to this article.

## Data availability

All field data for this study can be found via the Open Science Framework at https://doi.org/10.17605/OSF.IO/PC9FM[108], https://github.com/WildEcology/DudneyNatCommSEKI or upon request jdudney@berkeley.edu. PRISM downscaled historic climate data can be found at https://prism.oregonstate.edu/ and MACA downscaled forecasted climate data can be found at http://www.climatologylab.org/maca.html.

## Code availability

All analyses and data carpentry were performed using R, with the exception of the area estimate across SEKI and the maps from Figs. 2a and 4e, which used QGIS. The code used to generate the results from this study are available through GitHub https://github.com/WildEcology/DudneyNatCommSEKI, which is also mirrored on the Open Science Framework at https://doi.org/10.17605/OSF.IO/PC9FM.

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

## Acknowledgements

Funding for this project was provided by the US Forest Service, Forest Health Monitoring program (IAA 15IA11052021200), the National Park Service Natural Resource Challenge and the Service-wide Inventory and Monitoring Program. Dudney acknowledges support from the National Science Foundation, the Wilderness Society, the Robert and Switzer Foundation, the Garden Club of America, the David H. Smith Conservation Research Fellowship, and the Lewis and Clark Research Scholarship. We thank Dr. Robert Heilmayr for FE panel modeling support and Dr. Allison Simler-Williamson for a friendly review. We also thank the Rizzo and Latimer labs at UC Davis for their feedback, as well as our crews and crew leads for the data collection. This research was conducted on unceded land of many Native American groups, including the Mono (Monache), Tübatulabal, and Owens Valley Paiute. Latimer acknowledges support from USDA Hatch project CA-D-PLS-2017-H. Research permits included SEKI-2017-SCI-0029 and SEKI-2015-2017-SCI-0045. Any use of trade, firm, or product names is for descriptive purposes only and does not imply endorsement by the U.S. Government.

## Author contributions

Author contributions are defined using the Contributor Roles Taxonomy (CRediT; https://casrai.org/credit/). Conceptualization: J.D.; drought study conceptualization: C.E.W. and J.D.; data curation: J.D., C.E.W., and J.C.B.N.; formal analysis: J.D. and A.M.L.; funding: J.D., J.C.B.N., J.J.B., and A.J.D.; drought study investigation: C.E.W. and J.D.; disease survey: J.D., J.C.B.N., J.J.B., and A.J.D.; methodology: J.D., J.C.B.N., J.J.B., and A.J.D.; project administration: J.D., J.C.B.N., J.J.B., and A.J.D.; supervision: J.D., J.C.B.N., A.J.D., and C.E.W.; visualization: J.D.; writing—original draft: J.D.; writing drought study—original draft: C.E.W. and J.D.; writing—review and editing: J.D., L.A.M., J.J.B., A.J.D., C.E.W., and J.C.B.N.

## Competing interests

The authors declare no competing interests.
