## [Peer Review File · Nature Communications]

REVIEWER COMMENTS

Reviewer #1 (Remarks to the Author):

The authors used data from two surveys, separated by ~20 years, to argue that the climate-induced range shift is asymmetric. The research topic is interesting. However, I am not convinced that the authors have provided sufficient evidence to support their argument. I first explain several concerns about their statistical analyses and interpretation. Then, I list some editorial comments, with the aim to help improve the readability of the paper.

Concerns on statistical analyses and interpretation:

1. Using VPD-square term in GLMM essentially assumes that the VPD effect is symmetrical hump-shaped. I am not sure this assumption is ecologically justified. Also, this assumption seems to contradict with the idea of asymmetrical range shift. I suggest the authors should use Generalized Additive Mixed-effect model (GAMM) that does not assume specific shape of response curve and can accommodate random effect. In GAMM, interactive effects can be included as tensor product.
2. The authors used spatial data; thus, data points closer in space are not independent. That is, the effective degree of freedom is much smaller than the sample size as indicated in analysis. Spatial autocorrelation caused biased results. The authors need to explicitly accommodate spatial autocorrelation in their statistical analyses.
3. The main conclusion of the manuscript is drawn from two surveys. It is difficult to argue the difference is due to climate or simple due to particular environmental conditions around different sampling years. Time series data are needed to justify their arguments.
4. The analyses throughout the work are essentially additive model. These analyses cannot accommodate nonlinear context dependent effects of climate and hosts. See Deyle, E. R., M. C. Maher, R. D. Hernandez, S. Basu, and G. Sugihara. 2016. Global environmental drivers of influenza. *Proceedings of the National Academy of Sciences, USA* 113:13081-13086.
Deyle, E. R., R. M. May, S. B. Munch, and G. Sugihara. 2016. Tracking and forecasting ecosystem interactions in real time. *Proceedings of the Royal Society of London B: Biological Sciences* 283:20152258.
5. The analyses and projections in Figure 4 did not consider error propagation in their predicted curves. That is, the numbers shown in Figure 4 actually contain very high uncertainty. Thus, the results are not so impressive as the authors claimed.
6. Line 297-341: As the manuscript is written in its current form, I cannot understand how these additional analyses help explain the asymmetric shift. The integration of these additional analyses with the other survey data analyses needs more careful thought and writing.
7. The FE model assume that effects of some observed variables (i.e., plot, elevation, slope, white pine host and Ribes species) are time invariant. It is unclear to me how the authors decided which variables are time-variant and which variables are invariant. Also, this assumption seems to contradict the authors' own argument that many variables are nonlinear interactive.
8. Figure 3: In panel e, majority of 95%CI of delta-prevalence substantially covers 0 for the purple and orange lines, suggesting that delta-VPD did not cause significant change in delta-prevalence between surveys in these condition. These results contradict authors' argument.

9. Figure 4: This way of prediction essentially assumes that the prevalence depends only on VPD. This contradicts the author's argument that many other factors contribute to the asymmetric shift.

Editorial comments:

The writing throughout the whole manuscript is extremely difficult to follow.

I listed some below. There are too many difficult sentences; I cannot possibly detail all of them.

Abstract:

1. Abstract needs substantial re-writing. The current writing focuses too much on introduction and implications, but lacks of sufficient explanation of results.
2. Unclear to me what is "nonlinear range shift".
3. Line 40: "but also contributed to disease declines....." How much decline?
4. Line 40: "The disease shift was asymmetric". The logic flow is unclear in this sentence. Readers cannot catch why asymmetric disease shift can contribute decline in mean disease prevalence. I can't see the clear causal link by reading the sentence.

Main:

1. Line 51-58: Change in prevalence and shift in distribution do not necessarily have causal relationship; thus, I cannot understand the argument in this paragraph. The sentences sound that it must be one way or the other. However, ecologically, both can occur. I suggest that authors need to reframe their Introduction.
2. Line 56: "Others counter". I cannot understand this sentence. Observed VS predicted VS no change? I do not understand what the authors intend to say.
3. Line 68: Change first "is" to "in".

Methods:

1. Please elaborate what is "plot-level" data in SI Figure 1 and in the main text. Throughout the manuscript, I have no idea what is "plot-level" data and how this is different from tree data.

Figure legends:

1. Figure 4, Line 667: "Displaying". I can't tell what this sentence is referring to in the panel b. Where is CI?

Supplement

1. SI Figure 2: After reading the legends, I am confused whether the analyses were done using GLMM or GAM.

Reviewer #2 (Remarks to the Author):

This is one of the most interesting manuscripts I have reviewed in many years. The work significantly advances understanding of climate change impacts on forest pathogens by showing with a convincing suite of analyses that the occurrence of white pine blister rust (WPBR) rust has shifted asymmetrically in concert with increasing vapor pressure deficit over the last 20 years in the central Sierra Nevada in California. WPBR occurrence decreased at lower elevations, due to high VPD and high host mortality, whereas its occurrence increased at high elevations as the climatic conditions for the pathogen moved upslope into more naïve hosts. The work uses a novel combination of empirical, modelling, statistical and physiological approaches to make a very strong case for the asymmetrical shift in WPBR. The model system is well suited for detection of climate-change impacts on climate-host-pathogen

interactions. The manuscript writing is clear and engaging. I have only a few comments and suggestions below.

1. L53: Consider using a less value-laden term than "devastated" here to describe impacts of Dutch elm disease and chestnut blight. The forests where these trees used to occur are still very much intact, but they are different. Consider using "altered" or "restructured" instead of "devastated."
2. L109: Revise with a clearer subject than "it"
3. L112: The term "tree physiology" here is not completely accurate. You investigated drought impacts on tree physiological processes. Or, you could use the term tree physiological stress.
4. L118: I understand you are being cautious here by using "likely." I recommend that you err on the side of boldness and delete "likely." Your results are convincing to me.
5. L233: Here you cite Figure 5 before citing Figure 4. Figures are usually cited in order.
6. L316-328: This section could be clearer on the defense mechanisms of white pines against blister rust. What are the defenses, other than stomatal closure? Can you provide a brief example and citation of white pine defenses that are linked with carbon acquisition?

Thomas Kolb

Reviewer #3 (Remarks to the Author):

This manuscript analyses changes in white pine blister rust infections between two surveys 20 year apart. While the data and basic analyses appear robust, there are several areas which must be addressed before publication. The two most important relate to the motivation and ecological theory behind the manuscript, and the interpretation of the results.

The motivation behind the manuscript is not compelling. The authors establish a 'polarized debate' (Abstract) between those expecting range expansions, and those expecting range shifts, under climate change. I am not compelled by this argument. Fundamentally, either case is possible depending on any changes in the projection of the realized niche into geographic space under climate change. The authors cite one of Jorge Soberón's extremely insightful BAM model for species distributions, and I strongly recommend that they also read his other works on this model (Soberón, 2007; Soberón & Nakamura, 2009). In the interests of openness, I have advocated for the application of this model in understanding changes in the distributions of plant pathogens (Bebber, 2015) and believe it would be helpful here. B, the region in geographic space corresponding to the biotic niche, in this case is determined by the distributions of susceptible hosts (itself determined by climate). A, the geographic area corresponding to the abiotic (climate) niche, is the one affected directly by climate change. M is the geographic region reachable by movement, in this case we expect this to be unrestricted due to aerial spore dispersal. Explicitly plotting these geographic regions on a map (or at least A and B) and showing the overlap under past, current and future conditions, would be a very strong contribution to the literature. A map, and perhaps a schematic illustrating a mountain, would really help readers to grasp what is happening in this system. The schematic in Fig 1 goes some way toward this, but I didn't find it intuitive to understand. I found the Introduction rather confusing and at odds with ecological theory and the discourse on climate change effects and species distribution modelling. Phenomena like "multidirectional climate change impacts on disease prevalence" (L 70) and "asymmetrical range shifts" (L.) become less important when we consider that species distributions are determined by suitable areas in geographic space. Different types of shifts can be expected at different geographic scales, different topologies, and different climate responses of hosts and pathogens. At the global scale, warming is happening faster at higher latitudes, so we can expect

faster range expansions at higher latitudes than range contractions at lower latitudes. For plants, day length is a limiting factor so this will moderate range expansion in comparison with any pathogens. A nice example of potential mismatches between range shifts by a pest and host is given by (Berzitis et al., 2014). At the local scale, for example a mountain, asymmetric shifts are expected due to the shape of mountains. Because the land area shrinks with altitude, range expansions at high altitude will necessarily be smaller than range contractions at low altitude, other things being equal. The BAM model clarifies and simplifies all these considerations.

My second major concern relates to the way that disease incidence/risk is modelled in relation to climate. The authors find a significant relationship with VPD, which they correctly identify is a function of temperature and humidity. The problem is, that this statistical relationship hides the mechanistic relationship between temperature, moisture, and disease risk. Rust fungi in particular tend to require liquid water on the leaf surface, or at least high humidity, to germinate and infect. I've developed some models for this, but to avoid over-citing my own work I found the model by Robert Magarey and colleagues to be very helpful (Magarey et al., 2005). The authors mentions "hump" shaped relationships of infection rates to temperature (temperate performance curves), and may wish to read our recent synthesis of these for fungal plant pathogens (Chaloner et al., 2020). I have just checked our database and we have cardinal temperature estimates for *C. ribicola*, which the authors might like to discuss, or even implement in a mechanistic model (Bebber et al., 2020). A statistical model based on VPD hides the biological mechanism behind infection risk and doesn't help us understand how climate is driving range shifts. The non-linear function of temperature and humidity that gives VPD means that the same VPD can occur at 32C, 67% RH and 21C, 38% RH. Confounding temperature and moisture by using VPD as a predictor makes interpretation of the result difficult. While statistical species distribution models remain popular and can often give a reliable result, they suffer from several issues including extrapolation outside the training range. This is particularly problematic for climate change projections. A forceful critique of these methods is given by Sutherst (2014).

Specific points (with line numbers)

33, 55. I don't agree that there's a significant debate. Ranges will change in various ways depending on the changes in the BAM regions. As Altizer et al. (2013) write "Climate change will continue to limit the transmission of some pathogens and create opportunities for others". The debate they discuss is more around the different results from statistical and mechanistic models. I disagree with the way that Lafferty (2009) is cited. Much of his paper discusses the differences between statistical and mechanistic models. However, he does reinforce the importance of the BAM framework "...while a reduction in habitat suitability should reduce a species' range, an increase in habitat suitability does not necessarily result in an increase in geographic distribution. This is because other factors besides climate, such as barriers to dispersal, competition, and predation, affect the realized niche."

68. Not clear what relationship this is referring to. Rather than "multidirectional" do you mean a nonlinear relationship with temperature?

83. Again, asymmetrical range shifts are to be expected in all be rare cases of perfectly flat topography with no confounding biological relationships etc. Asymmetry vs. symmetry is not really the important question here.

88. There are new citations for drought projections, e.g. (Cook et al., 2018).

95. What is a multidirectional impact of drought?

118. The paper doesn't discuss the thermal response functions of WPBR.

Fig. 1. A figure which more explicitly shows the geographical areas of suitable hosts and infection risk would be more helpful. Perhaps adapt the map from Supplementary Figures.

150. Not sure if this site is unique in this regard.

174. It would be interesting to use a beta function or similar to map risk, using these temperature tolerances. This would give a semi-mechanistic model to compare with observations.

195. The "hump-shape" can be modelled with a beta function (or others).

203. Again, I'm uncomfortable with the use of VPD as it combines two variables, temperature which determines growth rate, and moisture which determines ability to grow.

221. What does 'multidirectional' mean in this context?

241. The linear change in elevation is not a range expansion. Ranges are areas, so the range expansion should really be measured in hectares in this instance. Because mountains are pointed, the actual range increase will likely be smaller.

263. The manuscript is lacking a good description of the WPBR life cycle, and how climate affects different processes within the cycle. At minimum, we should know the role of moisture and temperature in the infection process.

278. Again, it would be great to see maps of host mortality and physiological stress, and how these have changed/might change in future.

299. Interesting result.

316. Interesting result. Droughts do different things to host susceptibility, depending on the pest/pathogen. For insects, leaf chewers are negatively affected because of reduced production of fresh leaves. Stem borers benefit because of reduced defensive sapflow (in conifers).

346. Again, I don't think 'multidirectional' and 'nonlinear' are used in the right way here. The focus of the discussion should really be on a) what aspect of climate change drove the shift in range (avoiding discussion of VPD, as this is not directly a driver), and b) how biotic interactions with the host affected the shift. (Berzitis et al., 2014) is a good model for this. WRT malaria, remarkably a non-linear temperature response was only included in models in 2013 (Mordecai et al., 2013) - before this models assumed that transmission would simply keep increasing with temperature!

393. The map is really an important and should be a main figure.

400. 4km is still coarse resolution, particularly given the steep topography. How did you account for altitudinal changes in temperature within pixels? New tools are available for this (Bütikofer et al., 2020).

437. The GLMMs appear very thorough but did you control for spatial autocorrelation among plots? I would expect strong spatial autocorrelation because of aerial spore dispersal.

458. These types of issues are avoided when using a mechanistic model of the biological process.

Fig. 2. Why is only Tmin given?

Fig. 3. I don't think graphs of coefficient estimates are helpful. Better in text or tabulated.

Fig. 4. Again, I'd like to see how changes in % infected are driven by the disease risk itself, and changes in host range (and alternate host) and susceptibility.

Fig. 5. Panel a clearly shows that the change in prevalence is at low altitudes, not high. That is useful. In all panels, data points should be shown where possible (rather than just statistical summaries).

References

- Altizer, S., Ostfeld, R.S., Johnson, P.T.J., Kutz, S. & Harvell, C.D. (2013) Climate Change and Infectious Diseases: From Evidence to a Predictive Framework. *Science*, 341, 514–519.
- Bebber, D.P. (2015) Range-Expanding Pests and Pathogens in a Warming World. *Annual Review of Phytopathology*, 53, 335–356.
- Bebber, D.P., Chaloner, T.M. & Gurr, S.J. (2020) Fungal and Oomycete cardinal temperatures (the Togashi dataset). 2424892 bytes.
- Berzitis, E.A., Minigan, J.N., Hallett, R.H. & Newman, J.A. (2014) Climate and host plant availability impact the future distribution of the bean leaf beetle (*Cerotoma trifurcata*). *Global Change Biology*, 20, 2778–2792.
- Bütikofer, L., Anderson, K., Bebber, D.P., Bennie, J.J., Early, R.I. & Maclean, I.M.D. (2020) The problem of scale in predicting biological responses to climate. *Global Change Biology*, in press.
- Chaloner, T.M., Gurr, S.J. & Bebber, D.P. (2020) Geometry and evolution of the ecological niche in plant-associated microbes. *Nature Communications*, 11, 2955.
- Cook, B.I., Mankin, J.S. & Anchukaitis, K.J. (2018) Climate Change and Drought: From Past to Future.

Current Climate Change Reports, 4, 164–179.

Lafferty, K.D. (2009) The ecology of climate change and infectious diseases. *Ecology*, 90, 888–900.

Magarey, R.D., Sutton, T.B. & Thayer, C.L. (2005) A Simple Generic Infection Model for Foliar Fungal Plant Pathogens. *Phytopathology*, 95, 92–100.

Mordecai, E.A., Paaijmans, K.P., Johnson, L.R., Balzer, C., Ben-Horin, T., Moor, E. de, McNally, A., Pawar, S., Ryan, S.J., Smith, T.C. & Lafferty, K.D. (2013) Optimal temperature for malaria transmission is dramatically lower than previously predicted. *Ecology Letters*, 16, 22–30.

Soberón, J. (2007) Grinnellian and Eltonian niches and geographic distributions of species. *Ecology Letters*, 10, 1115–1123.

Soberón, J. & Nakamura, M. (2009) Niches and distributional areas: concepts, methods, and assumptions. *Proceedings of the National Academy of Sciences*, 106, 19644–19650.

Sutherst, R.W. (2014) Pest species distribution modelling: origins and lessons from history. *Biological Invasions*, 16, 239–256.

Signed: Dan Bebber

Nonlinear climate change impacts on infectious tree disease

Manuscript originally submitted to *Nature Communications*, 8/10/2020

Review response submitted, 1/17/2020

REVIEWER 1

Broad Comments

The authors used data from two surveys, separated by ~20 years, to argue that the climate-induced range shift is asymmetric. The research topic is interesting. However, I am not convinced that the authors have provided sufficient evidence to support their argument. I first explain several concerns about their statistical analyses and interpretation. Then, I list some editorial comments, with the aim to help improve the readability of the paper.

Response: Thank you for these comments and suggestions. They have greatly increased the clarity and rigor of our results and descriptions.

Concerns on statistical analyses and interpretation:

1. Using VPD-square term in GLMM essentially assumes that the VPD effect is symmetrical hump-shaped. I am not sure this assumption is ecologically justified. Also, this assumption seems to contradict with the idea of asymmetrical range shift. I suggest the authors should use Generalized Additive Mixed-effect model (GAMM) that does not assume specific shape of response curve and can accommodate random effect. In GAMM, interactive effects can be included as tensor product.

Response:

Our data span a very large climate gradient and are therefore ideal for isolating nonlinear effects—see Kreyling et al. (2018) where they clearly demonstrate that nonlinear effects are best captured across strong gradients (and replication often adds noise). While we agree that estimating a GAMM model could better characterize the shape of the VPD-blister rust response function, our sample size is likely not high enough to precisely estimate a more flexible functional form like a nonparametric spline. Additionally, our aim in this paper was not to demonstrate that the relationship between VPD and prevalence is asymmetric—rather that once the nonlinear VPD-blister rust relationship is overlaid onto multiple other abiotic-biotic drivers of disease, the actual effect of climate change is an asymmetric shift in prevalence (greater declines at low elevations than increases at high elevations). Based on this comment, your comments (**linked**) **6,9**, and Reviewer 3's comments **5,9**, we realized that our focus on asymmetry may have been confusing. We have fully rewritten the paper to change the emphasis and focus instead on the significant nonlinear relationship between blister rust and VPD that contributed both to decreases and increases in disease prevalence across elevation.

2. The authors used spatial data; thus, data points closer in space are not independent. That is, the effective degree of freedom is much smaller than the sample size as indicated in analysis. Spatial autocorrelation caused biased results. The authors need to explicitly accommodate spatial autocorrelation in their statistical analyses.

Response: This is a critical point that we sought to address in the previous version of our paper; however, this comment and R3's comment **27** highlights that these tests were not clearly presented and explained. We have now clarified our approach and presented additional tests here to control for spatial autocorrelation.

To control for spatial autocorrelation in observational studies of organisms, the following four-step approach is generally recommended in ecology²: 1) consider the potential causes of spatial autocorrelation and the spatial scales they may be acting on, 2) conduct tests for spatial autocorrelation in raw data, 3) if there is spatial autocorrelation in the raw data, use covariates that may be causing the spatial autocorrelation and/or test to see if using a hierarchical GLMM with a random term, like site/plot, accounts for spatial dependencies, and 4) if spatial autocorrelation is unable to be controlled for by model covariates, conduct additional analyses that can account for the spatial configuration of the data. We conducted steps 1-3 and found that using our covariates, including plot as a random effect, controls for positive spatial autocorrelation, the values largely of concern². However, to further verify these results, we added two more robustness tests that confirmed our results are unlikely to be biased by spatial autocorrelation.

Step 1) The most common cause of spatial autocorrelation in species distribution data is dispersal³. While other environmental factors can also exhibit specific autocorrelative patterns (e.g., topography, climate), we can more easily control for these sources simply by including these variables in our GLMMs. In our system, the dispersal of blister rust from *Ribes* to pine hosts and vice versa could cause spatial autocorrelation. Our study area (~100 km x 70 km), is much smaller than the estimated range that aeciospores can travel from white pines to infect *Ribes* (~1,200 km as reported by Geils, Hummer, and Hunt 2010). Thus, given the spatial scale of our prevalence data, dispersal by aeciospores was unlikely a major cause of spatial autocorrelation. In contrast, estimated dispersal from *Ribes* to white pines is much shorter, a few kilometers at most⁴; thus dispersal from *Ribes* may cause positive autocorrelation in the raw data at shorter scales, likely less than ~3km.

Step 2-3) Following methods outlined by Zuur et al. (2009), we estimated a spline correlogram with 95% pointwise bootstrapped CI (grey) and Pearson residuals autocorrelation (Fig 1). We see that indeed, in the raw data of blister rust prevalence from the first survey, there are some signs of positive spatial autocorrelation at shorter distances, but the confidence intervals are crossing 0 (Fig 1).

Fig 1. Spline correlogram and 95% CI using blister rust prevalence from the first survey and corresponding Distance measured in UTM's ($0.2^\circ = 22\text{km}$). The distance at which *Ribes* is likely acting on the system is less than $\sim 3\text{km}$, or less than 0.02° . This is where we see a little bit more positive spatial autocorrelation than estimated at longer distances, but again, the 95% CIs cross 0.

To test whether adding in *Ribes* occurrence helps control for this positive spatial autocorrelation, which we expect to be the likely cause, we see that indeed, the spatial autocorrelation is effectively controlled for by including *Ribes* as a covariate (Fig 2).

Fig 2. Spline correlogram and 95% CI including *Ribes* only as a covariate. Distance measured in UTM's ($0.2^\circ = 22\text{km}$).

When we include all covariates and plot as a random effect (Fig 3), which we used in our GLMMs estimating infections, we see that that plot does perhaps control for a little more spatial autocorrelation. Neither of these correlograms identify significant positive autocorrelation, however.

Fig 3. Spline correlogram and 95% CI including all covariates and plot random effects.

Additionally, we find that the neither the raw data (Fig. 4 left), the models just with *Ribes*, nor all variables (Fig 4 right) exhibit significant positive spatial autocorrelation using the second survey prevalence (Fig. 4); Right figure with all covariates has a mean correlation of 0.00096 with a 95% CI of (-0.276, 0.265). This suggests that our analyses estimating disease prevalence are not strongly biased by spatial autocorrelation.

Fig 4. Left, spline correlogram and 95% CI of raw second survey data. Right, spline correlogram and 95% CI including all covariates.

Step 4) In an effort to verify these results, we added two more robustness checks:

Test A. Testing for possible confounding by dispersal dynamics in our blister rust-VPD relationship. On lines 513-523, following methods also described by Baker-Austin et al. (2013), we described how we tested whether blister rust dispersal had confounded the VPD coefficient estimates. Specifically, we fitted a GLMM with an interaction between time period and the linear and quadratic VPD terms. If coefficient estimates of the interactions between VPD and time were significant, this would suggest that the relationship between prevalence and VPD had changed through time. A change in the relationship between VPD and blister rust would suggest that other factors, like dispersal, may be driving changes in prevalence. If the interactions were insignificant, however, this would suggest that blister rust dispersal was unlikely to be confounding. Most importantly, a similar VPD-blister rust relationship between surveys would help confirm that blister rust would likely track changes in VPD – which was important to test

for given that we were estimating the effect of a change in VPD on the change in prevalence in our FE panel models. Our fitted models demonstrated that our coefficient estimates were unlikely to be biased by blister rust dispersal.

Blister rust model with time interactions			
Predictors	Coefficient	Std. Error	P-Value
Intercept	-5.19	1.11	< 0.001
VPD	2.53	0.45	< 0.001
Time	-0.17	0.30	0.574
VPD ²	-0.83	0.18	< 0.001
Ribes	1.16	0.38	0.003
Aspect	-0.48	0.37	0.198
Slope	-0.02	0.02	0.428
Density	-0.63	0.18	0.001
DBH	-0.31	0.07	< 0.001
VPD*Time	-0.52	0.44	0.2 ¹ 31
VPD ² *Time	0.23	0.17	0.169
Density*Time	1.03	0.20	< 0.001
DBH*Time	0.41	0.10	< 0.001
N _{plot}	147		
N _{species}	4		
Observations	12447		
Marg. R ² / Cond. R ²	0.350 / 0.77		

Supplementary Table 1: Logistic regression estimates, standard errors, and corresponding p-values for GLMM estimating infected trees with time (first and second survey) as interaction terms. The relationship of VPD and VPD² is not significantly different between the first and second surveys (interaction terms were not significant), suggesting that the relationship between infections and climate has remained relatively stable through time.

Test B. Estimating dispersal distance between new infections and nearest known historic infection. Using the first and second survey data, we also estimated the geographic distance between any new infected tree and a known infected tree in plot_i at t₀. Including distance to nearest infection at t₀ for each tree as a variable in the GLMM model estimating new infections at t₁, we find that including this variable into the *model does not improve model fit (AIC is slightly higher), the variable is not significant, and our estimates of the primary coefficients of interest are largely unchanged.*

Predictors	Second survey without dispersal variable			Second survey WITH dispersal variable		
	Coefficient	Std. Error	P-Value	Coefficient	Std. Error	P-Value
Intercept	-5.32	1.04	<0.001	-5.31	1.04	<0.001
VPD	1.93	0.51	<0.001	1.91	0.55	0.001
VPD ²	-0.56	0.18	0.002	-0.55	0.19	0.004
Density	-0.05	0.21	0.809	-0.05	0.21	0.807
Ribes	0.96	0.44	0.028	0.95	0.45	0.035
DBH	0.12	0.08	0.157	0.12	0.08	0.156
Slope	-0.00	0.02	0.852	-0.00	0.02	0.848
Aspect	-0.96	0.43	0.026	-0.96	0.43	0.025
Nearest infection				-0.04	0.32	0.911
N	147 _{plot}			147 _{plot}		
	4 _{species}			4 _{species}		
Observations	5416			5416		
AIC	1100.5			1102.5		

Table 1. Showing the difference between GLMM models estimating second survey infections that do not include (left side) and include (right side) an estimated distance to nearest infection. Note that the AIC without the dispersal variable on the left is slightly lower and the variable “Nearest infection” is not significant at $P = 0.911$.

Our GLMM results in Table 1 may be a result of many factors, including: 1) this dispersal distance variable has high measurement error (likely) and 2) dispersal dynamics were not well captured at the spatial scale of our study (likely true in the case of aeciospores).

Finally, GLMMs do not allow us to control for endogenous and exogenous unmeasured variables that may also be spatially autocorrelated. The fixed effects (FE) panel modelling approach, however, does allow us to control for all unmeasured variables that are spatially autocorrelated and time-invariant⁶. Also see comment 7 below.

We have revised the text on lines 508-523 to include and better describe these tests ensuring that our primary conclusions are robust to possible confounding due to dispersal or other potential causes of spatial autocorrelation.

3. The main conclusion of the manuscript is drawn from two surveys. It is difficult to argue the difference is due to climate or simple due to particular environmental conditions around different sampling years. Time series data are needed to justify their arguments.

Response: Thank you for this comment. Our primary conclusion reflects three observations of our timeseries data: 1) the climate-disease relationship is significant and nonlinear in the first survey period GLMM, 2) the climate-disease relationship is *consistently* significant and nonlinear in the second survey GLMM *and* the coefficient estimates do not significantly change through time (see response to your comment 1), and finally 3) this nonlinear relationship holds when looking at changes in prevalence across the two time periods in the FE panel model (see Fig. 3 in comment 9 below). Thus, our two-survey timeseries data provide multiple lines of evidence to support our hypotheses that VPD is strongly associated with blister rust prevalence and changes in disease prevalence.

Although our time series data only has two time periods, we believe that a greater density of time periods would not change our results based on the ecology of this pathosystem. Specifically, blister rust is slow moving and wave years (when climatic conditions are suitable) occur relatively infrequently, every 7-10 years⁷. Not only is the disease slow spreading but it 1) kills its hosts slowly (even some smaller DBH trees in our system with severe bole infections were still alive 20 years later), 2) new infections are not often visible for the first 2-5 years⁸ depending on the size of the tree, and 3) larger trees can recover from the disease eventually. Thus, given that the disease is slow spreading, slow killing, *and* our surveys captured the cumulative effect of disease at both time periods, we conclude that this timeseries is actually pretty good given the behavior of this disease. While we likely missed a few infections that killed trees before the second survey, it is highly unlikely that these missed infections would lead to major modifications in the estimated climate-disease relationship in our system, given that our data clearly demonstrate that this climate-disease relationship is very consistent over a longer time interval. In addition, including standing dead trees that showed symptoms of blister rust did not change the results of our study, further suggesting that the blister rust-climate relationship we have estimated is stable and highly robust.

4. The analyses throughout the work are essentially additive model. These analyses cannot accommodate nonlinear context dependent effects of climate and hosts. See Deyle, E. R., M. C. Maher, R. D. Hernandez, S. Basu, and G. Sugihara. 2016. Global environmental drivers of influenza. *Proceedings of the National Academy of Sciences, USA* 113:13081-13086.
Deyle, E. R., R. M. May, S. B. Munch, and G. Sugihara. 2016. Tracking and forecasting

ecosystem interactions in real time. *Proceedings of the Royal Society of London B: Biological Sciences* 283:20152258.

Response: This is true and thank you for highlighting these cutting-edge methods. We would be really interested to investigate further the context dependencies of climate, drought, and hosts on blister rust prevalence—this would be an excellent next step and we have included these ideas into discussion as follows, lines 416-420:

“These abiotic-biotic interactions may be nonlinear, potentially shifting in strength and direction as global warming proceeds^{75,76}. Investigating the context dependencies of plant-pathogen-drought interactions will be critical to estimate the magnitude of these interacting effects on plant pathogen range shifts.”

5. The analyses and projections in Figure 4 did not consider error propagation in their predicted curves. That is, the numbers shown in Figure 4 actually contain very high uncertainty. Thus, the results are not so impressive as the authors claimed.

Response: This an excellent point and as a result, it enabled us to greatly improve the rigor of our estimates of climate change impacts. Specifically, we redid this analysis to incorporate model uncertainty/error from our prevalence predictions, as well as the uncertainty of 20 CMIP5 experiments. Please see lines 576-616 for a full description of the methods. In brief, we explained the Monte Carlo simulation approach on lines 588-599:

“Specifically, we used the R function “predict” from the R stats package “rms”⁹ to obtain predicted prevalence values for the three scenarios. All predictions included the same demographic changes in DBH and tree density but varied VPD by the three scenarios, thereby isolating the effect of a change in VPD on blister rust prevalence. The uncertainty of the panel model predictions was captured using a Monte Carlo simulation. First, we randomly sampled 10,000 draws of the FE panel model coefficients using their estimated mean values and variance-covariance matrix. Then we predicted prevalence 10,000 times for the three different scenarios using random draws of these coefficients. To propagate uncertainty in the future climate change predictions, we randomly sampled VPD from the 20 CMIP5 models provided by MACA for each iteration of the MC simulation. Future predictions used the changes in host size and tree density from the past 20 years as a proxy for demographic changes in the future, and therefore should be interpreted with caution.”

Incorporating these sources of error into the predictions widened the CIs, as expected, but the direction of the effect of changing VPD on disease prevalence remained the same. All of our figures and statistical results using FE panel model predictions now show a mean and estimated 95% CI based on this Monte Carlo simulation approach.

6. Line 297-341: As the manuscript is written in its current form, I cannot understand how these additional analyses help explain the asymmetric shift. The integration of these additional analyses with the other survey data analyses needs more careful thought and writing.

Response: Thank you for pointing this out. First, we have revised the framing of this paper and are no longer using asymmetry as a main focus. Second, to better integrate the results of our *in situ* drought-disease study, we have revised the results sections. Not only have we greatly clarified the explanations of our climate change results (see lines 253-292), but we have also greatly improved the integration of additional modelling and drought impact results, including a description of two pathways leading to the suppression of infections and two physiological mechanisms that explain pathway 2 (see 293-392).

For example, we better integrated the additional analyses on lines 292-325:

“Drought and host interactions likely modified disease prevalence

Though climate change shifted the climate optimum of blister rust into higher elevations thereby increasing the number of vulnerable hosts to infection (Fig. 4e), mean observed prevalence declined between surveys. Specifically, increases in new infections at high elevations were more than offset by decreases at low elevations (Fig. 5a). In aggregate, this resulted in a 32.79% decrease in mean prevalence between surveys. Though many factors likely contributed to this decline, here we focus on the combination of host-pathogen interactions, a strong water availability gradient across elevation, and spatially varying drought impacts. We suggest that these abiotic-biotic interactions modified the disease distribution through two pathways: 1) fewer alternate hosts at high elevations dampened infection probabilities, even as warmer conditions ameliorated the historic climatic constraints on spread and 2) higher water deficit in arid regions decreased prevalence, an effect that was likely amplified by drought.

Pathway 1: Fewer alternate hosts at high elevations dampened infection probabilities

Host-pathogen interactions likely reduced infection probabilities at high elevations. GLMMs estimating the likelihood of blister rust infections, for instance, highlighted that *Ribes* is important for white pine infection in this system (Fig. 3b,d), as it is necessary for blister rust to complete its life cycle⁴⁵. *Ribes*, however, occurs less frequently in colder, higher elevation regions in SEKI (Fig. 5c, Fig. 6b, Supplementary Table 7). Because fragile basidiospores travel short distances from *Ribes* to infect white pines (meters to few kilometers)—which is a stark contrast to aeciospores that can infect *Ribes* up to 1,200 km away⁴⁵—the lower occurrence of *Ribes* at high elevations may have suppressed white pine infection rates. Consequently, even as climatic conditions became more suitable for blister rust at high elevations, the probability of infection remained lower (Supplementary Figure 3c,d). Infections at high elevations could increase in the future, however, if high-elevation *Ribes* populations densify and spread in response to climate change.

Pathway 2: Water deficit linked to declines in prevalence at low elevations

Spatially varying abiotic interactions with drought and water availability gradients also likely modified the blister rust distribution as it shifted in response to climate change. To disentangle these complex, interacting effects, we combined a GLMM of infected host mortality (using data from our observational field study) with an *in situ* test of physiological responses to drought in sugar pine hosts. Our results highlighted two important mechanisms that contributed to faster declines in disease prevalence in arid regions (low elevations) compared to mesic regions (high elevations) . . .”

7. The FE model assume that effects of some observed variables (i.e., plot, elevation, slope, white pine host and *Ribes* species) are time invariant. It is unclear to me how the authors decided which variables are time-variant and which variables are invariant. Also, this assumption seems to contradict the authors' own argument that many variables are nonlinear interactive.

Response: Time invariant variables (both observed and unobserved) are the variables which, over the time period we studied, have not changed within our plots. The observed time-invariant variables in this study included elevation, slope, plot, white pine species and *Ribes* occurrence. Specifically, all variables in the model with a value of 0 after they were demeaned were mathematically removed from the analysis. By subtracting the mean values of the dependent and independent variables between the two time periods, we effectively subtracted out the time-invariant observed variables (i.e., plot, elevation, slope, white pine host, and *Ribes* species), as well as the time-invariant unmeasured variables (a_i). We have updated the text to make this clearer (lines 557-562):

“Subtracting the mean value from each dependent and independent variable mathematically removes all time-invariant variables, including both observed (e.g., plot, elevation, slope, white pine host, and *Ribes* species), and the unobserved time-invariant variables (e.g., soil class, solar radiation) represented by a_i . In this way, the panel model allowed us to control for time-invariant but unobservable variables which might otherwise confound our analysis¹⁰ of the relationship between VPD and blister rust.”

8. Figure 3: In panel e, majority of 95%CI of delta-prevalence substantially covers 0 for the purple and orange lines, suggesting that delta-VPD did not cause significant change in delta-prevalence between surveys in these condition. These results contradict authors' argument.

Response: We agree with this interpretation. Panel 3e illustrates the effect of delta-VPD on delta-prevalence but divides the responses into three gradients (i.e., terciles) to more clearly show the nonlinear relationship. The purple and orange lines represent the cooler regions of the gradient (lower two VPD terciles). Increasing delta-VPD (i.e., warmer and drier) in these initially cool regions had only a modest positive impact on prevalence. So as noted by R1, the 95%CI's cover 0 for much of the delta-VPD gradient. However, in the initially warmer plots (upper VPD tercile, green line), increasing delta-VPD tracks steep decreases in disease prevalence. This sorting of responses by VPD tercile emphasizes a key point of the paper: the nonlinear and unequal influence of climate change on the range of the disease. At the cooler sites (i.e., at higher elevations), increasing VPD has only a minor impact on increasing prevalence likely due to host-pathogen interactions (e.g., lower prevalence of *Ribes* which need to occur within short distances (<3km) to white pines for infection to occur). In contrast, at the warmer sites (i.e., at lower elevations), increasing VPD has a major impact. This is likely a result of many factors, including higher rates of host mortality, the suppression of new infections likely due to higher rates of stomatal closure in response to higher aridity. Thus our results in Fig 3 are consistent with the results illustrated in Fig 4-5.

9. Figure 4: This way of prediction essentially assumes that the prevalence depends only on VPD. This contradicts the author's argument that many other factors contribute to the asymmetric shift.

Response: Though our model does focus on estimating the causal relationship between VPD and blister rust prevalence, it is carefully designed in a way that acknowledges that prevalence is simultaneously affected by many other factors. Indeed, the beauty of the fixed effects panel model is that it allows us to control for a diversity of time-invariant observable and unobservable variables, without explicitly estimating their individual effects (we do estimate the observed variable effects in our GLMMs, however, to complement our FE panel model results). The FE model gives us a more accurate estimate of the marginal effect of VPD that isn't confounded by these other important determinants of prevalence.

Furthermore, the FE panel model is not assuming that these time-invariant variables were not important. Rather, it assumes that their effects on prevalence were constant through time; since they are stable across time periods they would be unlikely to *cause a change* in disease prevalence. We see in our GLMMs (Fig 3 below), for example, that the time-invariant variables aspect (second survey) and *Ribes* (first survey and second survey) explained the probability of infection, but not a change in infection, since their effects were time-invariant. So, in summary, we are using the GLMM models to explain the probability of infection, including what variables were important for infection to occur and whether the relationship between VPD and blister rust was nonlinear. We are using the FE panel model to test how the variables that changed between surveys, including VPD, tree density and growth, affected the observed change in disease prevalence—which can be interpreted as a more causal relationship¹⁰.

Fig 3: Nonlinear relationship between blister rust and VPD. (a, c) Proportion of infected white pines (smoothed lines) and the number of tree stems across VPD (green bars) for the first (1995-1999) and second (2013-2016) surveys; shaded areas of smoothed lines represent 95% C.I. around local non-parametric regression (loess) estimates. (b, d) Coefficient estimates from logistic

regression models explaining tree-level blister infections from the first and second surveys with standard error bars. **e**, plot-level change in prevalence (% infected/total live stems) as a function of the change in VPD between surveys. Plots were divided into VPD terciles defined at t_2 , thereby demonstrating both an increase in blister rust prevalence in cold regions (low VPD) and a decrease in prevalence in more arid regions (higher VPD); showing shaded 95% C.I. **f**, Coefficient estimates of four explanatory variables from the FE panel model with standard error bars. ** $p < 0.01$, * $p < 0.05$.

Additionally, all time-varying variables were used to estimate the three climate change scenarios in Fig 4. That is, the estimate of the counterfactual and the two climate change scenarios included changes in DBH and tree density. This is potentially problematic for the future climate change scenario, however, as we are assuming the changes in DBH and tree density are similar to the future changes in these variables. We caveat our forecasts in the paper on lines 278-292.

“These projected changes in prevalence under RCP4.5 2056-60 do not account for potential changes in drought severity and host distributional shifts that may also occur in response to future climate change. Though host species’ recruitment rates vary spatially and by species¹¹, white pine demographic changes will likely lag behind blister rust expansion, particularly at high elevations where trees are very slow growing (can live for >1,000 years) and slow recruiting (~0.2 %/year)¹¹. Using the past ~20 years of demographic change as a control in our projection of the next ~30 years is likely a reasonable approach. In contrast, *Ribes* spp. often respond positively to fire¹². Predicted changes in fire activity¹³ may lead to greater *Ribes* spp. recruitment in addition to potential elevational shifts in response to climate change; we did not account for these potential shifts in our future climate change projections. Finally, climate change may shift the blister rust pathosystem farther from equilibrium with the environment, which could lead to greater variability in disease prevalence in the future. These combined effects, in addition to predicted changes in drought frequency and severity¹⁴ (see below), could further modify disease prevalence in the future. Consequently, our projections of climate change impacts on blister rust should be interpreted cautiously.

Editorial comments:

The writing throughout the whole manuscript is extremely difficult to follow. I listed some below. There are too many difficult sentences; I cannot possible detail all of them.

Response: Thank you for highlighting areas of this manuscript that are unclear. We have worked to clarify the writing throughout the manuscript.

Abstract:

10. Abstract needs substantial re-writing. The current writing focuses too much on introduction and implications, but lacks of sufficient explanation of results.

Response: Thank you—most of the reviewers are also in agreement with this statement and we have updated the abstract on lines 33-45.

“Infectious plant disease range shifts are expected under climate change. As pathogens move, emergent abiotic-biotic interactions are predicted to modify disease distributions, leading to unpredictable changes in disease risk. Evidence of these complex range shifts remains largely speculative, however. Combining a long-term study of the tree disease caused by *Cronartium ribicola* with a six-year field assessment of drought-disease interactions in the southern Sierra Nevada, we found that climate change moved the climate disease optimum into higher elevations. This shift likely decreased prevalence by 5.5 (4.4, 6.6) p.p. in arid regions and increased prevalence by 6.8 (5.8, 7.9) p.p. in colder regions. Though climate change increased the suitable area for blister rust by an estimated 367.7 (0, 514.2) km², the combination of spatially-varying host-pathogen interactions and drought impacts contributed to a 32.79% decrease in prevalence between surveys. Here, complex host-pathogen-drought interactions initially dampened disease risk, but the long-term impacts of range expansions remain concerning for naïve hosts.”

11. Unclear to me what is "nonlinear range shift".

Response: We took this out.

12. Line 40: “but also contributed to disease declines.....” How much decline?

Response: Thank you, we now state on lines 39-40:

“This shift likely decreased prevalence by 5.5 (4.4, 6.6) p.p. in arid regions and increased prevalence by 6.8 (5.8, 7.9) p.p. in colder regions.”

13. Line 40: “The disease shift was asymmetric”. The logic flow is unclear in this sentence. Readers cannot catch why asymmetric disease shift can contribute decline in mean disease prevalence. I can't see the clear causal link by reading the sentence.

Response: Thank you, we have taken out the asymmetric framing of the paper.

Main:

14. Line 51-58: Change in prevalence and shift in distribution do not necessary have causal relationship; thus, I cannot understand the argument in this paragraph. The sentences sound that it must be one way or the other. However, ecologically, both can occur. I suggest that authors need to reframe their Introduction.

Response: Thank you for pointing this out. We have completely re-written this section of our introduction in response to your concern and R3's concern, lines 52-61:

“Infectious plant diseases are reshaping ecosystems, disrupting global food supplies, and threatening human health^{1,2}. Range expansions of maize lethal necrosis (MLN), for example, threaten global corn production³, and both Dutch elm disease and chestnut blight

fundamentally altered forests in North America^{4,5}. Experimental and field studies have demonstrated that infectious diseases can be highly responsive to changes in temperature and moisture conditions and many studies predict range expansions of infectious plant disease under climate change⁶⁻¹⁰. Surprisingly little evidence, however, directly links climate change to plant disease range expansions^{11,12} (though see¹). This may be a result of data limitations and the presence of confounding factors, such as land-use change and species translocations, that can often obfuscate the climate signal^{1,14}.”

15. Line 56: “Others counter”. I cannot understand this sentence. Observed VS predicted VS no change? I do not understand what the authors intend to say.

Response: Thank you, we no longer use this description, see comment **14**, above.

16. Line 68: Change first "is" to "in".

Response: Thank you, changed.

Methods:

17. Please elaborate what is "plot-level" data in SI Figure 1 and in the main text. Throughout the manuscript, I have no idea what is "plot-level" data and how this is different from tree data.

Response: We used two primary metrics of disease impact. To measure range shifts, we used % prevalence of the disease. This metric is defined as the % of trees in a plot that are infected by blister rust. It is a metric from plot-level data. To help understand the drivers of the changes in prevalence, we explored the responses of individual trees. For these analyses (e.g., Fig 3b, Fig 3d, and 5c), we used infection status of each tree as the binary response variable (i.e., infected, not-infected). Thus these logistic regressions use tree-level data.

We have explained the differences between our plot-level and tree-level data in our method section on lines 450-456:

“We used tree and plot-level data from two surveys, conducted between 1995-1999 and 2013-2017 that spanned 147 plots containing 7,809 white pine stems (Fig. 2a). Tree-level data measured in the field included diameter at breast height (1.37 m, DBH), occurrence of blister rust symptoms (see Supplementary Section 3 for details), and mortality. Plot-level data measured in the field included elevation, slope, aspect (south, southeast and southwest facing = 1, north, northeast and northwest facing = 0), presence/absence of alternative host species (i.e., *Ribes* spp.), and white pine host density (# trees/ha).”

Figure legends:

18. Figure 4, Line 667: “Displaying”. I can't tell what this sentence is referring to in the panel b. Where is CI?

Response: We have clarified this sentence to “Points are the local y maxima for loess smoothed lines with 95% C.I.” We have also included the 95%CI.

Supplement:

19. SI Figure 2: After reading the legends, I am confused whether the analyses were done using GLMM or GAM.

Response: We have changed the wording to make it clearer that we plotted the marginal effects of VPD from the coefficient estimates from the tree-level infection GLMMs. Then, we drew a line from the peak of the estimated curve of this relationship (using the default gam function in ggplot2) to compare between surveys. We have clarified the text as follows:

Supplementary Figure 1: Comparing the marginal effects of VPD between the first and second survey. a, Displaying the predicted probability of infection using the GLMM coefficient estimates of VPD (marginal effects of VPD) for the first (blue lines) and second (yellow lines). Dashed vertical lines are drawn from the peak values of each line (these smoothed lines were estimated using the generalized additive mode smoothing method (“gam”).

REVIEWER 2: Thomas Kolb

Broad Comments

This is one of the most interesting manuscripts I have reviewed in many years. The work significantly advances understanding of climate change impacts on forest pathogens by showing with a convincing suite of analyses that the occurrence of white pine blister rust (WPBR) rust has shifted asymmetrically in concert with increasing vapor pressure deficit over the last 20 years in the central Sierra Nevada in California. WPBR occurrence decreased at lower elevations, due to high VPD and high host mortality, whereas its occurrence increased at high elevations as the climatic conditions for the pathogen moved upslope into more naïve hosts. The work uses a novel combination of empirical, modelling, statistical and physiological approaches to make a very strong case for the asymmetrical shift in WPBR. The model system is well suited for detection of climate-change impacts on climate-host-pathogen interactions. The manuscript writing is clear and engaging. I have only a few comments and suggestions below.

Response: Thank you so much, Dr. Kolb, for this highly accurate summary and your kinds words. We truly appreciate it.

Specific Comments

1. L53: Consider using a less value-laden term than “devastated” here to describe impacts of Dutch elm disease and chestnut blight. The forests where these trees used to occur are still very much intact, but they are different. Consider using “altered” or “restructured” instead of “devastated.”

Response: Thank you for your input. We agree and have modified the wording in response.

2. L109: Revise with a clearer subject than “it”

Response: Thank you, this has been revised to: L113, “As a result of the severe impacts on white pines, WPBR ranks as one of the worst tree pandemics in modern history.”

3. L112: The term “tree physiology” here is not completely accurate. You investigated drought impacts on tree physiological processes. Or, you could use the term tree physiological stress.

Response: Thank you for pointing this out. We agree and have modified the wording: “*in situ* test of drought impacts on physiological processes of host trees.”

4. L118: I understand you are being cautious here by using “likely.” I recommend that you err on the side of boldness and delete “likely.” Your results are convincing to me.

Response: Thank you, we have modified this sentence.

5. L233: Here you cite Figure 5 before citing Figure 4. Figures are usually cited in order.

Response: We agree and we have updated the text so the figures are now cited in order.

6. L316-328: This section could be clearer on the defense mechanisms of white pines against blister rust. What are the defenses, other than stomatal closure? Can you provide a brief example and citation of white pine defenses that are linked with carbon acquisition?

Response: We modified this section to avoid talking about the plant defense chemistry as we did not directly measure plant defense. The focus is instead on the likely shifts in vulnerability to bark beetle attack. Lines 346-356:

“Though water-deficit had more severe consequences for infected hosts, the cause of death was likely a combination of factors related to altered host physiology and increased vulnerability to biotic attack¹⁵. Both mountain pine beetle (*Dendroctonus ponderosae*) and bark weevils (*Pissodes* spp.) can preferentially select trees weakened by blister rust and/or drought^{11,16}; though during extreme drought, bark beetles can kill trees indiscriminately¹⁷. Over the past ~20 years, these biotic attacks likely occurred more frequently at low elevations in SEKI where aridity is higher and pests and pathogens are more abundant^{11,17}. Additionally, recent climate change may have increased physiological stress in infected sugar pine growing near their range limits, potentially increasing the probability of mortality¹¹ from blister rust infections or other biotic agents. Thus trees growing in arid regions were more likely to die following infection, and drought events may have accelerated mortality, particularly in arid regions.”

REVIEWER 3: Dan Bebber

Broad Comments

1. This manuscript analyses changes in white pine blister rust infections between two surveys 20 year apart. While the data and basic analyses appear robust, there are several areas which must be addressed before publication. The two most important relate to the motivation and ecological theory behind the manuscript, and the interpretation of the results.

Response: Thank you for these comments. They have improved the quality and rigor of the analyses, as well as the manuscript's framing and clarity. We have extensively addressed, point-by-point, all of your concerns below.

2. The motivation behind the manuscript is not compelling. The authors establish a 'polarized debate' (Abstract) between those expecting range expansions, and those expecting range shifts, under climate change. I am not compelled by this argument. Fundamentally, either case is possible depending on any changes in the projection of the realized niche into geographic space under climate change.

Response: We have revised both the abstract and introduction. We have taken out all references to the debate and have focused instead on the possible scenarios that could emerge as pathogens shift in space and interact with the changing abiotic-biotic environment. Please see below for a completely re-written abstract and lines 56-125 for a majorly reframed introduction, Ls 33-45:

“Infectious plant disease range shifts are expected under climate change. As pathogens migrate, emergent abiotic-biotic interactions are predicted to modify disease distributions, leading to unpredictable changes in disease risk. Evidence of these complex range shifts remains largely speculative, however. Combining a long-term study of the tree disease caused by *Cronartium ribicola* with a six-year field assessment of drought-disease interactions in the southern Sierra Nevada, we found that climate change moved the climate disease optimum into higher elevations. This shift likely decreased prevalence by 5.5 (4.4, 6.6) p.p. in arid regions and increased prevalence by 6.8 (5.8, 7.9) p.p. in colder regions. Though climate change increased the suitable area for blister rust by an estimated 367.7 (0, 514.2) km², the combination of spatially-varying host-pathogen interactions and drought impacts contributed to a 32.79% decrease in prevalence between surveys. Here, complex host-pathogen-drought interactions initially dampened disease risk, but the long-term impacts of range expansions remain concerning for naive hosts.”

3. The authors cite one of Jorge Soberón's extremely insightful BAM model for species distributions, and I strongly recommend that they also read his other works on this model (Soberón, 2007; Soberón & Nakamura, 2009). In the interests of openness, I have advocated for the application of this model in understanding changes in the distributions of plant pathogens (Bebber, 2015) and believe it would be helpful here.

Response: We thank the reviewer for calling out the BAM framework and suggesting we engage with it explicitly in the paper. We have completely revised the conceptual figure to better incorporate the BAM model. Additionally, our focus in this paper was less about disease distribution modelling, but rather to estimate the causal link between disease and climate in order to estimate the impact of climate change on prevalence. This causal objective would be more limited using widely applied SDM's (see¹⁰).

We also aimed to describe and disentangle how the direct effect of climate change on disease prevalence may have been modified by interacting abiotic-biotic factors within the niche space that we captured, which we agree, fits nicely into the BAM framework. The two most important biotic (B) factors involved in the niche of this pathogen are white pine hosts and *Ribes* alternate hosts, which we take into account. We acknowledge that movement (M) may influence this shift as well, and so we tested whether spatial migration patterns might lead to a biased estimate of the relationship between climate (A) and the abundance or prevalence of the pathogen (see linked comment **31** below). Thus, we definitely agree with the reviewer that considering the BAM factors is a useful framework for disease distributions. To acknowledge this, we now cite Bebbler 2015 and mention the BAM framework explicitly in the text. We have also added biotic and abiotic factors into the conceptual figure to more explicitly draw this framework out (Fig 1 here).

Fig. 1. Conceptual figure of the biotic-abiotic factors that can interact with *Cronartium ribicola* as it moves into higher elevations in response to climate change (sensu the biotic-abiotic-migration (BAM) framework^{18,19}). These interactions can further modify the spatial position and shape of the white pine blister rust distribution. Specifically, climate change is predicted to shift the pathogens' thermal optimum in space (a), which could lead to an upslope migration (a). As the pathogen shifts, emergent abiotic and biotic interactions can alter the disease distribution. Climate change-induced increases in stochastic disturbances, like droughts (b), can interact with both the host and the pathogen at different spatial scales. The probability of drought stress, for example, is likely higher at low elevations where water is more limiting, potentially resulting in a skewed distribution (b). Additionally, host-pathogen interactions (c), including the varying density and susceptibility of different hosts and/or alternate hosts, can also modify the size and shape of the distribution. (Hosts and alternate hosts are expected to shift at a slower pace than pathogens in response to climate change). The combination of these spatially varying drought and host-pathogen interactions can

modify disease range shifts to increase or decrease prevalence depending on the direction of the interacting effects (c). Mountain figure designed by Zuzanna Drozd.

4. B, the region in geographic space corresponding to the biotic niche, in this case is determined by the distributions of susceptible hosts (itself determined by climate). A, the geographic area corresponding to the abiotic (climate) niche, is the one affected directly by climate change. M is the geographic region reachable by movement, in this case we expect this to be unrestricted due to aerial spore dispersal. Explicitly plotting these geographic regions on a map (or at least A and B) and showing the overlap under past, current and future conditions, would be a very strong contribution to the literature. A map, and perhaps a schematic illustrating a mountain, would really help readers to grasp what is happening in this system. The schematic in Fig 1 goes some way toward this, but I didn't find it intuitive to understand.

Response: These are great suggestions. We have incorporated this feedback and have included abiotic/biotic/drought interactions in the schematic, as well as created Fig 6, see below, that uses this framework to illustrate the abiotic-biotic interactions that we tested for and included in this study. We think this has majorly improved the clarity and framing of our paper and provides a really nice visual to bring the drought study, *Ribes*, and host mortality analysis (which are less conducive for maps, as they are plot-level data) together with the disease prevalence results. We also attempted to overlay blister rust range shifts onto Fig. 6, but it was difficult with so many overlapping colors. So, we included a map (see Fig. 2 below) to display the shift in disease as a result of climate change and then used Fig 6 to illustrate the spatially varying abiotic-biotic interactions across elevation that also likely modified disease prevalence.

Fig. 1. Spatially varying abiotic-biotic interactions that likely modified blister rust range shifts. *a*, Predicted relative needle expansion shifted from more negative (-0.7 cm) to positive (0.64 cm) across elevation in response to drought (**Error! Reference source not found.**). The probability of *Ribes* occurrence (*b*) decreased at higher elevations (probability ranged from 0.80 to 0.09) (**Error! Reference source not found.**). *c*, Probability of infected host mortality declined with increasing elevation from 0.85 to 0.36 (**Error! Reference source not found.**). *d*, Observed blister rust prevalence and 95% C.I. for the first and second survey across elevation. Green shades show the elevational ranges of the susceptible white pine hosts. Mountain figure designed by Zuzanna Drozd.

5. I found the Introduction rather confusing and at odds with ecological theory and the discourse on climate change effects and species distribution modelling. Phenomena like “multidirectional climate change impacts on disease prevalence” (L 70) and “asymmetrical range shifts” (L.) become less important when we consider that species distributions are determined by suitable areas in geographic space. Different types of shifts can be expected at different geographic scales, different topologies, and different climate responses of hosts and pathogens. At the global scale, warming is happening faster at higher latitudes, so we can expect faster range expansions at higher latitudes than range contractions at lower latitudes. For plants, day length is a limiting factor so this will moderate range expansion in comparison with any pathogens. A nice example of potential mismatches between range shifts by a pest and host is given by (Berzitis et al., 2014). At the local scale, for example a mountain, asymmetric shifts are expected due to the shape of mountains. Because the land area shrinks with altitude, range expansions at high altitude will necessarily be smaller than range contractions at low altitude, other things being equal. The BAM model clarifies and simplifies all these considerations.

Response: We agree that the ecological theory highlights the importance of these different geographic scales, topologies, suitable spatial scales, and biotic interactions in shaping distributions. We have majorly reframed the introduction to reflect the literature including Berzitis et al., 2014, see lines 53-120. We have also cited an additional 15 papers to better integrate the literature into both the introduction and discussion.

6. My second major concern relates to the way that disease incidence/risk is modelled in relation to climate. The authors find a significant relationship with VPD, which they correctly identify is a function of temperature and humidity. The problem is, that this statistical relationship hides the mechanistic relationship between temperature, moisture, and disease risk. Rust fungi in particular tend to require liquid water on the leaf surface, or at least high humidity, to germinate and infect. I’ve developed some models for this, but to avoid over-citing my own work I found the model by Robert Magarey and colleagues to be very helpful (Magarey et al., 2005). The authors mentions “hump” shaped relationships of infection rates to temperature (temperate performance curves), and may wish to read our recent synthesis of these for fungal plant pathogens (Chaloner et al., 2020). I have just checked our database and we have cardinal temperature estimates for *C. ribicola*, which the authors might like to discuss, or even implement in a mechanistic model (Bebber et al., 2020). A statistical model based on VPD hides the biological mechanism behind infection risk and doesn’t help us understand how climate is driving range shifts. The non-linear function of temperature and humidity that gives VPD means that the same VPD can occur at 32C, 67% RH and 21C, 38% RH. Confounding temperature and moisture by using VPD as a

predictor makes interpretation of the result difficult. While statistical species distribution models remain popular and can often give a reliable result, they suffer from several issues including extrapolation outside the training range. This is particularly problematic for climate change projections. A forceful critique of these methods is given by Sutherst (2014).

Response: Thank you for this important comment. While we agree that it is less common to use a derived, integrated term like VPD to estimate the relationship between climate and pathogen prevalence, we chose to use VPD in our analysis for two main reasons. First, an increasing number of studies demonstrate the integrated VPD term can outperform or complement temperature as a predictor of various diseases and their vectors. For example, Davis et al. (2018) found that the best-fitting model estimating West Nile virus outbreaks included VPD. Similarly, de la Vega and Schilman (2017) found that their models using VPD (they interpreted high values as “hot and dry”) were important predictors of Chagas disease vectors and recommended that their approach should be used for predictions of movement of disease-causing vectors into novel regions. Further, Talley, Coley, and Kursar (2002) estimated favorable habitat for 25 fungal pathogen communities in the Intermountain West and found that “measures of moisture availability, such as relative humidity and vapor pressure deficit, explained more of the variance in fungal abundance and richness than did temperature.” Finally, Manstretta and Rossi (2015) found that both “precipitation and VPD were the variables that most affected ascospore discharge” of *Fusarium graminearum*. These studies suggest that VPD may better capture the biological mechanisms associated with infection and spread of diseases than temperature alone (particularly *in situ* at the landscape-scale) likely because VPD is integrating the moisture content of the air, which is often important for pathogen reproduction, including *C. ribicola*, as you pointed out.

Second, when we conducted our model selection, the inclusion of VPD in our GLMMs led to the lowest AIC values compared to all other climate variables, including temperature, precipitation, and dewpoint temperature (see S. Table 1). Thus, we followed recommended statistical approach and chose the most parsimonious model to conduct our full climate change disease analysis. Additionally, following these standard statistical methods, we concluded that our estimations of climate change impacts on the blister rust pathosystem would be the most robust using the model that explained the most variance—the FE panel model with VPD.

We fully acknowledge and agree that VPD is less interpretable than temperature, however, and we have added a temperature analysis to the supplement that includes both GLMMs and FE panel models. These analyses complement the VPD analyses by showing that our results can also be interpreted in temperature space—that is, rising temperatures are also significantly associated with changes in the prevalence and elevational range extension of blister rust. We include these supplemental results here.

Supplementary Figure 2: (a,c) Proportion of infected white pines (yellow line) and the number of tree stems across maximum temperature for the first (1995-1999) and second (2013-2016) surveys. (b,d) Proportion of infected white pines (yellow line) and the number of tree stems across dewpoint temperature for the first (1995-1999) and second (2013-2016) surveys. Shaded areas of yellow line represent 95% C.I around local non-parametric regression (loess) estimates.

Predictors	First survey max. temperature			Second survey max. temperature		
	Coefficient	Std. Error	P-Value	Coefficient	Std. Error	P-Value
Intercept	-6.39	0.54	<0.001	-5.50	0.95	<0.001
Max temp.	3.47	0.99	<0.001	1.37	0.41	0.001
Max temp.^2	-1.18	0.49	0.016	-0.44	0.18	0.013
Density	-0.58	0.38	0.129	0.07	0.20	0.723
Ribes	2.15	0.67	0.001	1.07	0.44	0.016
DBH	-0.32	0.07	<0.001	0.09	0.09	0.293

Slope	-0.27	0.29	0.344	-0.02	0.18	0.892
Aspect	-0.15	0.33	0.657	-0.44	0.22	0.045
N	147 _{plot}			147 _{plot}		
	4 _{species}			4 _{species}		
Observations	7031			5457		
Marginal R ² / Conditional R ²	0.61 / 0.85			0.19 / 0.67		
AIC	1990.8			1107.4		

Supplementary Table 2: Estimated relationship between blister rust infections and maximum temperatures. Showing logistic regression estimates, standard errors, and corresponding p-values for GLMM estimating first and second survey tree-level infections with plot and white pine species as crossed random effects. Pseudo R² values calculated using the “MuMIn” R package, which calculates a revised statistic for mixed effects models.

FE panel model with temperature			
Predictors	Coefficient	Std. Error	P-Value
Max temps	0.229	0.10	0.02
Max temps ²	0.008	0.002	0.00
DBH	-0.002	0.002	0.31
Density	-0.000	0.000	0.09
Observations	147		
Adj. pseudo R ²	0.57		

Supplementary Table 3: Fixed effects panel model results with maximum temperature. Showing coefficient estimates, standard errors, and p-values for four measured time-varying variables. Standard errors are clustered by plot and adjusted pseudo R² is equivalent to the GLM model pseudo R² with a Gaussian family¹⁵.

We also found that when we ran our fixed effects panel model, VPD is a slightly better predictor than temperature (higher R²)—leading us to conclude that while VPD is acting very similarly to temperature, the inclusion of humidity is better capturing the changes in the abiotic climate that are interacting with the spread of blister rust.

In summary, Supplementary Fig. 1 shows the relationships between temperature and dewpoint temperature with blister rust prevalence for the first and second survey. Supplementary Table 3 shows the GLMM results using max temperature instead of VPD. (Supplementary Table 4). We find that VPD outperforms max temperature across most metrics. We conclude that we could

also conduct our entire analysis using temperature and it would provide very similar results to our VPD analysis.

In addition, we appreciate the major advance that Chaloner et al., 2020 make and estimating the temperate performance curves would be a great next step for future research. Our approach, however, was aimed at leveraging the observational data using a FE panel model that also enables a more causal interpretation of the estimated relationship between climate and disease. Indeed, the beauty of the fixed effects panel model is that it allows us to control for a diversity of time-invariant observable and unobservable variables without explicitly estimating their individual effects (we do estimate the observed variable effects in our GLMMs, however, to complement our FE panel model results). This gives us a more accurate estimate of the marginal effect of climate change that isn't confounded by the other important determinants of prevalence. See response **11** above to R1 for more details.

Specific Comments

7. 33, 55. I don't agree that there's a significant debate. Ranges will change in various ways depending on the changes in the BAM regions. As Altizer et al. (2013) write "Climate change will continue to limit the transmission of some pathogens and create opportunities for others". The debate they discuss is more around the different results from statistical and mechanistic models. I disagree with the way that Lafferty (2009) is cited. Much of his paper discusses the differences between statistical and mechanistic models. However, he does reinforce the importance of the BAM framework "...while a reduction in habitat suitability should reduce a species' range, an increase in habitat suitability does not necessarily result in an increase in geographic distribution. This is because other factors besides climate, such as barriers to dispersal, competition, and predation, affect the realized niche."

Response: We have majorly reframed our introduction in response to this comment, including the first paragraph, which now reads:

Lines 52-61: "Infectious plant diseases are reshaping ecosystems, disrupting global food supplies, and threatening human health²⁴⁻²⁶. Range expansions of maize lethal necrosis (MLN), for example, threaten global corn production²⁷, and both Dutch elm disease and chestnut blight fundamentally altered forests in North America^{28,29}. Experimental and field studies have demonstrated that infectious diseases can be highly responsive to changes in temperature³⁰ and moisture conditions³¹ and many studies predict range expansions of infectious plant disease under climate change³²⁻³⁶. Surprisingly little evidence, however, directly links climate change to plant disease range expansions^{18,37} (though see³⁸). This may be a result of data limitations and the presence of confounding factors, such as land-use change and species translocations, that can often obfuscate the climate signal^{24,39}."

8. 68. Not clear what relationship this is referring to. Rather than "multidirectional" do you mean a nonlinear relationship with temperature?

Response: Yes, we have changed it to nonlinear.

9. 83. Again, asymmetrical range shifts are to be expected in all be rare cases of perfectly flat topography with no confounding biological relationships etc. Asymmetry vs. symmetry is not really the important question here.

Response: We have now largely taken out the discussion of asymmetry and focused on shape of the disease distribution changing in response to interacting abiotic-biotic factors.

10. 88. There are new citations for drought projections, e.g. (Cook et al., 2018).

Response: Thank you, we included this new citation.

11. 95. What is a multidirectional impact of drought?

Response: To improve clarity, we took this multidirectional word out. It now reads:

Lines 83-84: “predicted increases in drought frequency could result in more skewed disease distributions under climate change (Fig. b).”

12. 118. The paper doesn't discuss the thermal response functions of WPBR.

Response: We agree, our description in the paper was very short. We have added a supplement to explain the known relationships between WPBR and climate:

Supplementary Section 1: Blister rust life cycle and climate tolerances

White pine blister rust is a macrocyclic heteroecious rust; it has a complex life cycle that alternates between the white pine aecial hosts and the alternate telial hosts from the genera *Ribes*, *Castellja* and *Pedicularis*^{4,40}. The spores are largely wind dispersed from both hosts. Aeciospores, for example, can travel up to an estimated 1,200 km via wind currents to infect telial hosts⁴¹ during the spring and summer. In contrast, winds carry fragile basidiospores, produced on the telial hosts, much shorter distances (< 3 km) to infect pine hosts through needle stomata during late summer^{42,43}.

Both temperature and moisture differently affect various parts of the complex life cycle, including the rate of infection, spore-bearing structure development, germination, and spore release. Teliospores, for instance germinate best within 4 to 9 days of age and following 12 hours of exposure to high moisture conditions at moderate temperatures (the optimum temperatures range is between 10-18° C (viable range 0-22° C))^{7,30}. Basidiospore production from telia begins when moisture is high (including contact with water or > 97% relative humidity), and the optimal temperature range for germination is also between 10 - 18° C^{7,44}. Aeciospores, produced on white pine hosts in the spring, can tolerate warmer temperatures than basidiospores and teliospores ranging from 16 to 28° C⁷. Saturated air or free water is required for aeciospores to survive. Generally, blister rust reproduces best under high moisture and moderate temperatures and is therefore considered a cool weather disease⁴⁵.

13. Fig. 1. A figure which more explicitly shows the geographical areas of suitable hosts and infection risk would be more helpful. Perhaps adapt the map from Supplementary Figures.

Response: Great idea, this suggestion has much improved the clarity and informativeness of Figs 1 & 2. We have now majorly revised Fig. 2 to include the four white pine host species, presence of *Ribes* spp., blister rust prevalence, the plot locations, and the average weather variables across these plots. We have also included a mountain illustration to increase the clarity of our conceptual Fig. 1.

Fig. 2: Site description. **a**, map of SEKI and the long-term monitoring plots. Includes plots with white pine blister rust infections in the first survey (purple dots), plots with new infections in the second survey (orange dots), and plots that remained uninfected (black dots). Color fills denote ranges of white pine hosts and stars illustrate plots with the presence of alternate hosts (*Ribes* spp.). **b**, Mean annual VPD between 1975 to 2016; blue dashed lines denote the time period of the first and second survey. **c**, Total annual precipitation (blue bars) across all plots. **d**, Mean annual maximum and minimum temperatures; dashed horizontal lines show the mean weather value across all year.

14. Not sure if this site is unique in this regard.

Response: Our aim was to describe why the SEKI data was highly suitable to estimate the effects of climate change across a large portion of the abiotic-biotic niche, given the four hosts, alternate host occurrence, and the large elevational gradient.

15. 174. It would be interesting to use a beta function or similar to map risk, using these temperature tolerances. This would give a semi-mechanistic model to compare with observations.

Response: Thanks for this suggestion. We agree that representing the temperature tolerance envelope of the species with such a function could produce nice range maps. For this paper, our aim was to leverage observational data using complementary modelling approaches to characterize climate effects across an aridity gradient and rigorously quantify the impacts of climate change on disease prevalence. Your suggestion would be a great next step for this research.

16. 195. The “hump-shape” can be modelled with a beta function (or others).

Response: Yes, we completely agree. In our analyses, however, the quadratic didn’t have any real fit problems, and worked well for this particular application. We also chose the quadratic because it enabled the analyses to be statistically and theoretically consistent with causal inference fixed effects (FE) models. FE models don’t yet enable beta functions, but they do allow quadratic terms. So by using a quadratic term in both the GLMMs and FE models, these models informed each other and verified that the relationship between climate and climate change was in fact significantly “hump-shaped”. We also agree that using a beta function for risk mapping would be an excellent next step.

17. 203. Again, I’m uncomfortable with the use of VPD as it combines two variables, temperature which determines growth rate, and moisture which determines ability to grow.

Response: Thanks for this comment. Please see our response to your second major concern, linked comment **20** above.

18. 221. What does ‘multidirectional’ mean in this context?

Response: We no longer use this term.

19. 241. The linear change in elevation is not a range expansion. Ranges are areas, so the range expansion should really be measured in hectares in this instance. Because mountains are pointed, the actual range increase will likely be smaller.

Response: This is a fair point, movement upslope does not tell you the new area occupied, and our wording of this sentence suggested that we were equating the two. We have now estimated the area and 95% C.I. that was exposed under observed and future climate change by clipping the SEKI DEM and white pine host species map by the estimated max and min elevation extension/contraction from our MC simulation, please refer to lines 601-616 in the manuscript. This produced the following map (see Fig 4e below). In our study system, we found that the suitable area increased under observed climate change due to the unique spatial configuration of white pine hosts in this system (greater proportion of hosts occurring in high elevations where climate suitability for blister rust was historically very low).

Fig. 2. Climate change impacts on blister rust in SEKI. *a, c*, Displaying the raw values of VPD and blister rust prevalence. *a*, Changes in the distribution of VPD between the two survey periods (1975-1994 and 1995-2016) and predicted VPD for 20 CMIP5 RCP4.5 (2020-2060) scenarios. *c*, Measured plot-level prevalence (% infected trees/plot) across elevation with loess smoothed lines and 95% C.I. *b, d*, Displaying the Monte Carlo (MC) simulation results of the FE panel model predictions. *b*, Random sample of 1000 predicted values of blister rust prevalence for the three VPD scenarios across elevation; “Counterfactual” is the no-climate change scenario and “Climate change 2016” scenario corresponds to the observed change in VPD over the past ~20 years. Points are the local y maxima for loess smoothed lines and 95 % C.I. *d*, Estimated percentage point (p.p.) difference in predicted prevalence attributed to a change in VPD; Climate change 2016 and RCP4.5 2056-60 values were differenced from the counterfactual at the three elevation terciles. Displaying 95% C.I. error bars. *e*, Estimated blister rust expansion (red and orange) and contraction (aqua) for the two climate change scenarios compared to the counterfactual (blue). First map shows upper elevation expansion estimated under observed climate change 2016 (red) relative to the estimated counterfactual (blue). Second map shows estimated upper elevation expansion (orange) under RCP4.5 2056-60 and estimated contraction at lower elevation (aqua). Grey gradient fill on both maps illustrates the digital elevation model (DEM). Green shading in both maps shows remaining uninfected host range.

20. 263. The manuscript is lacking a good description of the WPBR life cycle, and how climate affects different processes within the cycle. At minimum, we should know the role of moisture and temperature in the infection process.

Response: Thank you, this is an excellent suggestion. We have now added a Supplementary Section 1 which describes the life cycle and the thermal/moisture tolerances (see response to your comment 25 above).

21. 278. Again, it would be great to see maps of host mortality and physiological stress, and how these have changed/might change in future.

Response: While creating maps of physiological stress and host mortality would be ideal, it would potentially be inaccurate given the plot-level data that we would be using to extrapolate to a very heterogenous landscape. However, to incorporate this excellent point, we have added in a quantitative illustration depicting the mortality and host physiological stress across elevation (see **Fig. 2** pasted in comment 19). We use the estimated ranges of mortality and relative growth of hosts from our plot-level data to demonstrate where these occur across the mountain map. We

hope this clarifies the results and presents a more informative picture of how these different factors are interacting with the spread of blister rust as it moves into higher elevations in response to warming.

22. 299. Interesting result.

Response: Thank you.

23. 316. Interesting result. Droughts do different things to host susceptibility, depending on the pest/pathogen. For insects, leaf chewers are negatively affected because of reduced production of fresh leaves. Stem borers benefit because of reduced defensive sapflow (in conifers).

Response: Agreed. Droughts will not have a uniform impact on different pests/pathogens (Kolb et al 2016). For many primary foliar fungal/fungal-like diseases that require entry into the host through stomata (including rusts), droughts and declining water-availability (e.g. increased evapotranspiration) often result in decreased host susceptibility (Kolb et al 2016; Erye et al 2013, *Ecology and Epidemiology*; Desprez-Loustau et al., 2006 *Annals of forest science*). And the reduced sapflow you mention in conifers might be a mechanism for why we often see weevils boring into blister rust cankers in the field.

Additionally, we have revised our results and discussion of the abiotic-biotic factors interacting with blister rust range shifts in response to R1's comment **8** above.

24. 346. Again, I don't think 'multidirectional' and 'nonlinear' are used in the right way here. The focus of the discussion should really be on a) what aspect of climate change drove the shift in range (avoiding discussion of VPD, as this is not directly a driver), and b) how biotic interactions with the host affected the shift. (Berzitis et al., 2014) is a good model for this. WRT malaria, remarkably a non-linear temperature response was only included in models in 2013 (Mordecai et al., 2013) - before this models assumed that transmission would simply keep increasing with temperature!

Response: Thank you for pointing out this excellent example, which we have now included in the manuscript with summaries of the main results and important conclusions we drew from these results on lines 395-433:

“Here we demonstrated that changing climatic conditions between 1996 and 2016 in SEKI likely decreased blister rust prevalence by 5.5 (4.4, 6.6) p.p. at low elevations and increased prevalence by 6.8 (5.8, 7.9) p.p. at high elevations (Fig. 4d). These changes indicated that the climate optimum for blister rust shifted upslope in response to climate change, expanding the suitable area for blister rust infection into higher elevations by an estimated 367.7 (0, 514.2) km². Thus, even with small increases in mean VPD (~1.35 hPa) between surveys, the impact on the blister rust pathosystem was detectable (an effect that has long been predicted^{12,16}). Under future climate change scenarios, blister rust prevalence may contract further at low elevations, as climatic conditions become too hot and dry, while less extreme climatic

conditions at high elevations may simultaneously expose the majority of high elevation hosts in SEKI (Fig. 4e).

Though the direct effect of climate change likely increased the suitable area for blister rust infection in SEKI, mean observed prevalence declined by 32.79%, highlighting that the direct effect of climate change can be strongly mediated by complex host-pathogen-drought interactions. The probability of infection decreased at higher elevations, for example, in part due to fewer alternate host species (Fig. 6b). These emergent interactions have long-been anticipated^{1,14,40,74}, and we provide evidence supporting the prediction that host availability is more important for predicting shifts in endobiotic pathogen distributions³¹ and disease risk. Additionally, drought events, as well as the aridity gradient, interacted with white pine hosts, leading to faster declines in infections in arid regions that were not offset by increases at high elevations (Fig. 6d). Thus, the blister rust range shift was not only moderated by host species interactions, but also by abiotic feedbacks that initially dampened infection risk. These abiotic-biotic interactions may be nonlinear, potentially shifting in strength and direction as global warming proceeds^{75,76}. Investigating the context dependencies of plant-pathogen-drought interactions will be critical to estimate the magnitude of these interacting effects on plant pathogen range shifts.

Though climate change is often predicted to increase susceptibility to pathogens at the host range edges, our results suggest the opposite. The thermal mismatch hypothesis, for example, predicts that host susceptibility will increase as climate change shifts hosts away from their optimal temperatures¹⁶, which has been found in infectious amphibian disease⁷⁷. Similarly, root rot pathogen genera, including *Armillaria* and *Heterobasidion*, are predicted to expand their ranges during drought because they successfully colonize stressed trees^{6,78}. In contrast, for pathogen that infect through needle stomata, we found that extreme drought likely had opposing effects on host susceptibility, increasing infection probabilities at high elevations where the growing season was extended and decreasing susceptibility in arid regions where growth declined (Fig. 6a). Identifying how host infection surfaces (i.e., roots, leaves, stem tissue) respond to water-deficit across aridity gradients will be critical to forecast climate change-induced disease range shifts across heterogenous terrain.”

25. 393. The map is really an important and should be a main figure.

Response: We agree—thanks for this suggestion. We have combined the map and the corresponding changes in climate into Figure 2. Please see response to **25** above for more details.

26. 400. 4km is still coarse resolution, particularly given the steep topography. How did you account for altitudinal changes in temperature within pixels? New tools are available for this (Bütikofer et al., 2020).

Response: We agree that in mountainous terrain the 4km pixel size can cover quite a wide range of elevations and is not ideal for temperature and VPD estimation at the scale of the plot. We used the larger pixel size as it was the data available to us, and we do not have the expertise to downscale it ourselves. That said, there is little reason to expect the errors to be systematic (e.g., biased in one direction or another), and such errors would be more likely to weaken rather than

strengthen relationships with climate. Therefore, we believe that downscaling would be unlikely to change our qualitative result. We agree, however, that this would be ideal terrain for future analyses aimed at predictions or forecasts.

27. 437. The GLMMs appear very thorough but did you control for spatial autocorrelation among plots? I would expect strong spatial autocorrelation because of aerial spore dispersal.

Response: This is certainly a critical point that we sought to address in the previous version of our paper; however, this comment and R1's comment 2 highlights that these tests were not clearly presented and explained. We have now clarified our approach and presented additional tests here to control for spatial autocorrelation.

To control for spatial autocorrelation in observational studies of organisms, the following four-step approach is generally recommended in ecology²: 1) consider the potential causes of spatial autocorrelation and the spatial scales they may be acting on, 2) conduct tests for spatial autocorrelation in raw data, 3) if there is spatial autocorrelation in the raw data, use covariates that may be causing the spatial autocorrelation and/or test to see if using a hierarchical GLMM with a random term, like site/plot, accounts for spatial dependencies, and 4) if spatial autocorrelation is unable to be controlled for by model covariates, conduct additional analyses that can account for the spatial configuration of the data. We conducted steps 1-3 and found that using our covariates, including plot as a random effect, controls for positive spatial autocorrelation, the values largely of concern². However, to further verify these results, we added two more robustness tests that confirmed our results are unlikely to be biased by spatial autocorrelation.

Step 1) The most common cause of spatial autocorrelation in species distribution data is dispersal³. While other environmental factors can also exhibit specific autocorrelative patterns (e.g., topography, climate), we can more easily control for these sources simply by including these variables in our GLMMs. In our system, the dispersal of blister rust from *Ribes* to pine hosts and vice versa could cause spatial autocorrelation. Our study area (~100 km x 70 km), is much smaller than the estimated range that aeciospores can travel from white pines to infect *Ribes* (~1,200 km as reported by Geils, Hummer, and Hunt 2010). Thus, given the spatial scale of our prevalence data, dispersal by aeciospores was unlikely a major cause of spatial autocorrelation. In contrast, estimated dispersal from *Ribes* to white pines is much shorter, a few kilometers at most⁴; thus dispersal from *Ribes* may cause positive autocorrelation in the raw data at shorter scales, likely less than ~3km.

Step 2-3) Following methods outlined by Zuur et al. (2009), we estimated a spline correlogram with 95% pointwise bootstrapped CI (grey) and Pearson residuals autocorrelation (Fig 1). We see that indeed, in the raw data of blister rust prevalence from the first survey, there are some signs of positive spatial autocorrelation at shorter distances, but the confidence intervals are crossing 0 (Fig 1).

Fig 1. Spline correlogram and 95% CI using blister rust prevalence from the first survey and corresponding Distance measured in UTM's ($0.2^\circ = 22\text{km}$). The distance at which *Ribes* is likely acting on the system is less than $\sim 3\text{km}$, or less than 0.02° . This is where we see a little bit more positive spatial autocorrelation than estimated at longer distances, but again, the 95% CIs cross 0.

To test whether adding in *Ribes* occurrence helps control for this positive spatial autocorrelation, which we expect to be the likely cause, we see that indeed, the spatial autocorrelation is effectively controlled for by including *Ribes* as a covariate (Fig 2).

Fig 2. Spline correlogram and 95% CI including *Ribes* only as a covariate. Distance measured in UTM's ($0.2^\circ = 22\text{km}$).

When we include all covariates and plot as a random effect (Fig 3), which we used in our GLMMs estimating infections, we see that that plot does perhaps control for a little more spatial autocorrelation. Neither of these correlograms identify significant positive autocorrelation, however.

Fig 3. Spline correlogram and 95% CI including all covariates and plot random effects.

Additionally, we find that the neither the raw data (Fig. 4 left), the models just with *Ribes*, nor all variables (Fig 4 right) exhibit significant positive spatial autocorrelation using the second survey prevalence (Fig. 4); Right figure with all covariates has a mean correlation of 0.00096 with a 95% CI of (-0.276, 0.265). This suggests that our analyses estimating disease prevalence are not strongly biased by spatial autocorrelation.

Fig 4. Left, spline correlogram and 95% CI of raw second survey data. Right, spline correlogram and 95% CI including all covariates.

Step 4) In an effort to verify these results, we added two more robustness checks:

Test A. Testing for possible confounding by dispersal dynamics in our blister rust-VPD relationship. On lines 513-523, following methods also described by Baker-Austin et al. (2013), we described how we tested whether blister rust dispersal had confounded the VPD coefficient estimates. Specifically, we fitted a GLMM with an interaction between time period and the linear and quadratic VPD terms. If coefficient estimates of the interactions between VPD and time were significant, this would suggest that the relationship between prevalence and VPD had changed through time. A change in the relationship between VPD and blister rust would suggest that other factors, like dispersal, may be driving changes in prevalence. If the interactions were insignificant, however, this would suggest that blister rust dispersal was unlikely to be confounding. Most importantly, a similar VPD-blister rust relationship between surveys would help confirm that blister rust would likely track changes in VPD – which was important to test

for given that we were estimating the effect of a change in VPD on the change in prevalence in our FE panel models. Our fitted models demonstrated that our coefficient estimates were unlikely to be biased by blister rust dispersal.

Blister rust model with time interactions			
Predictors	Coefficient	Std. Error	P-Value
Intercept	-5.19	1.11	< 0.001
VPD	2.53	0.45	< 0.001
Time	-0.17	0.30	0.574
VPD ²	-0.83	0.18	< 0.001
Ribes	1.16	0.38	0.003
Aspect	-0.48	0.37	0.198
Slope	-0.02	0.02	0.428
Density	-0.63	0.18	0.001
DBH	-0.31	0.07	< 0.001
VPD*Time	-0.52	0.44	0.2 ¹ 31
VPD ² *Time	0.23	0.17	0.169
Density*Time	1.03	0.20	< 0.001
DBH*Time	0.41	0.10	< 0.001
N _{plot}	147		
N _{species}	4		
Observations	12447		
Marg. R ² / Cond. R ²	0.350 / 0.77		

Supplementary Table 4: Logistic regression estimates, standard errors, and corresponding p-values for GLMM estimating infected trees with time (first and second survey) as interaction terms. The relationship of VPD and VPD² is not significantly different between the first and second surveys (interaction terms were not significant), suggesting that the relationship between infections and climate has remained relatively stable through time.

Test B. Estimating dispersal distance between new infections and nearest known historic infection. Using the first and second survey data, we also estimated the geographic distance between any new infected tree and a known infected tree in plot_i at t₀. Including distance to nearest infection at t₀ for each tree as a variable in the GLMM model estimating new infections at t₁, we find that including this variable into the *model does not improve model fit (AIC is slightly higher), the variable is not significant, and our estimates of the primary coefficients of interest are largely unchanged.*

Predictors	Second survey without dispersal variable			Second survey WITH dispersal variable		
	Coefficient	Std. Error	P-Value	Coefficient	Std. Error	P-Value
Intercept	-5.32	1.04	<0.001	-5.31	1.04	<0.001
VPD	1.93	0.51	<0.001	1.91	0.55	0.001
VPD ²	-0.56	0.18	0.002	-0.55	0.19	0.004
Density	-0.05	0.21	0.809	-0.05	0.21	0.807
Ribes	0.96	0.44	0.028	0.95	0.45	0.035
DBH	0.12	0.08	0.157	0.12	0.08	0.156
Slope	-0.00	0.02	0.852	-0.00	0.02	0.848
Aspect	-0.96	0.43	0.026	-0.96	0.43	0.025
Nearest infection				-0.04	0.32	0.911
N	147 _{plot}			147 _{plot}		
	4 _{species}			4 _{species}		
Observations	5416			5416		
AIC	1100.5			1102.5		

Table 2. Showing the difference between GLMM models estimating second survey infections that do not include (left side) and include (right side) an estimated distance to nearest infection. Note that the AIC without the dispersal variable on the left is slightly lower and the variable “Nearest infection” is not significant at $P = 0.911$.

Our GLMM results in Table 1 may be a result of many factors, including: 1) this dispersal distance variable has high measurement error (likely) and 2) dispersal dynamics were not well captured at the spatial scale of our study (likely true in the case of aeciospores).

Finally, GLMMs do not allow us to control for endogenous and exogenous unmeasured variables that may also be spatially autocorrelated. The fixed effects (FE) panel modelling approach, however, does allow us to control for all unmeasured variables that are spatially autocorrelated and time-invariant⁶. Also see comment 7 below.

We have revised the text on lines 508-527 to include and better describe these tests ensuring that our primary conclusions are robust to possible confounding due to dispersal or other potential causes of spatial autocorrelation.

28. 458. These types of issues are avoided when using a mechanistic model of the biological process.

Response: Yes, mechanistic models certainly advance the field in this regard, though they are less practical in these highly complex observational studies where multiple variables are correlated with climate.

29. Fig. 2. Why is only T_{min} given?

We only included T_{min} because t_{max} and t_{min} were very highly correlated (~.95) and the figures appeared redundant. We have now included both in Fig. 2 above.

30. Fig. 3. I don't think graphs of coefficient estimates are helpful. Better in text or tabulated.

Response: We completely agree that tables are very useful for displaying model results and we have created tables for all model results throughout this paper that have been placed in the supplement. We made coefficient figures for three reasons: 1) to draw a clear visual link between the estimated relationship of VPD and the raw prevalence of blister rust without the reader having to find both the figure and the table, 2) to show that the nonlinear, hump-shaped relationship between the observed prevalence and VPD corresponds to the largest coefficient estimate in the disease model and 3) to clearly illustrate that the coefficient estimate of VPD is consistently significant through time, as well as in the panel model. This figure was created to show in one place that the raw data and our statistical models are telling a very consistent story. If you still think this approach is poorly thought through, we are happy to change it.

31. Fig. 4. Again, I'd like to see how changes in % infected are driven by the disease risk itself, and changes in host range (and alternate host) and susceptibility.

Response: Great point. We have added Sup. Fig. 3 below to the supplement that shows the probability of infection as a function of *Ribes* presence/absence and across VPD terciles in our GLMMs. We find that the p(infection) increases with *Ribes* presence, but that the magnitude of this effect declines with decreasing VPD (i.e., at high elevations).

Supplementary Figure 3: Relationship between *Ribes* occurrence and probability of infection estimated from coefficient estimates from GLMMs of the first survey (a) and second survey (b) tree-level infections. Relationship between *Ribes* and probability of infection at the VPD tertiles (VPD is standardized with a mean of 0 and standard deviation of 1) for the first (c) and second survey (d) infections.

Furthermore, Sup. Fig 3 here is using the coefficient estimates from the GLMMs displayed in Fig. 3. The GLMM results demonstrated that *Ribes* was significantly correlated with infections at both time periods, while white pine hosts were included as crossed random effects with plot, as is recommended when sampling designs are stratified by species (see Supplementary Section 4: GLMMs of first and second survey infections for more details). We also find that when we compare the R^2 between GLMMs including white pine hosts as random effects and GLMMs without white pine random effects, that the conditional R^2 doesn't change by a lot (e.g., from 0.684 to 0.613 in the second survey models), suggesting that their influence is relatively minor compared to the factors that we have already discussed in this paper. We expect this is likely because there are sufficient white pine hosts available, resistance is low across all species, and dispersal from white pine occurs well beyond the range of our study.

Additionally, the GLMMs didn't explain what had likely changed between surveys to cause a shift in prevalence. This is where the FE panel model nicely complemented the GLMM analysis by allowing us to rigorously estimate the effect of a change in VPD on the change in prevalence. The results of the FE panel model are what we display in Fig. 4, which is estimating prevalence while controlling for the consistent, time-invariant impacts of *Ribes*, white pine hosts, and other observed and unobserved variables. Finally, Fig. 5 and now Fig. 6 bring it all together to show

how the spatially varying impacts of drought, *Ribes*, and infected host mortality contributed to greater declines at low elevations than increases at high elevations—thus an overall decline in mean disease prevalence.

32. Fig. 5. Panel a clearly shows that the change in prevalence is at low altitudes, not high. That is useful. In all panels, data points should be shown where possible (rather than just statistical summaries).

Response: To address this point, we have added points to panel b in Fig 5 (see below). Our *Ribes*, mortality and tree-level infection data are binary, which makes it challenging to present clearly in a figure, especially when combining multiple results into one figure—essentially the 0/1 point values wouldn't show up clearly when combined. Figure 8 here shows that the graphed binary values of infected white pine host mortality and *Ribes* display a very similar relationship to what we show with the predicted values in Fig. 5 below, though the estimated relationship in Fig. 5 is also accounting for model covariates.

Figure 8: Relationships between elevation and plot-level a) *Ribes* spp. (0 = absence, 1 = presence,) and b) infected white pine host mortality (0 = alive, 1 = dead). Shading showing 95% C.I. around the estimated linear relationship.

Fig. 3: Drought impacts, Ribes occurrence, and infected host mortality across the VPD-elevation gradient. **a**, Observed blister rust prevalence from the first (red) and second (blue) surveys across elevation; includes loess smoothed lines and 95% C.I. Inserted figure shows the percentage point (p.p.) change in prevalence between surveys (second survey-first survey). **b**, Three loess smoothed lines illustrate the mean VPD conditions at the plot-level among the 2012-2015 drought (yellow line), the 2007 drought (orange line), and non-drought years (1995-2006, 2008-2011, 2016; blue line). Boxplots show difference in needle expansion between high and low elevation regions during drought. Needle expansion increased in uninfected whitebark pine ($n = 225$) at high elevation and decreased in uninfected sugar pine ($n = 108$) at low elevation during drought years (non-drought years for needle length were 2012, 2016-2017). **c**, Predicted probability of mortality of infected hosts (black line) and of Ribes occurrence (maroon line) across elevation. Probability of mortality estimated only using infected white pine hosts at the time of the first survey; showing 95% CI.

References

1. Kreyling, J. *et al.* To replicate, or not to replicate – that is the question: how to tackle nonlinear responses in ecological experiments. *Ecol. Lett.* **21**, 1629–1638 (2018).
2. Zuur, A. F., Ieno, E. N., Walker, N., Saveliev, A. A. & Smith, G. M. *Mixed effects models and extensions in ecology with R.* (Springer New York, 2009).
3. Dormann, C. F. Effects of incorporating spatial autocorrelation into the analysis of species distribution data. *Glob. Ecol. Biogeogr.* **16**, 129–138 (2007).
4. Geils, B. W., Hummer, K. E. & Hunt, R. S. White pines, Ribes, and blister rust: a review and synthesis. *For. Pathol.* **40**, 147–185 (2010).
5. Baker-Austin, C. *et al.* Emerging Vibrio risk at high latitudes in response to ocean warming. *Nat. Clim. Change* **3**, 73–77 (2013).
6. Pawley, M. D. M. & McArdle, B. H. Spatial autocorrelation: bane or bonus? *bioRxiv* 385526 (2018) doi:10.1101/385526.
7. Arsdel, E. P. V., Geils, B. W. & Zambino, P. J. Epidemiology for hazard rating of white pine blister rust. *Guyon JC Comp Proc. 53rd West. Int. For. Dis. Work Conf. 2005 Sept. 26-30 Jackson WY US Dep. Agric. For. Serv. Intermt. Reg. Ogden UT* (2006).
8. Schwandt, J. W., Kearns, H. S. J., MARS-DEN, M. & Byler, J. W. White pine blister rust canker expansion on improved western white pine in northern Idaho: Implications for management of rust resistant stock. *USDA Serv Rep R1-13-03 North. Reg. Missoula MT* (2013).
9. Harrell, F. E. *rms: Regression Modeling Strategies.* (2020).
10. Larsen, A. E., Meng, K. & Kendall, B. E. Causal analysis in control–impact ecological studies with observational data. *Methods Ecol. Evol.* **10**, 924–934 (2019).
11. Dudley, J. C. *et al.* Compounding effects of white pine blister rust, mountain pine beetle, and fire threaten four white pine species. *Ecosphere* **11**, e03263 (2020).
12. Schwandt, J. W., Lockman, I. B., Kliejunas, J. T. & Muir, J. A. Current health issues and management strategies for white pines in the western United States and Canada. *For. Pathol.* **40**, 226–250 (2010).
13. Abatzoglou, J. T. & Williams, A. P. Impact of anthropogenic climate change on wildfire across western US forests. *Proc. Natl. Acad. Sci.* **113**, 11770–11775 (2016).
14. Cook, B. I., Mankin, J. S. & Anchukaitis, K. J. Climate Change and Drought: From Past to Future. *Curr. Clim. Change Rep.* **4**, 164–179 (2018).
15. McDowell, N. *et al.* Mechanisms of plant survival and mortality during drought: why do some plants survive while others succumb to drought? *New Phytol.* **178**, 719–739 (2008).
16. Bockino, N. K. & Tinker, D. B. Interactions of white pine blister rust and mountain pine beetle in whitebark pine ecosystems in the southern Greater Yellowstone Area. *Nat. Areas J.* **32**, 31–40 (2012).
17. Stephenson, N. L., Das, A. J., Amperssee, N. J., Bulaon, B. M. & Yee, J. L. Which trees die during drought? The key role of insect host-tree selection. *J. Ecol.* **0**, (2019).
18. Bebb, D. P. Range-Expanding Pests and Pathogens in a Warming World. *Annu. Rev. Phytopathol.* **53**, 335–356 (2015).
19. Soberón, J. & Nakamura, M. Niches and distributional areas: Concepts, methods, and assumptions. *Proc. Natl. Acad. Sci.* **106**, 19644–19650 (2009).

20. Davis, J. K. *et al.* Improving the prediction of arbovirus outbreaks: A comparison of climate-driven models for West Nile virus in an endemic region of the United States. *Acta Trop.* **185**, 242–250 (2018).
21. de la Vega, G. J. & Schilman, P. E. Using eco-physiological traits to understand the realized niche: the role of desiccation tolerance in Chagas disease vectors. *Oecologia* **185**, 607–618 (2017).
22. Talley, S. M., Coley, P. D. & Kursar, T. A. The effects of weather on fungal abundance and richness among 25 communities in the Intermountain West. *BMC Ecol.* **2**, 7 (2002).
23. Manstretta, V. & Rossi, V. Effects of Weather Variables on Ascospore Discharge from *Fusarium graminearum* Perithecia. *PLOS ONE* **10**, e0138860 (2015).
24. Harvell, C. D. *et al.* Climate warming and disease risks for terrestrial and marine Biota. *Science* **296**, 2158–2162 (2002).
25. Gautam, H. R., Bhardwaj, M. L. & Kumar, R. Climate change and its impact on plant diseases. *Curr. Sci.* **105**, 1685–1691 (2013).
26. Bebber, D. P. & Gurr, S. J. Crop-destroying fungal and oomycete pathogens challenge food security. *Fungal Genet. Biol.* **74**, 62–64 (2015).
27. Lukanda, M. *et al.* First Report of Maize chlorotic mottle virus Infecting Maize in the Democratic Republic of the Congo. *Plant Dis.* **98**, 1448–1448 (2014).
28. Brasier, C. M. Intercontinental Spread and Continuing Evolution of the Dutch Elm Disease Pathogens. in *The Elms: Breeding, Conservation, and Disease Management* (ed. Dunn, C. P.) 61–72 (Springer US, 2000). doi:10.1007/978-1-4615-4507-1_4.
29. Boyd, I. L., Freer-Smith, P. H., Gilligan, C. A. & Godfray, H. C. J. The Consequence of Tree Pests and Diseases for Ecosystem Services. *Science* **342**, 1235773 (2013).
30. Chaloner, T. M., Gurr, S. J. & Bebber, D. P. Geometry and evolution of the ecological niche in plant-associated microbes. *Nat. Commun.* **11**, 2955 (2020).
31. Donald, F., Green, S., Searle, K., Cunniffe, N. J. & Purse, B. V. Small scale variability in soil moisture drives infection of vulnerable juniper populations by invasive forest pathogen. *For. Ecol. Manag.* **473**, 118324 (2020).
32. Sturrock, R. N. *et al.* Climate change and forest diseases. *Plant Pathol.* **60**, 133–149 (2011).
33. Pathak, R., Singh, S. K., Tak, A. & Gehlot, P. Impact of Climate Change on Host, Pathogen and Plant Disease Adaptation Regime: A Review. *Biosci. Biotechnol. Res. Asia* **15**, 529–540 (2018).
34. Lafferty, K. D. The ecology of climate change and infectious diseases. *Ecology* **90**, 888–900 (2009).
35. Ghelardini, L., Pepori, A. L., Luchi, N., Capretti, P. & Santini, A. Drivers of emerging fungal diseases of forest trees. *For. Ecol. Manag.* **381**, 235–246 (2016).
36. Wyka, S. A. *et al.* Emergence of white pine needle damage in the northeastern United States is associated with changes in pathogen pressure in response to climate change. *Glob. Change Biol.* **23**, 394–405 (2017).
37. Garrett, K. A. *et al.* Chapter 21 - Plant Pathogens as Indicators of Climate Change. in *Climate Change (Second Edition)* (ed. Letcher, T. M.) 325–338 (Elsevier, 2016). doi:10.1016/B978-0-444-63524-2.00021-X.
38. Bebber, D. P., Ramotowski, M. A. T. & Gurr, S. J. Crop pests and pathogens move polewards in a warming world. *Nat. Clim. Change* **3**, 985–988 (2013).
39. Altizer, S., Ostfeld, R. S., Johnson, P. T. J., Kutz, S. & Harvell, C. D. Climate change and infectious diseases: from evidence to a predictive framework. *Science* **341**, 514–519 (2013).

40. McDonald, G. I., Richardson, B. A., Zambino, P. J., Klopfenstein, N. B. & Kim, M.-S. Pedicularis and Castilleja are natural hosts of Cronartium ribicola in North America: a first report. *For. Pathol.* **36**, 73–82 (2006).
41. Kearns, H. S. J. & Jacobi, W. R. The distribution and incidence of white pine blister rust in central and southeastern Wyoming and northern Colorado. *Can. J. For. Res.* **37**, 462–472 (2007).
42. Maloy, O. C. White pine blister rust control in North America: A case history. *Annu. Rev. Phytopathol.* **35**, 87–109 (1997).
43. Maloy, O. C. White pine blister rust. *Plant Health Prog.* **2**, 10 (2001).
44. Van Arsdel, E. P. Environment in relation to white pine blister rust infection. *Biol. Rust Resist. For. Trees* **1221**, 479–491 (1972).
45. Kinloch, B. B. White pine blister rust in North America: Past and prognosis. *Phytopathology* **93**, 1044–1047 (2003).

REVIEWER COMMENTS

Reviewer #1 (Remarks to the Author):

The authors have made substantial improvement in the revision. However, my most critical concern on their statistical model remains outstanding.

I am still not convinced by the authors' explanation and justification of using quadratic regression in their statistical modeling.

1. Quadratic regression assumes a hump-shape symmetrical response of the biotic variable to the environmental variable(s). This bold assumption of "symmetry" is rarely supported by any ecological data in natural systems. Moreover, the future projection under climate scenario critically depends on the fitted statistical model, which is based on this unjustified assumption of "symmetry". Also, many interpretations and predictions in the manuscript are critically dependent on the results of the quadratic regression model. Therefore, the issue is more than just "identifying existence of nonlinear response", as responded in their rebuttal. While the statistical significance of the quadratic term allows the authors to support their hypothesis of nonlinear response, the use of incorrect (at least, imprecise) statistical model prevents the authors from drawing conclusions and predictions of climate effects, as well as their discussion and proposed potential mechanisms.

2. I am not convinced that GAMM cannot be employed, considering the number of data (their Figure 2). Statistically, I am not convinced that GAMM will use much more degree of freedom than the multivariate quadratic regression in statistical fitting.

3. Reviewer3 echo my concern. He suggested to use Beta function, which is more flexible and does not make assumption of "symmetry". This is also a good alternative solution. So, either Beta function or GAMM is reasonable statistical model. In any case, quadratic regression is not ecologically justified.

Reviewer #2 (Remarks to the Author):

You addressed the earlier review comments very well. Thank you for providing a thorough revision.

Reviewer #3 (Remarks to the Author):

The authors have done a very thorough job revising this manuscript.

The issues with framing of the study have been resolved.

The manuscript is much clearer and easier to read now.

I'm satisfied with the authors' explanation for using VPD rather than humidity & temperature in this statistical analysis, but I don't think I'll be switching to using VPD in mechanistic models.

The new figures are excellent, particularly Figure 1 which is wonderful. I would only suggest that the P(drought) bar could be clearer, but this is a minor issue and the text explains it.

Some suggestions for the Abstract:

- Define "p.p"
- Second sentence uses predictable and unpredictable - perhaps change 'unpredictable' to 'complex' or 'unexpected'.
- Change 'naive hosts' to 'susceptible hosts'
- May want to mention 'white pine' or 'white pine blister rust' in the Abstract, so it is clear what system this study addresses.

Dan Bebber

Nonlinear climate change impacts on infectious tree disease

Manuscript originally submitted to *Nature Communications*, 8/10/2020

Review response submitted, 3/10/2021

REVIEWER 1

Main comment

The authors have made substantial improvement in the revision. However, my most critical concern on their statistical model remains outstanding.

1. Quadratic regression assumes a hump-shape symmetrical response of the biotic variable to the environmental variable(s). This bold assumption of “symmetry” is rarely supported by any ecological data in natural systems. Moreover, the future projection under climate scenario critically depends on the fitted statistical model, which is based on this unjustified assumption of “symmetry”. Also, many interpretations and predictions in the manuscript are critically dependent on the results of the quadratic regression model. Therefore, the issue is more than just “identifying existence of nonlinear response”, as responded in their rebuttal. While the statistical significance of the quadratic term allows the authors to support their hypothesis of nonlinear response, the use of incorrect (at least, imprecise) statistical model prevents the authors from drawing conclusions and predictions of climate effects, as well as their discussion and proposed potential mechanisms.

Response: We agree that non-parametric approaches, such as General Additive Models, enable greater flexibility in modeling functional relationships with an unknown underlying structure. Nevertheless, GAM/GAMMs also face several challenges that may at times offset this benefit. Since previous studies have found both symmetric and asymmetric relationships between climate and disease (Austin, 2007; Meynard & Quinn, 2007) (Fig. 1 below), we believe it is important to carefully consider the specific details of our study in assessing the appropriate model structure. Below we outline that (a) infections in our study system do exhibit symmetric patterns; (b) GAMs with a large number of knots tend to overfit data in our setting; (c) GAMs with a small number of knots tend to generate similar results derived from quadratic terms, and (d) a linear mixed model with a quadratic term yields a more parsimonious model than the GAMMs we considered. We discuss each of these points in greater detail below.

Fig. 1 from Mordecai et al., (2013) showing data validated thermal performance curves for mosquito and parasite life history traits. Both asymmetric and symmetric performance curves were represented.

Fig. 2. from Fig 3 in main paper: Nonlinear relationship between blister rust and VPD. a,c, show loess smoothed lines of relatively symmetrical, nonlinear relationships.

(a) *infections in our study system do exhibit symmetric patterns:* In this study, the infection-climate response function is likely more symmetric than highly asymmetric, so that a quadratic function can fit it reasonably well. In Fig. 3 of our paper (see Fig. 2 above), we used a loess smoother to visualize the relationship between VPD and infection. This smoother (often used in GAMs) is flexible and could take on less symmetric shapes if that would fit the data better, but instead it looks approximately symmetrical (see Fig. 2).

Fig 3. Displaying the difference between three GAM splines to loess splines between the first and second survey data.

b) GAMs with a large number of knots tend to overfit data in our setting: Fig. 3 here shows the comparison between loess and GAM splines with variable knots (“k”; knots). With a moderate amount of flexibility, the GAM response shape is equivalent to the loess smoother, and remains fairly symmetrical and therefore reasonably consistent with a quadratic function. If we allow very flexible response shapes, the GAM overfits with a complex response shape, especially as the number of observations declines with increasing VPD, likely reflecting the declining number of plots with increasing VPD, as well as randomness across VPD, than a biologically meaningful relationship.

Fig 4. Displaying the marginal effects of VPD on blister rust estimated from logistic GLMMs used in this paper (see Fig 2b,d above). The symmetry from the quadratic term is very similar to the loess fits and the GAM fits using $k=5$.

(c) GAMs with a small number of knots tend to generate similar results derived from quadratic terms. The marginal effects Fig. 4 of VPD from logistic GLMMs in the first and second survey illustrate that the quadratic term is estimating a similar, fairly symmetric curve to the loess and

less flexible gam ($k=5$) in Fig. 3 above. Thus, we conclude that the quadratic is performing similarly to a less flexible GAMM.

(d) a linear mixed model with a quadratic term yields a more parsimonious model than the GAMMs we considered. Beyond this visual comparison above, we also tested whether the additional flexibility in GAMM models provide a more precise estimate of the relationship between VPD and infections. Specifically, we estimated logistic mixed effects models with a variable flexibility ($k=5, 10, 20$) and compared these models using AIC to logistic mixed effects models with a linear and quadratic VPD term using the function `gamm4` from package `gamm4`, which uses `lme4` as the underlying fitting engine and is precisely comparable to our logistic mixed effects models presented in the paper (Fig. 2b,d). We found that inclusion of the quadratic VPD term resulted in the preferred model both in the first and second survey (Tables 1 and 2 below).

Comparing GAMMs using AIC

First survey

Model	AIC
linear VPD term	1996.439
quadratic VPD term	1986.924
GAMM, $k=5$	1996.79
GAMM, $k=10$	1996.082
GAMM, $k=20$	1996.11

Table 1: First survey logistic model comparison among logistic mixed models with variable knots (k) and a quadratic and linear term.

Second survey

Model	AIC
linear VPD term	1110.629
quadratic VPD term	1100.542
GAMM, $k=5$	1107.049
GAMM, $k=10$	1107.505
GAMM, $k=20$	1107.649

Table 2: Second survey model comparison among logistic mixed models with variable knots (k).

Given that the added flexibility of GAMM does not improve the model, and that qualitatively, the shape of the GAM response is consistent with the symmetrical quadratic response we use in our main analysis, we conclude that the quadratic response is reasonable for this data set.

To include reference to this analysis, we have added the following statement to the methods that includes this analysis:

Lines 511-513:

We also found that the symmetry of the quadratic VPD term in the GLMMs was reasonable using a non-parametric comparison approach (Supplementary Figure 9).

And we have included Fig. 3 above in the supplement, Supplementary Figure 9.

2. I am not convinced that GAMM cannot be employed, considering the number of data (their Figure 2). Statistically, I am not convinced that GAMM will use much more degree of freedom than the multivariate quadratic regression in statistical fitting.

Response: As discussed in response to the previous reviewer comment, we decided to go ahead and try fitting GAMMs to the data. The results as discussed above in the previous response were consistent with a relatively symmetric relationship between VPD and disease incidence.

3. Reviewer3 echo my concern. He suggested to use Beta function, which is more flexible and does not make assumption of “symmetry”. This is also a good alternative solution. So, either Beta function or GAMM is reasonable statistical model. In any case, quadratic regression is not ecologically justified.

Response: We agree that assuming a response shape is symmetrical is often not ecologically justified. On the other hand, sometimes response shapes for abundance vs climate do turn out to be symmetrical, as some of Austin’s work shows (see Fig 1). We appreciate the motivation to fully investigate this assumption. After considering multiple different GAM splines in addition to the loess splines already presented in the paper, and comparing GAMM models using AIC, we concluded that the assumption of symmetry is a reasonable modeling choice for this data set. We have now included a discussion of this analysis and Fig 3 in the paper (see response to comment 1 above).

REVIEWER 2

You addressed the earlier review comments very well. Thank you for providing a thorough revision.

Response: Thank you very much and thank you for your support in making this a stronger paper.

REVIEWER 3

1. The authors have done a very thorough job revising this manuscript. The issues with framing of the study have been resolved. The manuscript is much clearer and easier to read now.

Response: We are very glad to hear this and we thank you for helping to greatly improve the manuscript.

2. I'm satisfied with the authors' explanation for using VPD rather than humidity & temperature in this statistical analysis, but I don't think I'll be switching to using VPD in mechanistic models.

Response: Thank you and we agree that VPD may not be appropriate in other settings.

3. The new figures are excellent, particularly Figure 1 which is wonderful. I would only suggest that the P(drought) bar could be clearer, but this is a minor issue and the text explains it.

Response: We have added dots across the landscape to make the P(Drought) clearer, as follows:

We have also added the same dots but within the estimated elevation range of drought stress in this study in Fig. 5. We then added portions of the data presented in the previous Fig 5 to show the spatially specific components of the results and better match the data to the concepts presented in Fig 1, rather than keeping Fig 5 as a more conceptual piece. Finally, we better matched the pathways 1 and 2 in Fig 5 to the text in the main article to make it easier for readers. Since Fig 5 and 6 were presenting similar information, we have moved Fig. 5 to the supplement and kept only Fig 6 (now Fig. 5 in the main paper).

Fig. 1. Spatially varying abiotic-biotic interactions that likely modified blister rust range shifts. a,b, Predicted probability of *Ribes* occurrence (blue line) and mortality of infected hosts (black line) across elevation. Probability of mortality estimated only using infected white pine hosts at the time of the first survey; showing 95% C.I. **c,d,** Boxplots (displaying the median horizontal line, upper and lower quartile whiskers, outlier points in black, and the data points in color) show difference in needle expansion between high and low elevation regions during and following drought. Needle expansion increased in uninfected whitebark pine at high elevation and decreased in uninfected sugar pine at low elevation during drought years (drought years included: 2013-2015; non-drought years included: 2016-2017). **e,** Observed blister rust prevalence from the first (red) and second (blue) surveys across elevation; includes loess smoothed lines and 95% C.I. Blue shade on mountain displays the declining abundance of *Ribes* hosts with increasing elevation. Yellow dots show the predicted range that drought led to decreases in white pine host needle growth. Black lines on the mountain illustrate the ranges of the susceptible white pine hosts. Mountain figure designed by Zuzanna Drozd.

4. Some suggestions for the Abstract: Define "p.p"

Response: Yes, that would be ideal, but we are over word limit and because p.p. is a very commonly used abbreviation and we define it in the main text, we decided to keep in the other critical words that describe the study.

5. Second sentence uses predictable and unpredictable - perhaps change 'unpredictable' to 'complex' or 'unexpected'.

Response: We changed it to "complex".

6. Change 'naive hosts' to 'susceptible hosts'

Response: We changed it.

7. May want to mention 'white pine' or 'white pine blister rust' in the Abstract, so it is clear what system this study addresses.

Response: We used the Latin name for the pathogen *Cronartium ribicola* instead since it uses fewer words.

REVIEWER COMMENTS

Reviewer #4 (Remarks to the Author):

Review of Nonlinear climate change impacts on infectious tree disease

I was asked to review this paper focussing specifically on the issue of the adequacy of the authors' quadratic model relative to a generalized additive mixed model, as suggested by a referee not available at this round.

Firstly two points for the journal.

1. It must be at least 20 years since an author last submitted a manuscript produced on a manual type-writer to which they had to staple the figures and tables. Why, in that case, are you still insisting on (or is it just allowing?) figures and tables being at the end of the manuscript, which is quite inconvenient for the reviewer? It seems to me to be discourteous to reviewers who are doing this for free.
2. The paper format requirements that treat every paper as if it was the report of the results of an experiment, with the 'methods' in a section at the end, are unhelpful for papers based primarily on statistical modelling. You force the scrupulous/allow the unscrupulous to obscure what they have actually done. Statistical models are better stated clearly and mathematically in the main text, if the intention is that the analysis should be understandable.

On to the paper.

The authors' responses to the referee's suggestion of using generalized additive mixed models are difficult to follow, but do not appear to me to make much sense. It is hard to be certain, as the models actually fitted are not written down, and the code supplied in the referenced github repository does not appear to contain these models (or the loess smooth model used in figure 3 of the main paper). However, a sensible smooth model for these data should not be predicting any negative proportions, as the main paper figure 3 loess smooth and the smooth models shown in figure 3 of the response to referees, both do.

My guess is that these are somehow direct smooths of proportions data, ignoring other covariates, which is not at all what the referee was suggesting.

Whatever has been done for the plots, something else must have been done to get the AICs in tables 1 and 2 of the response to referees - the radically different smooths are not consistent with the very similar AIC scores.

Conversely, if it was really the case that the wildly different smooths in the plots all gave almost the same AIC, then the argument that the data somehow offer good support for the single humped model would be demolished, since it would then be clear that the data are simply uninformative about this feature.

I suspect that the referee had in mind that you do something like the following, after the pre-processing in the github repository file `GLMMs_pathogen.R`

```
library(mgcv)
dat95$plot <- factor(dat95$plot);dat95$species <- factor(dat95$species)

m952 <- gam(inc~poly(vpd,2)+density+ribes+dbh+slope+aspect+s(plot,bs="re")
+s(species,bs="re"),data=dat95,family=binomial,method="ML")

m953 <- gam(inc~poly(vpd,3)+density+ribes+dbh+slope+aspect+s(plot,bs="re")
+s(species,bs="re"),data=dat95,family=binomial,method="ML")

m95s <- gam(inc~s(vpd,k=20)+density+ribes+dbh+slope+aspect+s(plot,bs="re")
+s(species,bs="re"),data=dat95,family=binomial,method="REML")

AIC(m952,m953,m95s)
#           df           AIC
# m952 65.23537 1895.573
# m953 63.69974 1892.744
# m95s 65.91545 1891.944

dat00$plot <- factor(dat00$plot);dat00$species <- factor(dat00$species)

m002 <- gam(inc~poly(vpd,2)+density+ribes+dbh+slope+aspect+s(plot,bs="re")
+s(species,bs="re"),data=dat00,family=binomial,method="ML")

m003 <- gam(inc~poly(vpd,3)+density+ribes+dbh+slope+aspect+s(plot,bs="re")
+s(species,bs="re"),data=dat00,family=binomial,method="ML")
```

```
m00s <- gam(inc~s(vpd,k=20)+density+ribes+dbh+slope+aspect+s(plot,bs="re")
+s(species,bs="re"),data=dat00,family=binomial,method="REML")
```

```
AIC(m002,m003,m00s)
#      df      AIC
# m002 63.39999 1052.336
# m003 64.15482 1053.309
# m00s 67.10421 1054.046
```

```
par(mfrow=c(2,3))
plot(m952,all.terms=TRUE,select=3);plot(m953,all.terms=TRUE,select=3);
plot(m95s,select=1)
plot(m002,all.terms=TRUE,select=3);plot(m003,all.terms=TRUE,select=3);
plot(m00s,select=1)
```

The code compares models with a quadratic, cubic and smooth dependence on `vpd` for the two surveys. The models all have the form

$$\text{logit}\{E(\text{inc}_i)\} = \beta_0 + f(\text{vpd}_i) + \beta_1 \text{density}_i + \beta_2 \text{ribes}_i + \beta_3 \text{dbh}_i + \beta_4 \text{slope}_i + \beta_5 \text{aspect}_i + a_{p(i)} + b_{s(i)},$$

where inc_i has a Bernoulli distribution, the β_j are parameters to estimate, f is a smooth function to estimate (quadratic, cubic, or a spline), $p(i)$ indexes the plot for measurement i and $s(i)$ the species. a_j and b_j are simple Gaussian random effects, each with its own variance parameter.

In each case the AIC differences are not large, but for the first survey the smooth or cubic models do seem to be a bit better than the quadratic. For the second survey there is even less difference, but the quadratic model seems marginally preferable to the cubic. The first row of the plot shows the first survey `vpd` estimated effects (linear predictor scale), for the quadratic, cubic and smooth models. The second row is for the second survey.

These model fits are clearly not showing strong support for the quadratic model relative to alternatives, so it would be prudent to at least check that the paper's results are robust when repeated with a cubic dependence on `vpd` in place of the assumed quadratic. This would directly address the referee's point about the symmetry assumption, by relaxing it.

In my view the code in the github repository falls short of what is needed for reproducibility. It would be good to fix this.

Nonlinear climate change impacts on infectious tree disease

Manuscript originally submitted to *Nature Communications*, 8/10/2020

Review response submitted, 5/7/2021

REVIEWER 1

Main comments

1. I was asked to review this paper focusing specifically on the issue of the adequacy of the authors' quadratic model relative to a generalized additive mixed model, as suggested by a referee not available at this round.

Response: Thank you for your help with improving the quality and rigor of this manuscript.

2. Firstly two points for the journal. It must be at least 20 years since an author last submitted a manuscript produced on a manual type-writer to which they had to staple the figures and tables. Why, in that case, are you still insisting on (or is it just allowing?) figures and tables being at the end of the manuscript, which is quite inconvenient for the reviewer? It seems to me to be discourteous to reviewers who are doing this for free.

Response: We agree it is easier to read embedded figures. We have now included embedded figures throughout the manuscript.

3. The paper format requirements that treat every paper as if it was the report of the results of an experiment, with the 'methods' in a section at the end, are unhelpful for papers based primarily on statistical modelling. You force the scrupulous/allow the unscrupulous to obscure what they have actually done. Statistical models are better stated clearly and mathematically in the main text, if the intention is that the analysis should be understandable.

Response: We have placed more of the details of the glmm model description from the supplement into the main paper on lines 580-594.

“To explain the probability of infection at the time of the first and second survey, we developed two GLMMs that estimated: 1) probability that a tree was infected at the time of the first survey using first-survey infection status of all live trees as the (0/1) response variable ($n = 7,031$); and 2) probability that an uninfected tree became infected between the first and second surveys using second survey infection status of live, previously uninfected trees as the (0/1) response variable ($n = 5,416$). For both GLMMs, the explanatory variables included a quadratic VPD term to allow for a potentially nonlinear disease response and the previously listed independent variables; we included species and plot as crossed random effects. All non-binary variables were standardized across both survey periods data to have a mean of zero and a standard deviation of 1. We only included live trees that were uninfected at the time of the first survey (t_1) in the second survey models (t_2) in part to test whether unmeasured variables (e.g., stochastic dispersal dynamics, pathogen climate adaptation, and/or changes in virulence) may have affected the probability of infection following the first survey (**Error! Reference source not**

found.) We also verified that models that included all trees (i.e., both live and dead trees) gave similar coefficient estimates and p-values (**Error! Reference source not found.**)”

We mathematically described our fixed effects panel model in the main text on lines 650-673:

“Specifically, the FE estimated the relationship between changes in VPD and changes in blister rust between surveys(Larsen et al., 2019) (Fig. 1e,f). Our outcome variable was the share of live trees in plot i at time t that were infected by blister rust ($y_{it} = \text{\#infections}/\text{total live trees}$). We modeled this outcome as a function of plot-level independent variables, including slope, aspect, tree density (#live trees/ha), VPD, VPD², elevation, *Ribes* spp., plot, white pine species, and DBH. We derived the FE equation as follows(Wooldridge, 2016):

$$y_{it} = \beta_1 x_{it} \dots + a_i + \delta_t + u_{it}, t = 1,2 \quad (1)$$

for each plot (i), we averaged the equation over time (t), yielding:

$$\bar{y}_i = \beta_1 \bar{x}_i \dots + \delta + a_i + \bar{u}_i \quad (2)$$

Subtracting (2) from (1):

$$\dot{y}_{it} = \beta_1 \dot{x}_{it} \dots + \dot{\delta}_t + \dot{u}_{it}, t = 1,2 \quad (3)$$

where $\dot{y}_i = y_{it} - \bar{y}_i$ is the time-demeaned blister rust prevalence, $\dot{\delta}_t$ is the time fixed effect, and $\beta_1 \dot{x}_i = \beta_1 x_{it} - \beta_1 \bar{x}_i \dots$ represent the time-demeaned fixed effects (e.g., plot, white pine species, VPD). Subtracting the mean value from each dependent and independent variable mathematically subtracted out the observed time-invariant variables (including plot, elevation, slope, white pine host, and *Ribes* spp.), and the unobserved time-invariant variables (a_i). In this way, the panel model allowed us to control for time-invariant but unobservable variables which might otherwise confound our analysis(Larsen et al., 2019) of the relationship between VPD and blister rust. What remained in the model were the time-varying variables (i.e., changes in tree size (growth or DBH), changes in tree density (due to mortality), and changes in VPD), as well as the exogenous time-varying error (\dot{u}_{it}) (Equation 3).”

4. The authors’ responses to the referee’s suggestion of using generalized additive mixed models are difficult to follow, but do not appear to me to make much sense. It is hard to be certain, as the models actually fitted are not written down, and the code supplied in the referenced github repository does not appear to contain these models (or the loess smooth model used in figure 3 of the main paper). However, a sensible smooth model for these data should not be predicting any negative proportions, as the main paper figure 3 loess smooth and the smooth models shown in figure 3 of the response to referees, both do.

Response: Thanks for pointing this out. We have changed our loess smoothed lines in Fig. 3 to logistic regression smooths and we have updated the repository to include the figure code.

Fig. 1: Nonlinear relationship between blister rust and VPD. **a,c** Proportion of infected white pines (smoothed line estimated using a generalized linear logistic regression with a quadratic term) and the number of tree stems across VPD (green bars) for the first (1995-1999) and second (2013-2016) surveys; shaded region represents the standard error band. **b,d** Coefficient estimates from logistic regression models explaining tree-level blister rust infections from the first and second surveys with standard error bars. **e** Plot-level change in prevalence (proportion infected/total live stems) as a function of the change in VPD between surveys. Plots were divided into VPD terciles defined at t_2 , thereby demonstrating both an increase in blister rust prevalence in cold regions (low VPD) and a decrease in prevalence in more arid regions (higher VPD); showing shaded standard error bands. **f** Coefficient estimates of four explanatory variables from the FE panel model with standard error bars. ****** $p < 0.01$, ***** $p < 0.05$.

We have also removed the GAM spline figure in the supplementary section and changed all figures with binary data from linear smooths to logistic smooths.

- My guess is that these are somehow direct smooths of proportions data, ignoring other covariates, which is not at all what the referee was suggesting. Whatever has been done for the plots, something else must have been done to get the AICs in tables 1 and 2 of the response to referees - the radically different smooths are not consistent with the very similar AIC scores. Conversely, if it was really the case that the wildly different smooths in the plots all gave almost the same AIC, then the argument that the data somehow offer good support for the single humped model would be demolished, since it would then be clear that the data are simply uninformative about this feature.

Response: This is a good point. We used the function `gamm4` from R package `gamm4` to estimate the different models. The function `gamm4` was selected because it uses `lme4` as the underlying fitting engine and therefore is precisely comparable our `lme4` models presented elsewhere in the paper. In addition, in our case, `gamm4` is considered to be more robust than `gamm` because it “gives better performance for binary and low mean count data (Wood & Scheipl, 2020);” our data are binary. However, `gamm4`, as you noted, produced very similar

AICs even with variable knots that could allow for more vs less flexibility in the fit. Though gamm4 does allow specification of k values, it does *not* appear to estimate them very differently but rather optimizes the spline given the variable knots. Upon closer evaluation of the gamm4 function we find that: “Its main disadvantage is that it cannot handle most multi-penalty smooths (i.e. not te type tensor products or adaptive smooths) (Wood & Scheipl, 2020).” We have now shifted to include a more flexible cubic lme4 glmm model as you suggested and are no longer using gamm4 (see response 6 below).

6. I suspect that the referee had in mind that you do something like the following, after the pre-processing in the github repository file GLMMs pathogen.R. In each case the AIC differences are not large, but for the first survey the smooth or cubic models do seem to be a bit better than the quadratic. For the second survey there is even less difference, but the quadratic model seems marginally preferable to the cubic. The first row of the plot shows the first survey vpd estimated effects (linear predictor scale), for the quadratic, cubic and smooth models. The second row is for the second survey.

Response: Thank you – this is indeed what we were intending but used the gamm4 package instead of the mgcv package. Using gamm4 to compare AIC values across the quadratic, cubic, and default spline logistic regression models, we find that the gamm4 results are more similar to our glmer model results than gam from mgcv, as expected, given the functions gamm4 and glmer use the same lme4 underlying fitting engine. We have provided updated tables here comparing the AIC values across the three different R packages: 1) mgcv, 2) gamm4, and 3) lme4. For the paper, we are using lme4 and including the more flexible cubic term as you suggested to address the previous reviewer’s concerns about functional form (see response 7 below).

First survey disease model

Model	AIC mgcv	AIC gamm4	AIC lme4
Quadratic VPD term	1895.6	1986.9	1986.9
Cubic VPD term	1892.7	1986.9	1986.9
Default spline GAMM	1891.5	1996.1	NA

Second survey disease model

Model	AIC mgcv	AIC gamm4	AIC lme4
Quadratic VPD term	1052.3	1100.5	1100.5
Cubic VPD term	1053.3	1102.4	1102.4
Default spline GAMM	1053.9	1107.5	NA

7. These model fits are clearly not showing strong support for the quadratic model relative to alternatives, so it would be prudent to at least check that the paper’s results are robust when repeated with a cubic dependence on vpd in place of the assumed quadratic. This would directly address the referee’s point about the symmetry assumption, by relaxing it.

Response: Thank you for the clear statement of the problem and how to check the robustness of the result. We have now fitted a cubic term both for the GLMM models and checked the AIC of both quadratic and cubic fits (see tables above), as well as the qualitative shape of the response for both cubic and quadratic models (see Fig. R1). The results are consistent with your evaluation in the review. First, the AIC scores of lme4 GLMM quad and cubic models are similar to your analysis. As you also note, the data don't strongly support one over the other in the first survey, but quadratic is slightly preferable in the second survey. Second, the shapes are roughly "hump-shaped", particularly for the second survey infections (we have also included this analysis described here in our online repository).

Fig. R1. Showing the difference in response functions using a quadratic vs cubic VPD terms in GLMM lme4 models (first and second survey models). See script “GLMMs_pathogen.R” in the github repository.

Given that the quadratic model is sometimes, but not consistently, preferable to the cubic model, we follow the reviewer’s excellent advice and test the robustness of our paper’s key results to the alternate cubic model specification. Specifically, we have added a table of the cubic model results to the supplement (Supplementary Table 5). We have also tested whether the cubic relationship significantly shifts through time. One of our objectives for the GLMM models was to test whether the VPD coefficients changed significantly between the first and second surveys. A significant change through time would provide an indication of whether unmeasured variables may be biasing our coefficient estimates of VPD. We have conducted the same test using the

cubic GLMM model and found that the interactions between time period and VPD, VPD² and VPD³ were not significant, suggesting that our previous conclusions of the quadratic GLMM models were consistent when using a more flexible cubic term.

Cubic model testing interaction with time			
Predictors	Coefficient	Std. Error	P-Value
Intercept	-5.55	1.05	< 0.001
VPD	2.71	0.54	< 0.001
Time	-0.26	0.32	0.414
VPD ²	-1.06	0.51	0.037
VPD ³	0.06	0.17	0.711
Ribes	1.14	0.39	0.003
Aspect	-0.24	0.19	0.206
Slope	-0.13	0.16	0.416
Density	-0.62	0.18	0.001
DBH	-0.31	0.07	< 0.001
VPD*Time	-0.73	0.59	0.211
VPD ² *Time	0.62	0.58	0.284
VPD ³ *Time	-0.11	0.16	0.490
Density*Time	1.03	0.20	< 0.001
DBH*Time	0.41	0.10	< 0.001
N _{species}	4		
Observations	12447		
AIC	3214.5		

Table R1: Logistic regression estimates, standard errors, and corresponding p-values for GLMM estimating infected trees with time (first and second survey) as interaction terms. The relationship between blister rust and VPD, VPD² and VPD³ is not significantly different between the first and second surveys (interaction terms were not significant), suggesting that the relationship between infections and climate has remained relatively stable through time.

We have also added Supplementary Section 5 and Supplementary Figure 2 that compare our FE panel model results using the cubic and quadratic VPD terms:

Supplementary Section 1: Comparing quadratic and cubic models

Although our primary model specification imposes a quadratic functional form to estimate the relationship between VPD and blister rust incidence, we sought to test the robustness of these results to a more flexible functional form. To do this, we compared our estimates of the climate change effect over the past ~20 years between FE panel models with a quadratic and cubic VPD term. The outcome variable (the share of live trees in plot i at time t that were infected by blister rust) was modeled as a function of variables that changed through time, including tree density (#live trees/ha), VPD, VPD², VPD³, and DBH. Specifically the panel model had the form:

$$\ddot{y}_{it} = \beta_1 \ddot{x}_{it} \dots + \ddot{\delta}_t + \ddot{u}_{it}, t = 1,2 \quad (S1)$$

where $\ddot{y}_i = y_{it} - \bar{y}_i$ was the time-demeaned blister rust prevalence, $\ddot{\delta}_t$ was the time fixed effect, and $\beta_1 \ddot{x}_i = \beta_1 x_{it} - \beta_1 \bar{x}_i \dots$ represented the time-demeaned fixed effects. See methods section “Estimating the relationship between climate and disease prevalence” for the full model description that matched the quadratic model except for the VPD³ term.

Whether using quadratic or cubic models, the broader interpretation of the results was similar. Though the magnitude of the climate change effect differs between models (Supplementary Figure 2), the direction of the climate change effect does not change. For example, when we estimated the fixed effects panel model using a cubic VPD term, we found that the direction of observed climate change effect was the same, though the magnitude of the climate change effect was, on average, greater at higher elevations and less at low elevations. Specifically, the quadratic model estimated that climate change increased prevalence by 6.8 (5.8, 7.9) p.p. in colder regions (high elevations), while the cubic model estimated a 9.2 (8.0, 10.4). Furthermore, the quadratic model estimated that climate change decreased prevalence by 5.5 (4.4, 6.6) p.p. in arid regions (low elevations), while the cubic model estimated a 3.5 (2.2, 4.7) p.p. However, the estimated range expansion under observed climate change did not vary greatly between the cubic and quadratic models. The quadratic model estimated that warmer conditions over the past twenty years likely extended the prevalence into higher elevations by 777.9 (1.0, 1392.9) km², while the cubic model estimated a 793.7 (0.96, 1393.0) km² range expansion. Thus, the overall interpretation of our model results – that climate change led to both increases and decreases in prevalence across elevation and a range expansion of disease – does not change between the quadratic and cubic models.

Supplementary Figure 1: Comparing the estimated climate change effect on blister rust prevalence across elevation tercles between FE panel models that included a quadratic term (“Quadratic model;” grey bars) and cubic VPD term (“Cubic model;” black bars). Values are differences in predicted prevalence (percentage point (p.p.)) attributed to a change in VPD estimated by the MC simulation for the climate change 2016 scenarios. Climate change predicted prevalence was differenced from the counterfactual predicted prevalence at the three elevation tercles (see methods section “Estimating the relationship between climate and disease prevalence”). Displaying standard error bars.

We have included this analysis and further descriptions of this analysis in our main paper methods as well, including:

Lines 610-616:

“To determine whether a quadratic VPD term or a more flexible cubic term was preferable in this study, we fit first and second survey GLMMs described above with a cubic VPD term and compared models using AIC (**Error! Reference source not found.**). We found that the quadratic term resulted in a slightly more parsimonious model in the second survey and quadratic and cubic terms were equivalent in the first survey models. Thus, we used a quadratic VPD term for all disease models in this study and verified that our results were robust using a cubic VPD term (**Error! Reference source not found.**, Supplementary Figure 1).”

Lines 729-734:

“We also estimated the effect of observed climate change (“climate change 2016”) on blister rust prevalence and the corresponding range shift using the estimated parameters from a FE panel model that included a cubic term. We found the interpretation of our observed climate

change results were consistent between models (**Error! Reference source not found.**, Supplementary Figure 1).”

8. In my view the code in the github repository falls short of what is needed for reproducibility. It would be good to fix this.

Response: Yes, we agree, we had not fully updated the code to include the figure and analyses. We have now included the analyses and main figures that were coded in R in the github repository. In addition, we have added detailed annotations and five more scripts to ensure that the repository provides a more useful tool to enable reproduction and understanding of our empirical approach.

Works Cited:

- Larsen, A. E., Meng, K., & Kendall, B. E. (2019). Causal analysis in control–impact ecological studies with observational data. *Methods in Ecology and Evolution*, *10*(7), 924–934. <https://doi.org/10.1111/2041-210X.13190>
- Wood, S., & Scheipl, F. (2020). *gamm4: Generalized Additive Mixed Models using “mgcv” and “lme4”* (0.2-6) [Computer software]. <https://CRAN.R-project.org/package=gamm4>
- Wooldridge, J. M. (2016). *Introductory econometrics: A modern approach*. Nelson Education.

REVIEWERS' COMMENTS

Reviewer #4 (Remarks to the Author):

I have been over the revised paper and responses to the previous comments quite briefly due to pressure of time. The responses are for the most part reasonable, but I have one major comment.

Statistically there is really nothing to say that the quadratic model is better than the cubic model, but some of the key results quoted in the paper (e.g. the prevalence shifts quoted in the abstract) are very different between the two models. I do not understand why one would only quote the results from the quadratic model as if these were 'true'.

What do you actually believe to be true here? I can't see any reason to favour the results from the quadratic model over those from the cubic model, so it would be better to acknowledge the uncertainty in the model structure somehow. Model averaged results would be one possibility. e.g. <https://doi.org/10.2307/2533961>

This point is doubly important because the AICs quoted for the lme4 based models are quite problematic. They are 'marginal AIC' values computed using the model marginal likelihood - such AICs are known to over favour simpler models because Marginal Likelihood optimization underestimates variance components. Somehow plugging in REML here does not fix the problem because of REML not being comparable between models with different fixed effects structure. A conditional AIC (the ones quoted from mgcv are of this type) gets around this problem, but are still not perfect, and differences of the size obtained here are still not conclusive (especially at survey 2). See <https://doi.org/10.1093/biomet/asz051> for detail on the issues.

I realize that Nature journals tend to favour rather emphatic statements of results, but I think that the resultant tendency to understate uncertainty is at best a disservice to the wider society that funds this science, and at worst can be deeply damaging.

Nonlinear shifts in infectious rust disease due to climate change

Manuscript originally submitted to *Nature Communications*, 8/10/2020

Review response submitted, 6/18/2021

REVIEWER 4

Main comments

1. I have been over the revised paper and responses to the previous comments quite briefly due to pressure of time. The responses are for the most part reasonable, but I have one major comment.

Statistically there is really nothing to say that the quadratic model is better than the cubic model, but some of the key results quoted in the paper (e.g. the prevalence shifts quoted in the abstract) are very different between the two models. I do not understand why one would only quote the results from the quadratic model as if these were 'true'.

What do you actually believe to be true here? I can't see any reason to favour the results from the quadratic model over those from the cubic model, so it would be better to acknowledge the uncertainty in the model structure somehow. Model averaged results would be one possibility. e.g. <https://doi.org/10.2307/2533961>

Response: We have acknowledge the uncertainty in the model structure as follows:

1. We have included multiple statements in the main text that describes the model selection, where to find the model results, and how to interpret the differences in magnitude of the climate change effect when using a cubic vs quadratic model (see comment 2 below).
 2. We have also concluded that the added flexibility of the cubic form does not necessarily inform the biological reality of this pathosystem and may be capturing stochastic effects of sampling design instead—see Kreyling et al. 2018¹.
 3. In the abstract and main text, we have modified p.p. change values with “estimated” and we have included modifiers such as “likely” to highlight these are estimated not absolute results and values.
-
2. This point is doubly important because the AICs quoted for the lme4 based models are quite problematic. They are 'marginal AIC' values computed using the model marginal likelihood - such AICs are known to over favour simpler models because Marginal Likelihood optimization underestimates variance components. Somehow plugging in REML here does not fix the problem because of REML not being comparable between models with different fixed effects structure. A conditional AIC (the ones quoted from mgcv are of this type) gets around this problem, but are still not perfect, and differences of the size obtained here are still

not conclusive (especially at survey 2). See <https://doi.org/10.1093/biomet/asz051> for detail on the issues.

Response: Thank you for this reference and this important point. We have decided to include both quadratic and cubic model results in our paper, the former in the main text and the latter in the supplement, with clearly stated caveats about the interpretation of the quadratic model in the main text.

Lines 241-246: “Finally, when we compared the performance of models with more flexible functional forms of VPD (e.g., quadratic vs. cubic terms), we found the quadratic to return the lowest AIC. Nevertheless, this difference was marginal and inconsistent across surveys (Supplementary Table 5). Therefore, we reproduced our primary results using a cubic VPD term and include these results in the supplement (Supplementary Figure 2).”

Lines 268-274: “Using a more flexible VPD term (cubic) resulted in a similar increase at high elevations and decrease at low elevations; however, the cubic model estimated an even greater increase at high elevations and a more moderate decrease at low elevations (Supplementary Figure 2). While the direction of the results were consistent between quadratic and cubic models, the magnitude of the climate change effect was sensitive to changes in functional form. Therefore, the magnitude of the climate change effect should be interpreted cautiously.”

3. I realize that Nature journals tend to favour rather emphatic statements of results, but I think that the resultant tendency to understate uncertainty is at best a disservice to the wider society that funds this science, and at worst can be deeply damaging.

Response: As you may have noted, we have included qualifiers such as “likely”, “estimated”, “may have” throughout the manuscript to avoid overly emphatic statements about our results.

References:

1. Kreyling, J. *et al.* To replicate, or not to replicate – that is the question: how to tackle nonlinear responses in ecological experiments. *Ecol. Lett.* **21**, 1629–1638 (2018).